# Dynamic neural representations of memory and space during human ambulatory navigation

Sabrina L. L. Maoz[1,2,3], Matthias Stangl [3], Uros Topalovic [3], Daniel Batista[3], Sonja Hiller [3], Zahra M. Aghajan[3], Barbara Knowlton [4], John Stern[5], Jean-Philippe Langevin [6,7], Itzhak Fried [3,7,8], Dawn Eliashiv[5] & Nanthia Suthana [1,3,4,7] ✉

Our ability to recall memories of personal experiences is an essential part of daily life. These episodic memories often involve movement through space and thus require continuous encoding of one's position relative to the surrounding environment. The medial temporal lobe (MTL) is thought to be critically involved, based on studies in freely moving rodents and stationary humans. However, it remains unclear if and how the MTL represents both space and memory especially during physical navigation, given challenges associated with deep brain recordings in humans during movement. We recorded intracranial electroencephalographic (iEEG) activity while participants completed an ambulatory spatial memory task within an immersive virtual reality environment. MTL theta activity was modulated by successful memory retrieval or spatial positions within the environment, depending on dynamically changing behavioral goals. Altogether, these results demonstrate how human MTL oscillations can represent both memory and space in a temporally flexible manner during freely moving navigation.

The ability to learn and recall personal experiences, or episodic memories, is critical for everyday life and guiding of future behaviors. Encoding of the environmental (spatial) context in which an episode takes place is important for its successful subsequent recall. The medial temporal lobe (MTL) has long been identified as a brain region essential for successful episodic memory formation within a spatiotemporal context across rodents, non-human primates, and humans alike[1–4]. Current evidence from rodent studies suggests that oscillatory activity in the theta frequency band (~6–12 Hz)[5] in the MTL supports spatial navigation[6,7] and successful memory function[2,8] through its ability to temporally organize neural activity locally and across brain regions[2,8]. However, studies in humans show mixed results[9,10] regarding the presence of theta activity and its temporal dynamics during retrieval and encoding of subsequently recalled items[11–14]. Specifically, a majority of human memory studies identify that lower frequency theta (~3 Hz) activity increases/decreases during encoding/retrieval, thereby also calling into question the role of higher-frequency theta oscillations, analogous to those found in rodents, in human memory[9,10].

Given the difficulty of recording human deep brain activity during physical movement, it is currently unknown if and how MTL theta

[1]Department of Bioengineering, University of California, Los Angeles, Los Angeles, CA 90095, USA. [2]Medical Scientist Training Program, University of California, Los Angeles, Los Angeles, CA 90095, USA. [3]Department of Psychiatry and Biobehavioral Sciences, Jane and Terry Semel Institute for Neuroscience and Human Behavior, University of California, Los Angeles, Los Angeles, CA 90024, USA. [4]Department of Psychology, University of California, Los Angeles, Los Angeles, CA 90095, USA. [5]Department of Neurology, David Geffen School of Medicine, University of California, Los Angeles, Los Angeles, CA 90095, USA. [6]Neurosurgery Service, Department of Veterans Affairs Greater Los Angeles Healthcare System, Los Angeles, CA 90073, USA. [7]Department of Neurosurgery, David Geffen School of Medicine, University of California, Los Angeles, Los Angeles, CA 90095, USA. [8]Faculty of Medicine, Tel-Aviv University, Tel-Aviv 69978, Israel. ✉e-mail: nanthia@ucla.edu

oscillations flexibly support memory during ambulatory spatial navigation and/or during complex experiences that involve dynamically changing cognitive demands. Human neuroimaging studies of spatial memory during navigation have traditionally used view-based virtual reality (VR) to simulate movement through an environment while participants remained immobile and restricted due to large recording equipment that is susceptible to motion artifacts. Recent technological advancements in human mobile neuroimaging[15], however, have enabled the discovery of higher-frequency (~7 Hz) MTL theta oscillations that are modulated by physical movement (e.g., walking)[16–18] and proximity to environmental boundaries[17]. Nonetheless, it remains unclear if and how these theta oscillations support successful memory retrieval during ambulatory spatial navigation, and further, how to reconcile their role in flexibly representing both memory and space during a complex behavioral experience.

The current study capitalized on a recently developed mobile neuroimaging platform[15] that enables wireless recording of intracranial electroencephalographic (iEEG) activity from the MTL during unrestricted ambulatory movement in humans. Freely moving participants performed a spatial memory task in immersive VR environments while movement was simultaneously tracked to examine how memory-related processes and spatial features within the environment dynamically modulated MTL activity. Our results suggest that MTL theta

activity reflects both successful memory retrieval and spatial environmental features in a temporally dynamic and flexible manner that can remap based on environmental context and momentary task goals.

## Results

### Measuring spatial memory using ambulatory VR and motion tracking

We developed an ambulatory VR spatial memory task which six participants completed while MTL iEEG activity was recorded (Fig. 1a) from a chronically implanted responsive neurostimulator (RNS) system (Fig. 1b, see detailed information in Supplementary Table 1). The spatial memory task was carried out in an immersive room-scale VR environment (5.84 × 5.84 m, Fig. 1c–h) during which participants interactively navigated to, learned, and later recalled the position of uniquely colored visible translucent cylinders (halos). The physical movement of participants in the real room was mapped to body position in VR space such that the scene was updated according to each participants' motion in a one-to-one-manner. The spatial memory task consisted of learning (encoding) trials, visually guided navigation ("arrow") trials, and memory recall (retrieval) trials (Fig. 1c–f). During encoding trials, participants were instructed to navigate to a halo (Fig. 1c, Supplementary Movie 1) and learn its spatial location, which was fixed over the course of the task. During arrow (search) trials,

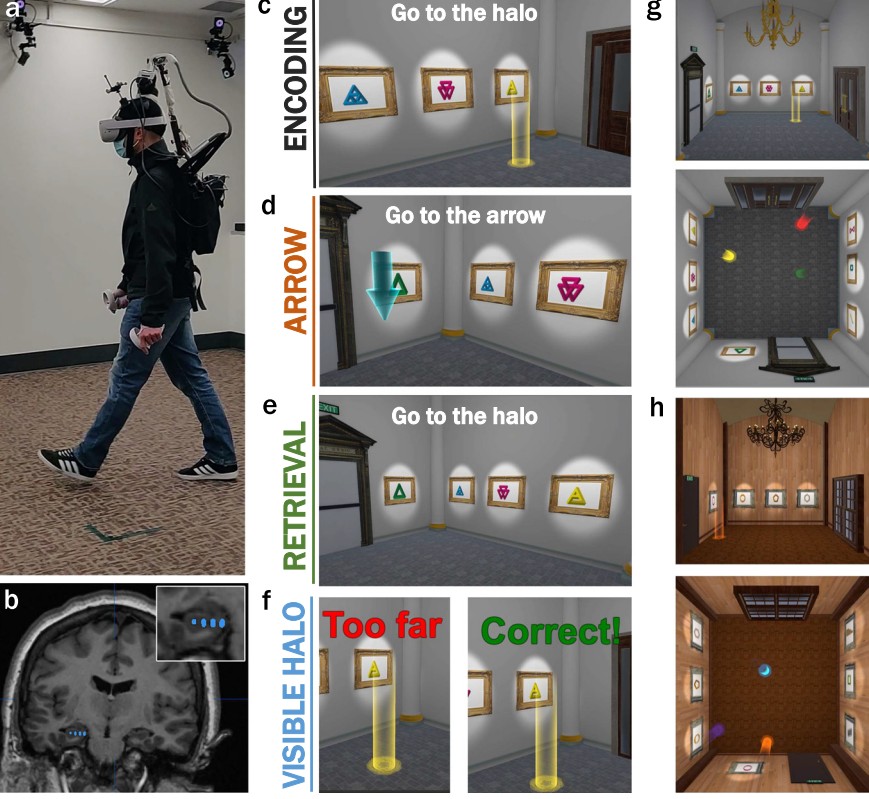

Fig. 1 | Experimental setup and ambulatory spatial memory task. a Equipment worn by participants including the Mobile-Deep Brain Recording and Stimulation (Mo-DBRS) system that enables recording of intracranial MTL activity[15], a virtual reality (VR) headset and associated handheld VR controllers. Also shown are wall-mounted motion capture cameras. Shown is an experimenter wearing the equipment for illustrative purposes (consent to publish obtained). b An example participant's intracranial electrode contacts (blue circles) localized to the left hippocampus from a post-implant CT registered to a pre-surgical MRI. c–f Ambulatory spatial memory VR task showing different types of trials: c Encoding trials, which consisted of navigation to multiple distinct visible halos, each presented one at a time (instructions: "Go to the [color] halo"), d Arrow trials, which consisted of navigation to a visible arrow randomly positioned along the perimeter of the room

(instructions: "Go to the arrow"), e Retrieval trials, which consisted of navigation to previously learned halo positions (halos not visible; instructions: "Go to the [color] halo location & press the button when you arrive") followed by f visible feedback and instructions to navigate to the correct halo position (halo is visible; feedback/instruction: "Too far away!/Correct! Walk to the [color] halo"). g, h Each participant completed the task in two different environmental contexts: g stone and h wooden. Perspective and top-down views of each context with halos visible. Contexts differed in terms of visual appearance (e.g., color/shape of wall artwork, doors, chandeliers, flooring, walls, etc.) but were matched in their geometric layout and placement of visual (wall) artwork. The three halo colors and positions were different between contexts.

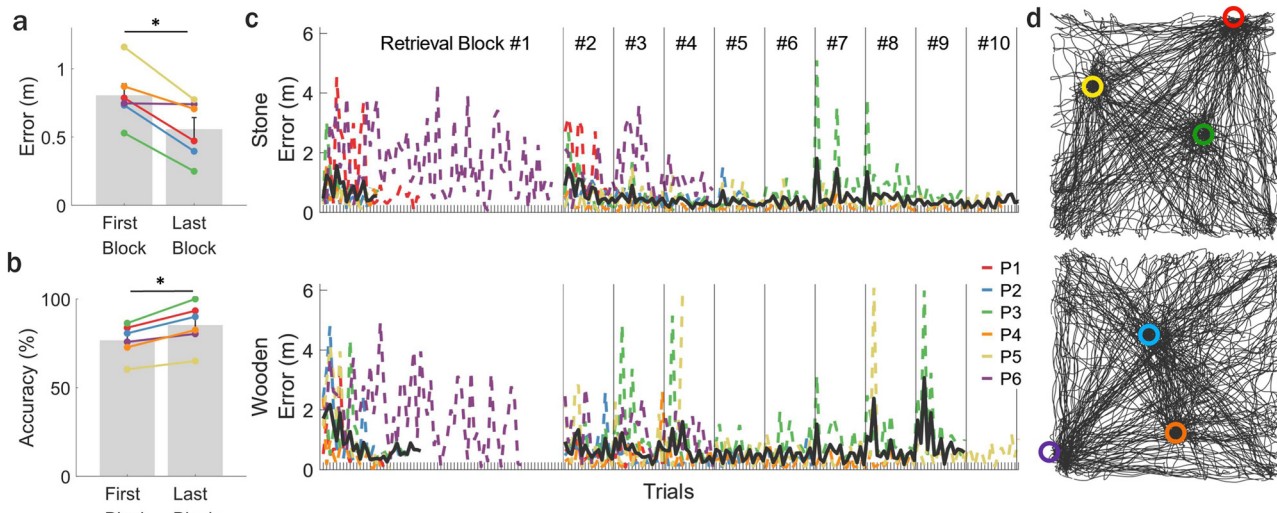

**Fig. 2 | Memory performance during the ambulatory spatial navigation task.**
Difference in **a** mean memory performance (error) and **b** accuracy (% correct) between the first (trials before learning criteria was met) and last retrieval block (**a** $p = 0.021$; **b** $p = 0.036$). Lines show data from individual participants ($n_{participants} = 6$). Error bars indicate s.e.m. Two-sided pairwise permutation tests, * = $p < 0.05$. **c** Mean distance error was measured for each of the six participants (P1–6, colored lines, $n_{halos} = 3$ halos) by calculating the average distance between recalled and correct halo locations across trials. The 1st retrieval block included a variable number of trials for P1–5 (15–30 trials) depending on when a learning criterion

(error for 15 consecutive trials <1.5 m) was reached. P6 did not show learning during retrieval block #1 and thus was manually advanced to subsequent retrieval blocks. Mean performance across participants is also shown (black line, $n_{participants} = 5$ participants, P1–5; P6 excluded due to inability to meet learning criterion). The total number of retrieval blocks varied across participants (5–10 blocks). **d** Top-down view of an example participant's walking trajectory collapsed over all encoding, retrieval, and arrow trials in the stone (top) and wooden (bottom) contexts. Halo colors and positions are indicated in each of the two environments. Source data are provided as a Source Data file.

participants were instructed to navigate to an arrow (Fig. 1d) located in the perimeter of the room, which appeared at a new randomized position in each trial. The task began with encoding trials (each repeated with unique halo colors and positions, Fig. 1g, h) interleaved with arrow trials. After one encoding and arrow trial was completed for each halo, in a one-by-one and sequential manner, participants began retrieval trials, during which they were instructed to navigate to a previously learned halo position from memory and indicate their arrival using a button press on a wireless handheld VR controller (Fig. 1e). After each retrieval trial, visual feedback ("correct" or "incorrect") appeared specifying whether the participant responded correctly or incorrectly. At the end of this feedback and regardless of performance, the halo became visible (visible halos, Fig. 1f) in its correct position until the participant navigated to its center, providing an opportunity to re-learn the halo position. Arrow trials were also interleaved in between retrieval trials similar to encoding trials. See Supplementary Movie 2 for example retrieval, feedback, and arrow trials. Participants completed the task with 15 retrieval and arrow trials (constituting one retrieval block) and alternated between two environmental contexts (stone room: Fig. 1g, wooden room: Fig. 1h) each of which contained three halos with unique colors and positions that differed between the two contexts and were fixed over the duration of the task. For further details see "Methods".

Memory performance during retrieval trials was measured by computing the distance (error) between the recalled position (button press) and the actual halo position (Supplementary Movie 2). After retrieval block #1, mean error across participants was 0.56 m (±0.01, standard error of the mean, s.e.m.) and significantly reduced during the last compared to the first retrieval block (see "Methods" for further details, $p = 0.021$, Fig. 2a). Accuracy (calculated as % correct) was also computed during the same retrieval blocks based on a 0.75 meter (m) radial distance threshold (from the center of the halo), which was used to provide visual feedback to the participant ("correct" or "incorrect"). Mean accuracy was 65% (±8.5% s.e.m.) across participants and improved significantly during the last compared to the first retrieval

block (see "Methods" for further details, $p = 0.036$, Fig. 2b, c). Furthermore, the complete trajectory of an example participant over the course of the entire task in each VR environment is shown in Fig. 2d, illustrating adequate and evenly distributed sampling of positions across the room as was seen in all participants. Altogether, these behavioral findings showcase the ability of ambulatory immersive VR combined with motion tracking to be used to precisely assess spatial memory performance in freely moving human participants with simultaneous iEEG recordings.

## Successful memory retrieval is associated with increased MTL theta band power

We next investigated whether MTL oscillatory activity was modulated by successful memory retrieval. To do this, we examined power across a range of oscillatory frequencies (3–120 Hz) during time periods around the instances of recall. Given the experimental task was predominantly self-paced in nature and participants were freely moving, pinpointing the precise moment of recall required additional consideration. We hypothesized that the temporal windows most likely to contain critical data indicative of memory retrieval would be either prior to button press upon reaching the recollected position and/or subsequent to retrieval cue (trial) onset. Prior to reaching the recollected position (button press), MTL oscillatory power significantly increased only at 6–8 Hz theta frequencies (6–8 Hz: all individual frequencies $p < 0.05$, after correcting for multiple comparisons using the false discovery rate [FDR][19,20], $n_{channels} = 19$, Fig. 3a–d). Specifically, this theta (6–8 Hz) band power was significantly elevated during correct but not incorrect retrieval trials, arrival at visible halos during feedback, or arrival at arrows during arrow trials and this increase occurred during the 0.5 s prior to arrival at the recalled position (Fig. 3i, j; correct vs. incorrect, $p = 0.003$; correct vs. visible halo, $p < 0.001$; correct vs. arrow, $p = 0.047$; incorrect vs. visible halo, $p = 0.190$; arrow vs. incorrect, $p = 0.280$; arrow vs. visible, $p = 0.022$; FDR corrected, $n_{channels} = 19$, Supplementary Movie 3). However, this finding of increased theta power for correct versus incorrect trials was not

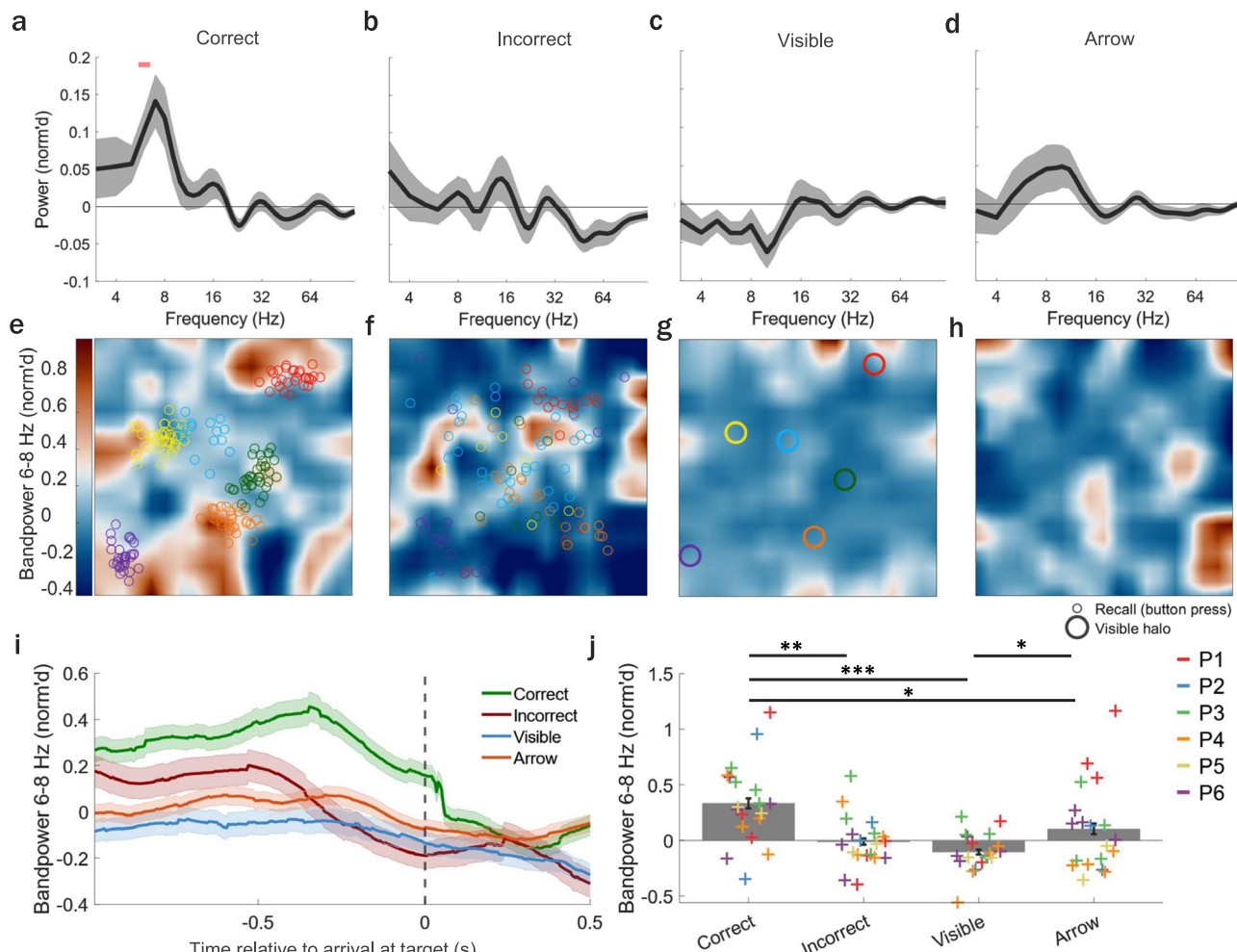

**Fig. 3 | MTL theta band power increased during correct compared to incorrect retrieval, visible halo, or arrow trials. a–d** Mean (±s.e.m.) normalized (norm'd) power across MTL channels ($n_{channels}$ = 19) for frequencies 3–120 Hz during the 0.5 s period prior to either the button press during **a** correct or **b** incorrect recall during retrieval trials, or **c** arrival at visible halos during feedback, or **d** arrival at arrows during arrow trials. MTL theta band power significantly increased during correct but not incorrect retrieval, visible halo, or arrow trials. Horizontal pink bar indicates significant power increase/decrease ($p < 0.05$, corrected using false discovery rate [FDR][19,20]). Top-down view of norm'd theta band power (6–8 Hz) in an example MTL channel averaged across all samples during retrieval **e** correct, **f** incorrect, **g** visible halo, and **h** arrow trials. Note, halos were not visible during correct or incorrect retrieval nor during arrow trials. **e, f** Colored circles reflect all recalled locations during retrieval for correct and incorrect trials. **g** Colored circles reflect locations of halos during visible feedback. **i** Mean (±s.e.m.) norm'd theta band power across

MTL channels ($n_{channels}$ = 19). Vertical gray dotted line (time = 0) indicates the moment (button press) when participants arrived at the remembered halo position (correct/incorrect) during retrieval trials, visible halo (visible, blue) during feedback, or arrow (orange) during arrow trials. **j** Mean (±s.e.m.) norm'd theta band power across MTL channels ($n_{channels}$ = 19) during correct/incorrect trials, visible feedback periods, and arrow trials. MTL theta band power significantly increased during the 0.5 s prior to arrival at the target position for correct compared to incorrect, visible (feedback), and arrow trials (correct vs. incorrect $p = 0.003$, correct vs. visible $p = 0.0004$, incorrect vs. visible $p = 0.19$, correct vs. arrow $p = 0.047$, arrow vs. incorrect $p = 0.28$, arrow vs. visible $p = 0.022$). Crosses (+) represent the mean norm'd band power across all trials for an individual channel with each color corresponding to channels from a single participant. * = $p < 0.05$, ** = $p < 0.01$, *** = $p < 0.0001$, two-sided pairwise permutation test, FDR corrected. Source data are provided as a Source Data file.

dependent on the specific temporal window (0.5 s, Supplementary Fig. 1a), was numerically present across participants (Supplementary Fig. 1b), persisted when averaging over channels for each participant ($p = 0.031$, $n_{participants}$ = 6, Supplementary Fig. 1c), and remained after a leave-one-out approach when each participant's data was excluded one at a time (Supplementary Fig. 1d), suggesting that findings were not driven by individual subjects.

Increases in MTL theta band power prior to arrival at the correctly recalled position only occurred in MTL not non-MTL channels (Supplementary Table 1). Specifically, in non-MTL channels ($n_{channels}$ = 5), there were no significant theta band power changes during successful (6–8 Hz: all individual frequencies $p > 0.05$, FDR corrected) or unsuccessful memory retrieval trials (6–8 Hz: all individual frequencies $p > 0.05$, FDR corrected), or for arrival at visible halos during feedback (6–8 Hz: all individual frequencies $p > 0.05$, FDR corrected). MTL

memory-related theta band power increases could not be explained by the presence of a virtual object since halos were not visible during retrieval trials (Fig. 3a, b, e, f). An example channel illustrating the effect is shown in Fig. 3, where MTL theta band power peaked near correctly (Fig. 3e) but not incorrectly (Fig. 3f) recalled halo positions, visible halo positions (Fig. 3g), or near halo locations during arrow trials (Fig. 3h).

Given prior studies showing that movement speed modulates the prevalence of theta oscillations[16,21,22], we evaluated whether there were differences in speed profiles during correct versus incorrect retrieval trials. We found no significant differences in movement speed during correct compared to incorrect retrieval trials nor between retrieval trials and visible feedback during the same 0.5 s prior to arrival at a visible halo, ($p = 0.115$, $p = 0.845$, $n_{participants}$ = 6, Supplementary Fig. 2), suggesting that the observed memory-related effects were not driven

by differences in movement speed between conditions. Additionally, there were no significant differences in movement speed between navigation in the stone and wooden context nor between the first-encountered and second-encountered context (since starting context was counterbalanced across participants, stone vs. wooden: $p = 0.293$, first vs. second: $p = 0.998$, $n_{participants} = 6$). Furthermore, we quantified the impact of movement speed and correct relative to incorrect memory performance on changes in theta band power during the last 0.5 s prior to reaching the recalled position, evaluated on a trial-by-trial basis using a linear mixed-effects model and found that only correct performance but not movement speed significantly predicted increases in theta band power during this temporal window (correct vs. incorrect, $p = 0.028$; movement speed, $p = 0.337$; $n_{participants} = 6$, Supplementary Fig. 3a).

Next, we examined the simultaneous contribution of multiple behavioral variables (distance to recalled position, distance to boundary, correct vs. incorrect performance, distance error) and movement-related variables (movement speed, angular velocity, movement direction) on fluctuations in theta band power during the entire duration of retrieval trials, up until arrival at the recalled position (when no cues were present; Supplementary Fig. 3b). Specifically, distance to the recalled position (button press) and distance error (distance between button press and target location) were significant predictors of theta band power fluctuations (movement speed, $p = 0.370$; angular velocity, $p = 0.998$; proximity to recalled position, $p = 0.044$; proximity to boundary, $p = 0.741$; correct vs. incorrect, $p = 0.340$; distance error, $p = 0.046$; movement direction = 0.290; $n_{participants} = 6$, Supplementary Fig. 3b).

We also explored successful memory-related theta band power changes during other time periods (Supplementary Fig. 4) and found that while theta band power (6–8 Hz) initially appeared to be similar between correct and incorrect trials after initial cue presentation during retrieval trials, a difference was detected around 1.5 s after cue onset (Supplementary Fig. 4a). Previous work has suggested that, within a broader theta frequency range, low-frequency theta oscillations (e.g. type II theta) are related to episodic memory and higher-frequency theta oscillations (e.g. type I theta) are movement-related[23,24]. As such, we also investigated differences between correct and incorrect trials in low-frequency oscillations (3–6 Hz) after cue presentation, and before the onset of movement (Supplementary Fig. 4c–f). Since participants often had multiple movement onset periods within a single trial, we specifically examined the last movement onset before button press, which we hypothesized would better capture the temporal window when participants initiated memory retrieval to determine their final recalled position for the indicated target halo on any given trial. Indeed, we found that low-frequency theta band power (3–6 Hz) was increased already around 0.5 s after cue onset (Supplementary Fig. 4c) and that this elevation could only be explained by distance from the cue itself and not other movement or behavioral variables (Supplementary Fig. 3c). Furthermore, both low-frequency (3–6 Hz) and high-frequency (6–8 Hz) theta band power was also increased prior to the last movement onset during retrieval trials (Supplementary Fig. 4b,d), another time period likely to capture moments of memory recall.

Given prior results illustrating that MTL theta oscillations occur in non-continuous bouts in freely ambulating humans[16,17], and that these bouts are modulated by behavioral variables (e.g., movement speed), we examined whether differences in the prevalence of theta bouts could explain memory-related effects on MTL theta band power (Supplementary Fig. 5). We found that MTL theta band power increases did occur in transient bouts and occurred at similar rates compared to previous studies[16,17], however, the prevalence of these bouts did not significantly differ between task conditions either during the entire retrieval trial period (retrieval vs. arrow vs. visible halo trials, $p > 0.05$; correct vs. incorrect, $p > 0.05$; across all individual frequencies

between 3–25 Hz, $n_{channels} = 19$, Supplementary Fig. 5a–c) or the last 0.5 s prior to arrival at the recalled position (retrieval vs. arrow vs. visible halo trials, $p < 0.05$; correct vs. incorrect, $p > 0.05$; across all individual frequencies between 3–25 Hz, $n_{channels} = 19$, Supplementary Fig. 5e, f), suggesting successful memory retrieval results in increased MTL theta band power in the absence of changes in its prevalence.

## MTL theta band power is modulated by spatial position

Next, we investigated whether MTL theta oscillations were modulated by one's location in the environment. To do this, we used data from both contexts (stone and wooden) and computed MTL theta band power across positions, separately in each room, during retrieval (when halos were not visible) and arrow (search) trials (when arrival positions at arrows were excluded). We first excluded iEEG data from retrieval periods immediately (0.5 s) preceding the button press. In this way, we could determine whether MTL theta band power was modulated by spatial position, independent of reaching a designated target goal halo position during retrieval. Thus, this analytic approach retained instances when participants incidentally traversed non-visible previously learned (non-target) halo positions along the participants' trajectory to the goal halo location. We examined MTL theta band power when participants were in positions that were classified as "close" to or "far" from the non-visible non-target halos during participants' trajectories to the target halo (of which the 0.5 s prior to target halo arrival was excluded). MTL theta (6–8 Hz) band power was significantly increased at "close" compared to "far" distances relative to the non-visible non-target halo positions. The difference in MTL theta band power between "close" and "far" positions peaked at a distance threshold of 2 m from non-target halo positions (distance thresholds of 1, 1.25, 1.5, and 2 m: all $p < 0.05$, $n_{channels \times conditions} = 38$, Fig. 4a, and $p < 0.05$ at distance thresholds of 1.5 and 2 m after FDR correction; illustration of 2 m threshold: close vs. far, $p = 0.008$, Fig. 4b). Interestingly, we did not observe such a pattern of results during arrow trials (Supplementary Fig. 6a), indicating that proximity to (non-target) halo locations modulated theta power only during memory retrieval but not when participants walked towards a visible cue. The spatial distribution of theta (6–8 Hz) band power increases was specific to relevant positions within each context separately (stone: $p = 0.012$; wooden: $p = 0.041$, $n_{channels} = 19$, Supplementary Fig. 6b), suggesting that MTL spatial representations can remap based on the perceived environment (see example channel showing theta activity in the stone (Fig. 4c) and wooden (Fig. 4d) context). Furthermore, the difference in theta band power between "close" and "far" positions was strongest in later blocks, likely after the participants developed robust spatial representations (Supplementary Fig. 6c). Together, these results suggest that MTL theta band power increased incidentally at meaningful spatial positions within a familiar environmental context.

We next examined how MTL oscillatory power was modulated by spatial positions near room boundaries (e.g., walls), based on evidence of boundary-related representations identified in a prior ambulatory spatial navigation study in humans[17]. Since the VR room dimensions in our study were identical to those in this previous navigation study[17], we used the same boundary-inner room area cutoff of 1.2 m from the wall (although, see Supplementary Fig. 7a for additional cutoffs used). Across widespread (3–120 Hz) oscillatory frequencies examined, mean power significantly increased at boundary compared to inner room positions only for lower theta frequencies (4–6 Hz) during arrow search trials (excluding 0.5 m prior to arrow arrival: 4–6 Hz: all individual frequencies $p < 0.05$, FDR corrected, $n_{channels} = 19$, Fig. 5a; boundary versus inner: $p < 0.001$, Fig. 5b; $n_{channels} = 19$). Conversely, there were no significant differences in mean theta band power between boundary and inner room positions during memory retrieval trials (4–6 Hz: all individual frequencies $p > 0.05$, FDR corrected, $n_{channels} = 19$). Boundary-related power increases were also observed at higher frequencies during the *entire* duration of arrow trials, including

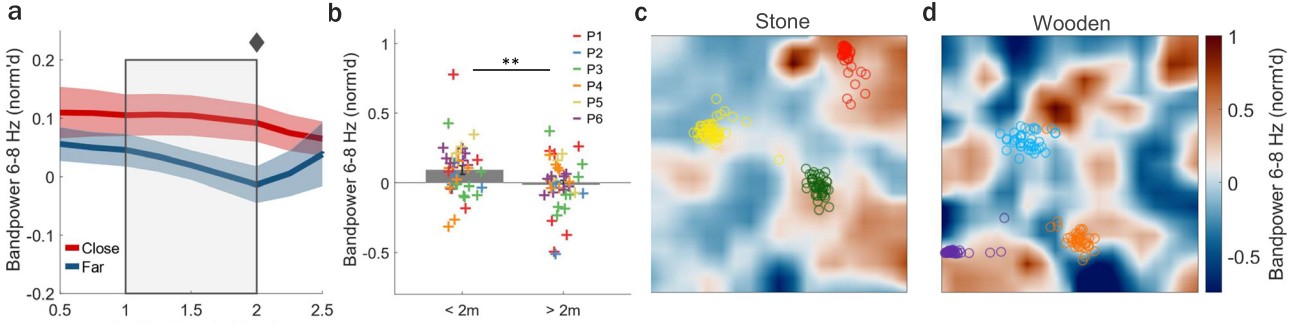

**Fig. 4 | MTL theta band power increased at non-target halo positions. a** Mean normalized (norm'd) theta (6–8 Hz) band power during retrieval trials (excluding last 0.5 s prior to recall of target halos) in positions close versus far from non-target halo positions shown across varying radius (distance to halo) thresholds used to determine the cutoff between "close" and "far" positions. Gray box highlights radius thresholds where theta band power significantly differed between "close" and "far" positions ($p < 0.05$, two-sided pairwise permutation test for each radius threshold; $p < 0.05$ for radius = 1.5 and radius = 2, after correction using false discovery rate [FDR][19,20]). Diamond indicates radius threshold (2 m) visually illustrated in (**b**). Shaded red/blue areas indicate s.e.m. **b** Detailed view of the mean norm'd theta band power across individual recording channels

($n_{channelsxcontexts} = 38$) using 2 m threshold (diamond in **a**), shown for illustrative purposes ($p = 0.045$, two-sided pairwise permutation test, FDR corrected for multiple comparisons shown in **a**). Crosses (+) represent the mean norm'd band power for each channel (colors correspond to individual participants) during retrieval trials, excluding the last 0.5 s prior to recall of a target halo. Error bars indicate s.e.m. **c, d** Top-down view of theta band power in an example channel across room positions, during retrieval trials, when no visible halos were present, and excluding the 0.5 s preceding recall of target halos. Circles represent positions where target halos were recalled (button press), split by stone (**c**) and wooden (**d**) contexts and with colors corresponding to the color of the halo that was recalled. ** = $p < 0.01$. Source data are provided as a Source Data file.

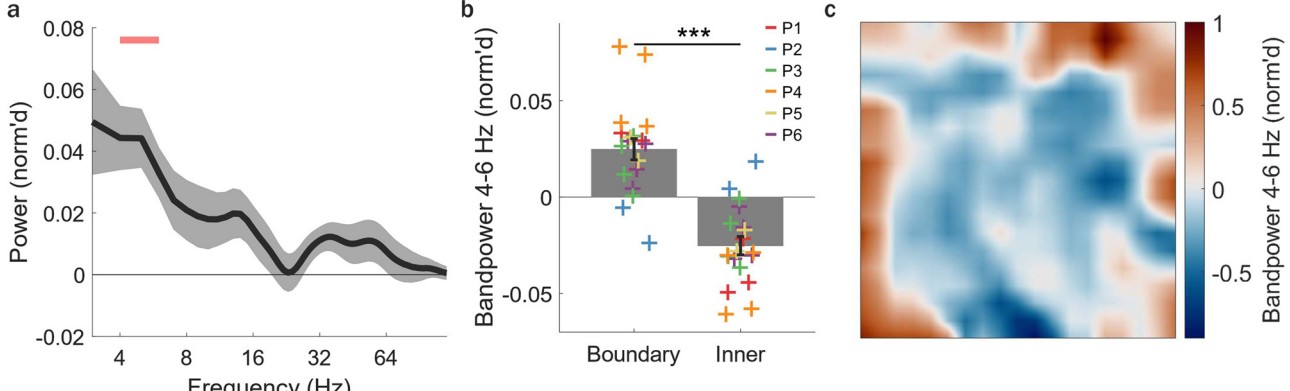

**Fig. 5 | MTL theta band power is modulated by position relative to environmental boundaries. a, b** Analysis performed over arrow trials, excluding the 0.5 m leading up to arrival at arrows. **a** Mean (±s.e.m.) normalized (norm'd) difference in power across frequencies (3–120 Hz) and MTL channels ($n_{channels} = 19$) between positions near (<1.2 m of walls, based on prior work[17]) versus away from boundaries. Significant differences in norm'd power in boundary compared to inner positions occurs for theta frequencies (4–6 Hz, horizontal pink bar = $p < 0.05$, corrected

using false discovery rate [FDR]). **b** Mean ± s.e.m. norm'd theta band power (4–6 Hz) across MTL channels ($n_{channels} = 19$) for boundary and inner positions ($p < 0.0001$, two-sided pairwise permutation test). Crosses (+) represent individual channels with colors corresponding to individual participants. *** = $p < 0.001$. **c** Top-down view of norm'd theta (4–6 Hz) band power in an example channel for data across the entire task, excluding the 0.5 m leading up to arrival at arrows. Source data are provided as a Source Data file.

the last 0.5 m prior to arrival at the arrow, (12–14, 31–35 Hz, all individual frequencies $p < 0.05$, FDR corrected, $n_{channels} = 19$) similar to a previous study[17]. The boundary-related theta band power increase was also present when looking at data over the *entire* task (again, excluding data from positions within 0.5 m of arrow arrival, 4–6 Hz band power, $p < 0.001$) and can be seen in an example channel in Fig. 5c. It is unlikely that differences in movement speed were driving boundary modulation of theta band power since speed was significantly elevated in inner relative to boundary positions, in direct opposition to the increased theta band power in boundary positions, given prior work in humans and non-human animals showing higher theta band power associated with faster movement speeds[16,18,25] ($p = 0.048$, $n_{participants} = 6$, Supplementary Fig. 2f). Further, after accounting for movement variables (speed, angular velocity, movement direction) in addition to behavioral variables in the previously described linear mixed-effects model approach, we found that movement speed was not a significant predictor of theta band power fluctuations (movement speed, $p = 0.966$;

angular velocity, $p = 0.617$; movement direction, $p = 0.507$; $n_{participants} = 6$, Supplementary Fig. 3d). Moreover, while proximity to boundary was not a significant predictor of theta band power in the previously described linear mixed-effects model during memory retrieval (Supplementary Fig. 3b), we found that during arrow search periods only distance to (nearest) boundary was a significant predictor of elevated theta band power, whereas proximity to the visible arrow cue (distance to arrow) were not (proximity to arrow, $p = 0.267$; proximity to boundary, $p = 0.048$; $n_{participants} = 6$, Supplementary Fig. 3d), suggesting that there is a linear relationship between boundary proximity and theta band power. Together, these results suggest that boundary-related theta increases are not driven by movement speed, movement-related variables, nor visible cues (arrows) and that theta is dynamically modulated by boundary proximity based on ongoing task demands.

Also, the prevalence of theta bouts was not significantly different between "boundary" and "inner" positions ($p > 0.05$, across all

individual frequencies 3–25 Hz, $n_{channels} = 19$, Supplementary Fig. 5d) similar to a previous study[17]. To examine whether encoding of visual information on walls was contributing to boundary modulation of theta power, we examined theta band power fluctuations in two separate conditions: when participants were moving towards the (nearest) wall and when participants were moving away from the (nearest) wall. We observed that boundary modulation of theta band power persisted in both conditions (towards: $p < 0.001$, away: $p < 0.001$, $n_{channels} = 19$, Supplementary Fig. 7d). Notably, boundary-related modulation of MTL theta band power was not present during retrieval trials, with or without including the 0.5 s of data preceding arrival at the recalled location (all retrieval trials: $p = 0.175$; excluding 0.5 s preceding recall: $p = 0.202$ boundary versus inner, 4–6 Hz band power, $n_{channels} = 19$), potentially due to competing modulation of theta activity by non-visible non-target halo positions during retrieval search periods as discussed previously. Also, recall-related theta increases during correct trials persisted when excluding data from boundary positions, suggesting that theta band power differences between correct and incorrect retrieval trials were not driven by boundary-related theta effects (correct vs. incorrect, $p = 0.033$, correct vs. visible, $p = 0.003$, incorrect vs. visible, $p = 0.401$, $n_{channels} = 19$, Supplementary Fig. 1e). Similarly, non-target modulation of theta power further persisted when examined only in the boundary region of the environment (Supplementary Fig. 6d, e), suggesting that this effect was also not driven by boundary modulation of theta band power. Lastly, boundary-related increases in theta power were not dependent on the specific 1.2 m boundary vs. inner cutoff (Supplementary Fig. 7a) and occurred within individual participants (Supplementary Fig. 7b), when averaged over individual channels of participants ($p = 0.049$, $n_{participants} = 6$, Supplementary Fig. 7c), and persisted during a leave-one-out approach when each participant's data was excluded one at a time (Supplementary Fig. 7e), suggesting that findings were consistent across participants and not driven by individual subjects. Taken together, these results demonstrate that MTL theta band power can be dynamically modulated by critical positions (e.g., that previously contained relevant objects or proximity to walls) depending on environmental context or task goal.

## Discussion

We have shown that human MTL theta band power is modulated dynamically by successful memory retrieval and spatial position depending on environmental context and momentary behavioral demands. While previous studies have used simultaneous ambulatory iEEG recordings and immersive VR[15], this is the first to collect empirical data to investigate human spatial memory. In this way, we were able to investigate how MTL oscillations represent memory and space flexibly during an ambulatory spatial navigation task that involves changes in context and behavioral demands. Our findings highlight two phenomena, one related to memory recall and the other to spatial position.

First, we find that MTL theta band power is elevated during memory recall. This increase is particularly pronounced around 0.5 s after cue onset and 0.5 s before reaching the retrieved location, when the recalled item (halo) is not visible, and only when it is recalled correctly. This pattern of results echoes previous findings in stationary humans, showing hippocampal reinstatement of low-frequency theta oscillations during early retrieval time windows (specifically within the first 0.5 s after a retrieval cue was presented)[26], stronger representational similarity of iEEG activity in the 1 s prior to recall during remembered relative to forgotten trials[27], and increased theta power during spatial memory retrieval in view-based navigation tasks[28–31]. Additionally, our finding that theta band power was elevated only during successfully recalled trials is in line with prior reports from human iEEG studies that identify low-frequency theta activity being modulated by memory performance during stationary view-based

spatial memory tasks[28–30,32]. It is possible that the higher-frequency theta effects seen prior to reaching the retrieved location are due to the fact that participants were physically navigating. In line with this hypothesis, we found elevated low-frequency theta band power (e.g. memory-related type II theta) in two time windows associated with less movement: around (1) 0.5 s after cue presentation and (2) 0.5 s prior to movement onset, while there was elevated higher-frequency theta band power (e.g. movement-related type I theta) in two time windows associated with more movement: (1) around 1.5 s after cue presentation and (2) in the 0.5 s prior to arrival at the recalled position[23,24]. Moreover, we also found that a continuous metric of memory performance (distance error) was linearly related to changes in theta band power over the entire duration of retrieval trials, suggesting that memory retrieval success modulated theta power fluctuations throughout the retrieval period. Importantly, we found no significant differences in speed profiles during correct versus incorrect memory retrieval trials or task conditions (memory retrieval, or arrival at arrows or visible cues) suggesting MTL high-frequency memory-related theta band power changes were not driven by changes in movement speed, nor was there any significant contribution of movement-related variables to theta band power fluctuations during retrieval trials. Further, while prior work in ambulatory humans has shown that theta prevalence (not power) is modulated by movement speed during a non-mnemonic walking task[16] our findings here suggest successful memory retrieval modulates theta band power in the absence of changes in its prevalence. Thus, our results emphasize the importance of investigating memory and spatial representations during ambulation, while highlighting the need for future studies to determine how high versus low-frequency memory-related theta changes differ between ambulatory compared to stationary (virtual) navigation[33].

Second, we found that MTL theta band power increased near previously learned object (halo) positions or environmental boundaries depending on context and momentary task goals. Specifically, MTL theta band power increased near non-target halos when participants were actively searching for and recalling a separate non-visible target halo. This neural representation of relevant halo positions was specific to each context (stone or wooden) and alternated as participants switched between environments, but did not persist during the interleaved arrow trials (visual search period that lacked a memory demand), consistent with the idea of context reinstatement during memory retrieval and in a manner relating to the trial objective[30]. Further, MTL theta band power increased at positions close to environmental boundaries (walls) but only when searching for boundary-positioned cues (arrows). Importantly, boundary modulation of theta band power persisted in conditions both when participants approached and moved away from the wall, suggesting that the visual information available when facing a wall was not driving this spatial representation. Additional analyses using a linear mixed-effects model approach further highlighted the dynamic nature of theta oscillations in that proximity to boundaries (and not proximity to visual cues) predicted theta band power fluctuations during arrow trials, but not during retrieval trials. However, although proximity to the (nearest) boundary but not proximity to the visual cue (arrow) was linearly related to theta band power, we cannot fully rule out the possibility that visible cues (arrows) contributed in some way to boundary modulation of theta band power. This spatial remapping of oscillatory activity based on the behavioral goal (memory recall versus cue-driven navigation) suggests that MTL theta band power can dynamically reflect multiple spatial and mnemonic variables in an on/off and flexible manner, where the presence of specific neural representations depends on the immediate cognitive requirements of the task at hand. For instance, boundary-related increases in theta power could be present exclusively during periods when the proximity to the spatial boundary is relevant (as seen in arrow trials, where arrows are positioned at room boundaries). On the other hand, during memory

retrieval phases when spatial boundaries are momentarily less relevant, these boundary-related representations might be notably absent. Similarly, it is possible that transient theta power increases may reflect relevant neural representations that are momentarily engaged during dynamic mind-wandering states in humans. Specifically, a momentary increase in power of theta bouts may reflect the relevant neural representations (e.g. for memory or space) that are recruited for a particular cognitive/behavioral goal, in contrast with rodents, where more continuous theta activity occurs during freely moving navigation. Thus, these findings provide a possible explanation for non-continuous theta bouts in humans[16] where behavioral/cognitive variables may play a more critical role in their modulation as compared to continuous movement-related theta oscillations in rodents[6,34]. Future studies will be needed to better understand the exact role of low- (type II) versus high- (type I) frequency theta oscillations in this context.

Mechanistically, remapping of MTL theta band power across different cognitive tasks could reflect coordinated remapping of local single-neuron activity, although the relationship between oscillatory and single-neuron remapping in humans requires further exploration. Rodent studies have shown that place cells, which encode particular positions in an environment, globally remap in different contexts and environments[35,36]. It is thus plausible that nearby local theta band power remapping may reflect or organize place cell remapping, or that changes in theta band power may reflect the summation of populations of nearby remapped place cells. Prior studies recording MTL single-neuron activity in stationary humans showed firing rate changes that dynamically changed during free recall tasks[37] when virtually approaching the position of a previously learned object[38], and in relation to egocentric directions while navigating towards local reference points[39]. Our results demonstrated a linear relationship between theta band power fluctuations and proximity to the recalled position (button press) during retrieval trials. Recently reported object vector trace cells[40] may represent a population of hippocampal neurons that could contribute to this effect by modulating firing rate patterns to create a vector field pointing to a previously encountered object's position. Our results also highlight that theta power is modulated by non-visible and non-target locations of previously learned halos in a latent manner. Consistent with this finding, object trace cells in the entorhinal cortex selectively fire in the location of previously encountered object positions, even at long time periods after the object has been removed from the environment[41]. Finally, we found robust boundary representations elicited in our task consistent with the existence of border cells and boundary-vector cells[42,43], which increase their firing rate when animals are near borders of an environment. Given that these MTL neuronal populations each exhibit characteristic tuning to memory and spatial features, it is possible that their summative activity may be coordinated in relation to broader regional theta oscillations to support successful memory retrieval and anchoring of the positions of spatial targets. Indeed, environmental (contextual) remapping of population-level neuronal signals identified with fMRI has also been shown in a stationary view-based virtual navigation study in humans where hippocampal-entorhinal cortex activity "flickered" between two contexts during incorrect memory retrieval trials as the participant struggled to identify the environment they were in[44].

Traditional human neuroimaging studies of memory retrieval and spatial navigation have been carried out in stationary participants viewing stimuli on a computer screen. Many of these studies were also designed to evaluate neural activity changes during brief stimulus presentations (e.g., 1–2 s when a cue is presented), which limits the ability to disentangle more complex neural dynamics related to multidimensional spatiotemporal experiences in an immersive environmental setting. In contrast, by utilizing 3D ambulatory VR, our study presents a critical advancement for future human behavioral studies measuring brain activity during freely moving behavior by creating a more ecologically valid setting that enables participants to physically explore, learn, and recall experimentally controlled stimuli in their environment. Furthermore, the use of VR in this way still allows for deliberate experimental control of the environmental context, as well as the timing and placement of stimuli.

In summary, our results provide insights into how human MTL oscillatory dynamics support cognitive representations that could dynamically reflect both memory and space in an ecologically valid setting that involves physical movement through distinct spatial environments. These findings suggest that MTL theta oscillations contain memory- and spatial contextual-related information that may enable transient changes in cognitive states during complex real-world experiences. Our combined deep brain recording and immersive VR approach also presents a unique opportunity for future cognitive and clinical neurosciences studies of naturalistic behavior in humans to unravel underlying mechanisms during complex freely moving behaviors that may be further impaired in patients with neurologic and psychiatric disorders.

## Methods
### Participants
There were 6 participants in the study (33–54 years of age, 4 female, mean = $43.3 \pm$ s.e.m. = 3.1). All the participants had pharmacoresistant epilepsy treated with a chronically implanted FDA-approved RNS System (Neuropace, Inc; 320 Model) that continuously records iEEG activity across 8 contacts (4 bipolar channels). Participants with at least 2 bipolar channels in MTL regions (e.g., hippocampus or entorhinal cortex) were recruited for the study (example electrode placement shown in Fig. 1b). The sites of electrode implants were determined by clinical criteria. Further, participants with low seizure activity and thus fewer average daily stimulation therapies were recruited for the study. Informed consent approved by the UCLA Medical Institutional Review Board (IRB) was obtained from all participants. Participants received financial compensation for taking part in this study. The participants' sex and gender was determined based on self-report. Due to the limited sample size, the participants' sex and gender was not considered in the study design.

### Spatial memory task in immersive virtual reality
Participants completed an ambulatory spatial memory task in two different immersive VR environments (room dimensions were $5.84 \times 5.84$ m) where they learned and retrieved various positions of translucent colored cylinders (halos) as discussed in the main text. All VR environments were matched in size to the real-world environment and constructed using the Unity game engine. VR headsets used included the Quest 2 VR headset (Meta, Inc., as seen in Fig. 1a) or the Pico Neo 1 and Pico Neo 2 VR headsets (Pico Immersive Pte. Ltd). Prior to performing the task, participants completed a 5-min practice version of the task in a distinct virtual environment to provide them familiarity with the immersive VR headset and to engage them in normal walking behavior. The first retrieval block included several repeated sets of retrieval and visible halos, until the participant met a learning criterion (completing 15 consecutive trials with error <1.5 m). Retrieval block #1 was completed in an identical manner in the second context, immediately following completion in the first context. The starting context (stone or wooden) was counterbalanced across participants. The total number of trials in retrieval block #1 varied across participants (15–30 trials in participants 1–5, Fig. 2c, see details in Supplementary Table 1). P6 was unable to learn all halos to meet the learning criterion in retrieval block #1 in both contexts, and as such, was manually advanced to retrieval block #2 after 40 min in each context (69 trials in the stone context, 60 trials in the wooden context). For retrieval block #2 and above there were a fixed number of trials (15 in each block), with the context per block alternating until a total

of 2–11 retrieval blocks were completed depending on the time available for each participant (see number of block details by participant in Supplementary Table 1). Total task time took approximately 30–200 min across participants. "Earlier" and "later" blocks were defined using a median split across all blocks that each participant completed.

Location and orientation tracking of participants was collected throughout the experiment with submillimeter resolution through the Opitrack motion tracking system using twenty-two high-resolution infrared wall-mounted cameras and MOTIVE application (Natural Point, Inc., see Fig. 1a). The cameras sampled the position of a collection of uniquely oriented rigid body position markers located atop the participants head at 120 Hz (Fig. 1a). Positional data was compared across VR headsets and Opitrack data collection, and analysis proceeded using VR headset data since positional accuracy was comparable. Movement speed was computed as the change in position between consecutive samples divided by the time lapse between samples. Angular velocity was computed as the change in yaw rotational dimension (radians) of consecutive samples divided by the time lapse between samples.

## iEEG data acquisition

The RNS System continuously records iEEG activity and delivers stimulation in a closed-loop fashion upon detection of abnormal (i.e., epileptic) activity patterns to prevent imminent seizure activity, and is implanted in the skull to support two penetrating electrode leads, 1.27 mm in diameter, with up 4 platinum-iridium electrode contacts spaced either 3.5 mm or 10 mm apart. In each participant, 4 bipolar channels were recorded at a sampling rate of 250 Hz. In accordance with the IRB protocol and with participant consent, closed-loop stimulation was turned off during the experimental recordings in order to remove potential stimulation artifacts from the data.

For the duration of the experiment, amplifier settings on the RNS System (320 model) were programmed to apply a 1 Hz high pass filter and a 90 Hz low pass filter (see Supplementary Fig. 8 for filter response). Wireless iEEG data was recorded from the RNS System as previously described[15]. Briefly, a "Wand" accessory wirelessly recorded iEEG from the implanted RNS System using near field telemetry. The Wand was positioned on the head, immediately above the implanted RNS System on the patients' head and secured in a custom-made Wand holder and attached to a backpack to allow for free movement (Fig. 1a). Data was stored as a continuous timeseries across channels and storage was remotely triggered wirelessly at the end of each session of continuous blocks. Of note, since there was no wired connection between the implanted RNS System (the recording apparatus), the VR headset, and an external power source, the iEEG data was free from power line noise.

To synchronize iEEG with behavioral data, the Unity application executed on the VR headset was programmed to trigger a signal (mark) wirelessly at specific time points inserted into the iEEG data. These synchronization marks were sent at specified times in the tasks, specifically at the start of each block (see Topalovic et al.[15] for synchronization details of the setup).

## Electrode localization

Precise localization of electrode contacts was performed by co-registering post-operative head CT scans with pre-operative MRI scans (T1 and/or T2-weighted sequences). One example localization of the four contacts on one electrode lead can be seen in Fig. 1b. Across the six participants, there were nineteen total channels localized to the MTL in regions including the hippocampus, parahippocampal cortex, perirhinal cortex, and entorhinal cortex. No recording contacts were located in the amygdala. For list of electrode contact localizations of all participants, see Supplementary Table 1.

## Detection of epileptic events

Inter-epileptic discharges (IEDs) are abnormal electrical distortions related to epilepsy that can occur intermittently and on an individual basis. IEDs were removed from all iEEG channels prior to normalization of power and all additional neurophysiological analyses. We applied IED detection methods previously described[16,17,45]. Briefly, IED detection used a double thresholding approach where for the first threshold, each sample was tested against two criteria to identify IEDs to be removed from analysis: (1) whether the envelope of the unfiltered signal was 6.5 standard deviations away from baseline, and (2) whether the envelope of filtered signals (15–80 Hz band pass filtered after signal rectification) was 6.5 standard deviations away from baseline activity. Once these IED samples were detected, a second threshold was applied to remove samples surrounding detected IED samples. Specifically, a smoothing gaussian filter with a moving kernel range of 0.1 s was applied to a binary vector with 1's denoting detected IEDs and a threshold of 0.01 was applied to the smoothed vector to identify all samples around and including detected IEDs, all of which were excluded from analysis in order to remove potential residual epileptic activity. In order to remove a wider window around high-amplitude IED events, this method was applied a second time with a higher 7.25 standard deviation cutoff for the first threshold and a wider 0.25 s smoothing window for the second threshold. Using this method, 2–10% of samples were removed per channel (Supplementary Table 1), similar to previous results[16,17]. We specifically recruited participants with low baseline IED activity based on their historical data from the RNS System (i.e., average daily number of stimulation events delivered in recent months).

## Behavioral analyses

Memory performance was computed as the distance error, or the distance between the position at which the participant pressed the button during retrieval trials to indicate the recalled halo position and the center position of the halo. Immediately after the button press, the participants received visual on-screen feedback of either "Correct!" (if they were within 0.75 m of the halo's center) or "Incorrect". To determine whether participants successfully learned halo positions over each experimental session, learning was evaluated by comparing each participants' mean error (e.g., memory performance) in retrieval block #1 across both contexts (excluding the last 15 trials which met the learning criterion threshold necessary to advance past retrieval block #1 for P1–5) to mean error during their last retrieval block in each context. The mean error performance across the last retrieval block compared to that during retrieval block #1 (before meeting the learning criterion) was evaluated for significance using a pairwise permutation test across participants.

For memory retrieval analyses, correct and incorrect trials were defined from when the participant received instructions to retrieve a particular halo (no visible halo cue was present) until the instance at which they recalled the halo position (button press). Visible halo (feedback) periods were defined from the instance of recall (button press), at which point the halo appeared in its correct location, until the instance at which the participant navigated to the visible halo. Visible halos occurred immediately following both correct and incorrect trials; feedback appeared after correct trials even when participants were within 0.75 m of the halo and thus participants were still required to navigate to the center of the visible halo.

For boundary versus inner room area analyses, we used a method similar to a previous study[17]. Since the same room dimensions used in this study were identical to those used in a previous study, the same 1.2 m proximity to boundary (i.e., wall) cutoff was used to separate "boundary" versus "inner" room areas (but see Supplementary Fig. 7a for alternative boundary-distance thresholds tested).

Movement onset time points were defined as the moment when movement speed changed from "no movement" (speeds of less than

0.2 m/s) to "movement" (speeds of 0.2 m/s or greater) and remained above 0.2 m/s for at least 1 s.

To distinguish movements towards versus away from boundaries, we first measured the distance to the nearest boundary for each sampling point, and then calculated whether this distance decreased (categorized as "towards boundary") or increased (categorized as "away from boundary") between two adjacent sampling points.

### iEEG data analysis

Time frequency analysis was performed by computing the oscillatory power at individual frequency steps of 1 Hz between 3 and 120 Hz using the BOSC toolbox[46,47] with a three cycle Morlet wavelet. We also repeated all analyses with a six cycle Morlet wavelet given previous approaches[16,17] and the results were qualitatively the same (i.e., none of the results in the manuscript text changed from "significant" to "non-significant" or vice-versa, and all patterns of data were virtually identical between the two analysis approaches), suggesting the robustness of the results with regards to this analysis parameter. Next, each channel's power timeseries was normalized for each frequency step using the MATLAB "zscore" function (after excluding IED samples). This normalization procedure was performed for each recording channel separately, and involved initially computing the mean and standard deviation across the complete timeseries for each recording session. Subsequently, each individual data sample within the entirety of the recording session duration (with the exception of IED samples) was subjected to z-score transformation using the computed mean and standard deviation values. Recorded timeseries that were separated by longer breaks (more than ~40 min; e.g. before/after a participant's lunch break) were treated as independent recording sessions and normalized separately.

For bar graphs comparing mean power across a band power range (i.e. 6–8 Hz), normalized power was summed over frequency steps (1 Hz) for all samples that fell within a particular task condition of interest (e.g., any sample that occurred during any correct or incorrect trial). Mean normalized power was then computed over the summed band power timeseries.

To evaluate the prevalence of significant theta oscillations, we used the BOSC toolbox to detect bouts of at least 2 cycles above 95% chance for 1 Hz frequency steps between 3–25 Hz as has been done previously[16,17]. Theta prevalence was computed as the percentage of detected bouts out of all relevant task condition samples.

### Linear mixed effect model analysis

Linear mixed effect models were calculated using each participant's normalized low (3–6 Hz) or high (6–8 Hz) frequency theta band power timeseries as the response variable with multiple movement- and task-related predictor variables. Movement-related predictor variables included speed and angular velocity as continuous variables, as well as movement direction as a categorical variable. The movement direction variable consisted of twelve possible rotational bins and test statistics were averaged over bins, because movement direction as a numeric variable (due to its circular nature) is not expected to have a linear relationship with theta band power (e.g., 0 and 2 pi radians are not expected to evoke substantially different theta band power fluctuations). Models were constructed to describe theta band power fluctuations across four specific time periods during the task: (1) the final 0.5 s of retrieval trials prior to the button press, (2) the initial 1 s of retrieval trials following cue onset, (3) the complete duration of retrieval trials, and (4) arrow trials. In addition to movement-related variables, these models incorporated several task-related behavioral variables as follows: For model (1) and (3), distance to the recalled position when a button press was made ("distance to recalled position", measured in meters), distance to the closest boundary ("distance to boundary", measured in meters), distance between the button press location and halo position ("distance error", measured in meters) all as

continuous variables, and correct versus incorrect trial performance ("correct/incorrect", treated as a categorical variable) were included. For model (2) all the variables present in (1) and (3) were included, while also introducing an additional variable: the distance from the cue onset position ("distance from cue", measured in meters). For model (4) the added behavioral variables included were: "distance to boundary" and distance to the trial-specific arrow ("distance to error", measured in meters), both as continuous variables. All predictor variables were defined as fixed effects. To account for individual channel-based variation, the recording channel was used as the grouping variable for random effects. The impact of each predictor variable on theta band power was evaluated by statistically comparing beta weights.

### Data subsampling

For analyses comparing oscillatory power or theta prevalence between conditions that had a differing number of samples, we performed all calculations on 500 iteratively generated, equally sized subsets of data. Specifically, we first compared the number of samples for all conditions to be compared (e.g., correct, incorrect, and visible halo conditions). For the condition with the fewest number of samples, we applied no correction. For the other conditions, we randomly selected the same number of samples for the fewest-sample condition from the longer timeseries and repeated this step iteratively 500 times, with replacement (using the MATLAB "datasample" function). For each iteration, we computed the parameter of interest (e.g., band power), then averaged this parameter of interest across all 500 iterations. We did this on a channel-by-channel basis and used the averaged result for all statistical comparisons and plotting of data.

### Statistical comparisons

Statistical comparisons between two conditions were performed using a paired-sample permutation test as follows: To compare two paired arrays of values (e.g., each of the recording channels' average band power during "correct" versus "incorrect" trials, or in the "boundary" versus the "inner" room area), the paired-sample permutation test calculates whether the mean difference between paired values is significantly different from zero. It estimates the sampling distribution of the mean difference under the null hypothesis, which assumes that the mean difference between the two conditions (correct vs. incorrect, or boundary vs. inner) is zero, by shuffling the condition assignments and recalculating the mean difference many times ($n_{perm}$ = 1000). The observed mean difference between conditions was then compared to this null distribution as a test of significance. The key steps of this procedure are described in more detail below.

Step 1: The observed difference between conditions is calculated, by first calculating the difference between conditions for each value pair (condition 1 value – condition 2 value), and then calculating the average difference across pairs.

Step 2: Condition labels are randomly shuffled within each value pair and the difference between "shuffled conditions" is calculated, by first calculating the difference between randomly labeled conditions for each value pair (value randomly labeled with condition 1–value randomly labeled with condition 2), and then calculating the average difference across pairs. This step is repeated $n_{perm}$ times (in $n_{perm}$ permutations), to generate a distribution of $n_{perm}$ "random differences" between conditions.

Step 3: The observed difference between conditions (calculated in step 1) is compared with the distribution of random differences from shuffling condition labels (calculated in step 2). The p-value is calculated by the number of random differences that are larger than the observed difference, divided by the total number of samples in the distribution. Two-sided permutations tests were used unless otherwise noted.

For multiple comparisons correction (e.g., when performing statistical tests for multiple frequency steps in a band power analysis, or across multiple conditions), p values were adjusted using the false discovery rate (FDR)[18,19]. For top-down maps of theta band power (e.g., Figs. 3–5), the room was divided into 19 × 19 bins. Mean band power over condition was computed for each bin, specifically the band power for all samples in which the participant was positioned in a bin was summed, then divided by the number of samples the participant occupied in that bin. A gaussian smoothing kernel of 0.2 standard deviations was applied to this heatmap, normalized to the peak power and finally, interpolated (using MATLAB function "interp2" with $k = 7$) for visualization.

Statistical tests for main effects were conducted separately across channels and across participants (averaged across each participant's individual channels). Significant effects were observed in both analyses, except for the following instances where significance was only observed across channels but not across participants: Fig. 4a, Supplementary Figs. 6b and 1e.

**Analysis software**
Data were analyzed using MATLAB 2020a (The MathWorks, Natick, MA).

**Reporting summary**
Further information on research design is available in the Nature Portfolio Reporting Summary linked to this article.

## Data availability
The data that support the findings of this study are available from the corresponding authors upon request. Source data are provided with this paper.

## Code availability
The custom computer code used to generate results that are reported in the paper are available from the corresponding authors upon request.

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

## Acknowledgements
This work was supported by the National Institutes of Health (NIH), the National Institute of Neurological Disorders and Stroke (NINDS), and the National Institute of Mental Health (NIMH), under award numbers U01NS103802 to N.S., U01NS117838 to N.S., K99NS126715 to M.S., F30MH125534 to S.L.L.M, by the McKnight Foundation (Technological Innovations Award in Neuroscience to N.S.) and a Keck Junior Faculty Award (to N.S.). We thank all participants for taking part in the study, and all members of the Suthana laboratory for discussions.

## Author contributions
Conceptualization: S.L.M., M.S., N.S.; Methodology: S.L.M., M.S., U.T., Z.M.A., B.K., N.S.; Software: S.L.M., M.S., U.T., D.B., Z.M.A.; Analysis: S.L.M., M.S.; Investigation: S.L.M., M.S., U.T., S.H., J.S., J.P.L., I.F., D.E., N.S.; Resources: J.S., J.P.L., I.F., D.E., N.S.; Data curation: S.L.M., M.S., U.T.; Writing-original draft preparation: S.L.M., M.S., N.S.; Writing-review and editing: S.L.M., M.S., U.T., D.B., S.H., Z.M.A., B.K., J.S., J.P.L., I.F., D.E., N.S.; Visualization: SLM; Supervision: M.S., J.S., J.P.L., I.F., D.E., N.S.; Project administration: S.L.M., S.H., J.S., J.P.L., I.F., D.E., N.S.; Funding acquisition: S.L.M., M.S., N.S.

## Competing interests
The authors declare no competing interests.
