## [Peer Review File · Nature Communications]

Dynamic neural representations of memory and space in freely-moving humansReviewers' Comments:

Reviewer #1:

Remarks to the Author:

The article by and colleagues describes preliminary experience using an entirely novel method to assay spatial navigation in humans, reflecting many years of innovative work to develop the capability to acquire data from the NeuroPace RNS system. This builds on previously published work linking activity in the MTL to theta oscillatory power during proximity to boundary events. The authors are to be commended for developing and establishing the RNS systems as a useful tool for understanding human memory function. The authors incorporate key insights in spatial navigation drawn from previous experimentation and their analysis reflects an excellent understanding of issues related to human spatial navigation. These findings will spark interest among investigators interested in spatial memory and MTL function.

In the paradigm, individuals are shown the location of a virtual color cylinder at different locations (encoding), which I believe is four different color cylinders at different locations around the room. Encoding is the first trial for each color, when the location is learned. This is followed by a variable number of retrieval events in which the individual is prompted to return to the location at which the color cylinder first appeared. Brain signal is recorded the whole time, mostly from the MTL/hippocampus.

The principal result is a difference in 6-8 Hz theta power for successful vs unsuccessful navigation events, where success is defined as a set distance from the actual cylinder location. The secondary result, which was previously reported by this group, is that there is elevated oscillatory power in this same frequency range when individuals move within the boundary areas as compared to the non-boundary areas.

The main question I have is whether the 0.5 sec immediately preceding the button press can really be considered a memory retrieval condition. The task structure seems to imply that retrieval begins when the individual is cued regarding which color cylinder she or he is supposed to go find. Then, the individual makes a decision about where to go. The period right before the button press may say something more about reward/feedback expectation than representation of navigational goals. This potential confound is maybe exacerbated by the fact that with repeated attempts at the same target, the comparison mixes early, low confidence versus later, high confidence trials. Some additional control analyses may help account for this. First, does the theta relationship hold for correct vs incorrect trials at the time of cue? The time/power plot shown suggests that theta power is elevated relative to baseline earlier in time, when the individuals may be making spatial memory judgments more actively than when navigation has ceased and they are about to press the button. Eye tracking data from the VR set up (or like head direction position) may solve this problem also. The BOSC analysis suggests that there are similar numbers of theta episodes during both correct and incorrect conditions, which seems at odds with the inference regarding memory-related theta power increases. Also, clearly the 0.5 sec threshold is fairly arbitrary. The authors do show that the power differences seem to persist earlier in time, but the same power comparisons should be made using different time windows to show how sensitive the main results are to this cutoff.

The boundary related theta power changes raise another confound, since boundary condition is associated with theta power changes. There may be more incorrect trials from non-boundary adjacent target cylinders. The authors should show that theta power differences persist for correct/incorrect comparisons for non-boundary events.

The environment is pretty small, which I understand reflects some practical limits on engineering/space etc. Do the theta power differences persist when individuals are looking at (but physically near) the boundary areas? The confound I am worried about is related to the amount of visual information available at the boundaries, with more pictures/complexity present. I don't know

exactly what the best control analysis would be for this but it would help nail down the findings as related to boundaries specifically rather than something related. Similar to above, the boundary cutoff is somewhat arbitrary. Is there a relationship between linear boundary proximity and theta power to support the result using the cutoff value?

Another concern is related to movement speed. The control analysis shown in extended data includes a ($df=5$) comparison of average speed for the different conditions, which does not reach significance, but there are some differences across individuals and the means. The authors should perform a standard multivariate analysis to show that corr/error differences persist when including movement speed as a factor in the analysis. This same concern applies to whether individuals are stationary or non-stationary at the time of .5 sec window prior to button press.

Regarding the statistical methods, I believe the main result is based on 1) calculating a power difference across the mean power for the corr/error trials 2) shuffling the corr/error trial labels, recalculating the power difference, and comparing the true values to the distribution of shuffled power differences. But I don't totally understand when the p value is extracted from the true/null distribution comparison (I'm sorry if I missed it). Some additional clarification is required. I don't think the shuffle occurs at the level of electrode, because that would only have 19 values to shuffle. The authors should also report the statistical result (in terms of the test statistic used) for the non-bootstrapped test as well as the shuffle test, which would make it easier to interpret the results with df reported.

Finally, there are only 6 subjects, which I think is ok given the difficulty of obtaining these really interesting data. This precludes a random effects analysis. But, the authors should add some confidence to the findings by showing that the main results related to corr/error and boundary hold when iteratively throwing out one subject at a time. This will not affect the df (less than 19 contacts) too much but would help to show that the results are not driven by strong effects in one individual.

I would like to reiterate that this is a novel dataset, difficult to acquire, that will be of interest to a wide range of investigators. I think it has great potential for impact.

Reviewer #2:

Remarks to the Author:

The authors examine intracranial EEG recordings from the medial temporal lobe (MTL) of freely ambulating people with epilepsy completing a spatial memory task in immersive virtual reality. They demonstrate that accurate spatial memory retrieval is associated with increased theta power (in the 6-8Hz band) immediately preceding a button press response; that increased theta power (in the same frequency band) is observed when participants pass close to (<2m) invisible, previously learned non-target goal locations; and that increased theta power (in the 4-6Hz band) is observed when participants are located close to the boundaries of the environment in visually guided navigation ('arrow') trials.

While these data are difficult to collect and rely on a relatively new (and exciting) invasive recording technology, the findings described here are not particularly novel (aside from the increases in theta power associated with proximity to non-target goal locations, which is not extensively investigated) and come from a relatively small sample (of $n=6$ patients). As such, it is hard to see how these results provide any new insight into the role of MTL theta in human memory function. Nonetheless, they add to a growing literature that utilises an ethological approach to study human memory function and demonstrate that some previous findings from immobile, desktop VR paradigms (i.e. increased MTL theta power during accurate memory retrieval, described in references 24-27 and others reviewed in reference 11) can be translated to more realistic contexts, which may be of interest to the community.

Major Concerns

All statistical analyses consider each electrode contact to be an independent sample ($n=19$ channels), despite the fact that many pairs of channels are located millimetres apart in the same brain tissue. This is not an appropriate use of statistics, although I appreciate that (sadly) it is fairly standard in this field of research. Ideally, the authors should repeat their analyses with data averaged across channels in each patient as $n=6$ independent samples. At the very least, the authors should include some indication of whether these effects are present in each patient (and / or hemisphere), and colour code the scatter points on Figures 3H, 4B, and 5B to indicate which data come from which patient

On a related note, the statistical analyses of power spectra (i.e. as illustrated in Fig 3A-C and 5A) are not described in sufficient detail. As far as I can tell, the authors examined 118 different frequency bands (from 3 to 120Hz in steps of 1Hz) and then used FDR correction to find significant increases or decreases in power between conditions, but it is not clear if they applied any cluster correction, or if the power increases illustrated in Fig 3A and 5A just happen to be restricted to adjacent 3Hz bands? Moreover, it is not clear – in the memory retrieval analysis – how the authors compared the 0.5s period prior to recall with other time windows (or which other time windows were examined); or what is meant by the statement: “memory-retrieval-related increases in MTL theta bandpower were not present over the entire retrieval period”. Please clarify

Finally, the analyses illustrated in Figure 4A and B would appear to be the most egregious example of double dipping – the authors first look for a radius that dissociates theta power close to and far from the non-target halo positions, and then show that theta power close to and far from those positions (defined using this radius) is significantly different. It is hard to see how this approach can be justified. It would also be useful to know what proportion of the total environment and what proportion of the boundary region falls into each ('close to' or 'far from') condition

It is not clear to me why the authors interpret increased theta power in the proximity of learned goal locations as driven by memory representations of those locations, but increased theta power in the proximity of arrow locations as driven by the environmental boundary, rather than the prominent visual arrow cue that just happens to be located near to the boundary. Can they provide any additional analyses to convince us that theta power increases in arrow trials are genuinely driven by the boundary, and not the visual cue? For example, if they compare theta power close to (i.e. within 2m) and further from (i.e. >2m from) the visible arrow, within the 1.2m boundary region, do they see any power differences? Or if they examine theta power within the 1.2m boundary region, excluding locations around the visual arrow cues, does the effect persist?

Finally, in terms of motivation and interpretation, there are various theoretical models (including some cited here) that describe why we might expect to see increased theta power during memory retrieval, but it is not clear at all why we might expect increased theta power near the boundaries of an environment. What hypothesis were the authors testing with these analyses? What mechanism might account for these findings? What do they tell us about human MTL memory function?

Minor Concerns

Given that the authors repeatedly allude to the distinction between low and high theta oscillations that has previously been made in the literature, it would be useful for them to state explicitly whether the effects they report are truly specific to the high theta band (i.e. 6-8Hz), or whether the 1Hz high-pass hardware filter simply prevents them from examining low theta. Would it be possible to show the frequency response of this filter as a supplementary figure, to get an idea of how much signal attenuation it introduces across low frequencies?

It would be useful to add to Table 1 some indication of how much data from each patient was excluded due to IEDs

In the introduction, the authors suggest that rodent hippocampal theta oscillations fall in the 6-8Hz band, but it is usually more like 6-12Hz (6-8Hz is just the frequency band used here) – please edit accordingly. On a similar note, the authors suggest that rodent hippocampal theta is implicated in memory retrieval, but a more representative summary of the literature would be that it is implicated in memory encoding (retrieval is just the focus of this study), so perhaps ‘memory function’ would be more accurate?

Where the authors cite their own study of iEEG recordings during ‘real world’ movement, they might also cite the study from Bohbot et al. published around the same time in Nature Communications

In the first paragraph of the results, where the authors first describe the structure of the task, they refer to ‘arrow trials’, before these have been introduced or described. This is a little confusing for the reader – perhaps they could describe them as ‘visually guided navigation (‘arrow’) trials’ or similar, for the sake of clarity

In the results, the authors state that “...these behavioral findings showcase the ability of ambulatory immersive VR combined with motion tracking to be used to precisely assess spatial memory performance in freely moving human participants with simultaneous iEEG recordings” – but this implies that this is the first demonstration of that technology, which is misleading. Perhaps they could add “consistent with previous studies”, or similar, and cite the 2020 Neuron paper, 2021 Nature paper, and / or recent Nature Neuroscience paper from the same lab that describe this technology in more detail

Similarly, in the discussion, they state that “this study is the first to our knowledge to combine simultaneous ambulatory iEEG recordings and immersive VR”, but that would appear to be patently false, given the list of previous publications above. Please clarify

In the discussion, “have been done in” is a little informal – perhaps “have been carried out in” would be preferable?

In the methods, I believe ‘sixth order Morlet wavelet’ should be ‘six cycle Morlet wavelet’

Reviewer #3:

Remarks to the Author:

In this manuscript, Maoz and colleagues leverage a unique opportunity to record intracranial EEG from the human medial temporal lobe (MTL) as subjects are freely ambulating. This allows the authors to investigate an important outstanding question in the field, which is the role of theta oscillations in spatial navigation and memory. There has been substantial evidence that such oscillations are prominent in rodents as they navigate spatial environments, but the evidence in humans is mixed. This is largely because most intracranial recordings are captured from subjects who are confined to their hospital bed, and navigation in these cases is limited to simple virtual navigation on a laptop. Here the authors use an RNS system for capturing data in an ambulatory setting, a method they have previously published on and demonstrated, to ask what these oscillations in the MTL are doing during a spatial navigation and memory task. In brief, they find that theta oscillations become stronger as individuals approach locations of targets that they have retrieved from memory. In addition, they also find evidence of stronger theta oscillations as individuals approach the boundaries of the environment. The main frequency band identified is in the higher theta range, which is consistent with theta observed in freely moving animals and somewhat distinct from theta that has been described in humans in immobile settings. Together, these data provide valuable insights into the potential role of theta oscillations in the human MTL and would be an important addition to the literature. The experiment is well conducted and the analyses are clear and appropriate. Overall, this is a good manuscript, but there are some suggestions that could either strengthen the conclusions of the work

or provide additional novel insights.

The authors report improved learning over blocks, which would suggest that individuals learn the overall task structure. But if the halos are randomized between blocks, then one should also look at learning within a block. Does this also improve over the trials in a block. Presumably, this is the role of the feedback that is provided. In addition, it would be helpful to also know about improvements in accuracy (as determined by whether the button press was correct or incorrect), to complement the changes in mean spatial error.

The analyses are all conducted by averaging the data across 19 electrodes from all participants. Clearly, in every patient there are only one or two contacts, which has motivated this approach. But it would be helpful to know what these effects look like at the subject level. Could it be that only 1 or 2 subjects are driving this overall effect, or is this consistent across subjects. This will likely be similar to the original results that are presented, since there are only a small number of electrodes per subject, but it would be helpful nonetheless to see the analyses averaged across participants rather than electrodes.

The comparison is made between correct trials to the period of visible feedback. But it seems like there is a natural and perfect build in control in the experimental design, which is the arrow trials. How does theta power look when approaching the arrow trials as compared to the approach during correct retrieval trials.

The analysis of spatial position suggests that there is an increase in MTL theta activity near meaningful positions (e.g., non-targets). But is there also an overall spatial modulation of theta. From the maps provided, it appears that the bouts of theta may be somewhat structured and regular as one simply navigates around the environment. Is this the case? Again, the arrow trials could be an opportunity to examine this.

The spatial map appears to change between contexts. But if this map is determined by largely by non-targets, then does the map also change within contexts but between blocks, when the location of the learned halos has changed?

The boundary analysis is performed during arrow trials and shows elevated theta near boundaries. Clearly, these trials involve moving to the arrows that are located at the boundary. So is this a boundary effect (and similarly a memory effect for inner regions), or is the increase in theta simply related to the goal of the trial. Interestingly, the analysis excludes the .5 s before arriving at the arrow. And this effect is also seen when analyzing the entire task. Could boundary activity be related to the pictures on walls, which could serve as memory cues?

In the discussion, they state that MTL theta activity is only elevated right before retrieval during successful recall, but this is also true for incorrect trials (compared to feedback at least).

It would also be interesting to see what happens when the subjects are first given the target cue and still have not moved yet. Are there also increases in theta during this immobile period before the make a movement, and are these increases now in low theta. This might suggest that the bands can be dissociated depending on whether movement is involved.

The results suggest that there is no changes in theta prevalence. In addition, there is higher prevalence for other higher frequencies. What does the analysis of theta prevalence reveal during only the .5 s retrieval time period?

It is not clear how much time elapses between encoding the halos and the retrieval period. Is there a distraction period or a distraction task?

Dear Reviewers,

We would like to thank you for your generous and constructive feedback on our submitted manuscript, “Dynamic neural representations of memory and space in freely moving humans”. Each reviewer made suggestions that would greatly strengthen the manuscript. We have now incorporated each of these suggestions into a revised manuscript. We hope that you will find the following updates responsive to your feedback.

Reviewer Comments:

Reviewer #1 [R1]: *The article by [Maoz] and colleagues describes preliminary experience using an entirely novel method to assay spatial navigation in humans, reflecting many years of innovative work to develop the capability to acquire data from the NeuroPace RNS system. This builds on previously published work linking activity in the MTL to theta oscillatory power during proximity to boundary events. The authors are to be commended for developing and establishing the RNS systems as a useful tool for understanding human memory function. The authors incorporate key insights in spatial navigation drawn from previous experimentation and their analysis reflects an excellent understanding of issues related to human spatial navigation. These findings will spark interest among investigators interested in spatial memory and MTL function.*

Response: We thank the reviewer for their time and effort to review our manuscript, and for the positive feedback. Please find our responses to each additional comment below.

Comment 1 [C1]: *In the paradigm, individuals are shown the location of a virtual color cylinder at different locations (encoding), which I believe is four different color cylinders at different locations around the room. Encoding is the first trial for each color, when the location is learned. This is followed by a variable number of retrieval events in which the individual is prompted to return to the location at which the color cylinder first appeared. Brain signal is recorded the whole time, mostly from the MTL/hippocampus. The principal result is a difference in 6-8 Hz theta power for successful vs unsuccessful navigation events, where success is defined as a set distance from the actual cylinder location. The secondary result, which was previously reported by this group, is that there is elevated oscillatory power in this same frequency range when individuals move within the boundary areas as compared to the non—boundary areas. The main question I have is whether the 0.5 sec immediately preceding the button press can really be considered a memory retrieval condition. The task structure seems to imply that retrieval begins when the individual is cued regarding which color cylinder she or he is supposed to go find. Then, the individual makes a decision about where to go. The period right before the button press may say something more about reward/feedback expectation than representation of navigational goals. This potential confound is maybe exacerbated by the fact that with repeated attempts at the same target, the comparison mixes early, low confidence versus later, high confidence trials. Some additional control analyses may help account for this. First, does the theta relationship hold for correct vs incorrect trials at the time of cue? The time/power plot shown suggests that theta power is elevated relative to baseline earlier in time, when the individuals may be making spatial memory judgments more actively than when navigation has ceased and they are about to press the button. Eye tracking data from the VR set up (or like movement direction position) may solve this problem also. The BOSC analysis suggests that there are similar numbers of theta episodes during both correct and incorrect conditions, which seems at odds with the inference regarding memory—related theta power increases. Also, clearly the 0.5 sec threshold is fairly arbitrary. The authors do show that the power differences seem to persist earlier in time, but the same power comparisons should be made using different time windows to show how sensitive the main results are to this cutoff.*

Response: We thank the reviewer for raising these important points. We have included several additional analyses in response:

First, as suggested by the reviewer, we quantified the theta bandpower increase for correct versus incorrect recall times at the time of the cue. As shown in Extended Data Fig. 4a, after cue presentation, theta power

initially appears to be similar between correct and incorrect trials, but a significant difference can be detected around 1.5 seconds after cue onset.

Please note, however, that our experimental task was largely self-paced and participants were freely moving, and thus we cannot fully determine at what exact point in time participants retrieved the memory of the actual target location. In fact, participants could freely decide at what point in time after cue onset they would start to move, and some participants may have had multiple movement onset periods (for instance, a participant may have started to walk immediately after cue onset, then stop again, think about the correct target location and perhaps re-consider their initial choice, before walking to the actual recalled position). As a result, we expect the time window immediately after cue onset to contain substantial 'noise' (i.e., variability between participants and trials), as this time window likely contains data from some trials when participants retrieved the correct memory and other trials when they retrieved the correct memory at a later point in time. Thus, we hypothesized that another interesting period most likely captures a time window critical for memory retrieval: the last movement onset before button press (i.e., the time when participants likely made their final decision as to where to go next and where to press the button). Interestingly, another reviewer raised a closely related point, and asked whether differences in theta power for correct vs. incorrect trials are present after cue onset and during immobility (i.e., before movement onset) of participants, and whether these effects might be present in the low-theta frequency range (see Reviewer 3, C7 for more details).

In response, we performed an additional analysis to test whether low-frequency (around 3-6 Hz) and high-frequency (around 6-8 Hz) theta oscillations exhibit differences in power between correct and incorrect trials after cue presentation and before movement onset. Indeed, in line with reviewer #3's hypothesis, we found that low-frequency theta power was increased already around 0.5 after cue onset, as well as around movement onset (Extended Data Fig. 4).

Together, these results suggest that, regarding differences in theta power between correct and incorrect trials, the earliest effects directly after cue onset and before movement onset are present in the low-frequency theta band (around 3-6 Hz), while later effects around 1.5 sec after cue onset and before button press appear to be more prevalent in the higher theta range (around 6-8 Hz).

Extended Data Figure 4: Successful memory modulation of theta bandpower after cue and movement onset. Mean (\pm s.e.m.) norm'd (normalized) theta bandpower across MTL channels ($n_{\text{channels}} = 19$) (a,c) after cue presentation and (b,d) around movement onset (last movement onset prior to button press in each retrieval trial) in (a-b) 6-8 Hz and (c-d) 3-6 Hz bandpower. Note, halos were

not visible during correct (green) or incorrect (red) retrieval trials. Gray bar indicates timepoints where $p < 0.05$ (one-sided permutation test at each time point, representing 4 ms steps at 250 Hz sampling rate).

We have added a discussion of these new findings to the manuscript (pg. 8, lines 185 -199):

“We also explored successful memory-related theta bandpower changes during other time periods (Extended Data Fig. 4) and found that while theta bandpower (6-8 Hz) initially appeared to be similar between correct and incorrect trials after initial cue presentation during retrieval trials, a difference was detected around 1.5 seconds after cue onset (Extended Data Fig. 4a). Previous work has suggested that, within a broader theta frequency range, low frequency theta oscillations (e.g. type II theta) are related to episodic memory and higher frequency theta oscillations (e.g. type I theta) are movement-related^{1,2}. As such, we also investigated differences between correct and incorrect trials in low frequency oscillations (3-6 Hz) after cue presentation, and before the onset of movement. Since participants often had multiple movement onset periods within a single trial, we specifically examined the last movement onset before button press, which we hypothesized would better capture the time window when participants initiated memory retrieval to determine their final recalled position for the indicated target halo on any given trial. Indeed, we found that low-frequency theta power was increased already around 0.5 s after cue onset (Extended Data Fig. 4c) and prior to movement onset during retrieval trials (Extended Data Fig. 4d).”

and (pg. 13, lines 330-335):

“In line with this hypothesis, we found elevated low-frequency theta bandpower (e.g. memory-related type II theta) in two time windows associated with less movement: around (1) 0.5 s after cue presentation and (2) 0.5 s prior to movement onset, while there was elevated higher-frequency theta bandpower (e.g. movement-related type I theta) in two time windows associated with more movement: (1) around 1.5 s after cue presentation and (2) in the 0.5 s prior to recall^{2,23}.”

We also agree with the reviewer that the time window of 0.5 s before button press reflects a somewhat arbitrary time window, and thus it is important to show that our presented results do not depend on this particular choice. Consequently, we analyzed theta power differences between correct and incorrect trials for varying time windows (0.25 ms to 1 s) prior to button press and find that the successful memory effect is sustained across all of these time windows (Extended Data Fig. 1a).

Extended Data Figure 1a: Supplementary analyses for successful memory modulation of theta. MTL theta bandpower in the 0.5 s prior to recall (button press) for correct and incorrect trials across various time windows within 1 s (0.25 – 1 s, $n_{\text{channels}} = 19$). Crosses (+) represent the mean norm'd (normalized) bandpower across all trials for individual where separate colors correspond to individual participants. * = $p < 0.05$, ** = $p < 0.01$, *** = $p < 0.001$.

Further, in response to this and other reviewer comments, we performed an additional analysis that sheds insight on the reviewer's question. Specifically, we used a linear mixed-effects model (a type of general linear model), which allows for investigation of the relationship between theta power and a variety of variables that might simultaneously impact theta oscillations, while taking into account repeated measures from different recording channels with different amounts of data per participant. With this approach, we examined how movement-related variables (speed, angular velocity, movement direction) and behavioral variables (distance

to recall, distance to boundary, correct/incorrect trials, distance error) contributed to theta bandpower fluctuations during the memory retrieval and arrow search periods, respectively.

With this approach, we demonstrated that whether the participants' responses were correct or incorrect (correct/incorrect) during retrieval trials was associated with significant increases in theta bandpower in the last 0.5 seconds prior to recall, even after accounting for movement speed (Extended Data Fig. 3a). We also incorporated movement-related and behavioral variables into another model over the entire retrieval trial time period (up until the instant of recall). In this retrieval trial model (Extended Data Fig. 3b), proximity to recall (distance to button press or instance of recall) and distance error (distance between button press and target location) were significant predictors of theta bandpower fluctuations. The proximity to recall results suggest that the memory process (e.g., proximity to the point of subjective recall) was a driver of theta bandpower fluctuations throughout the retrieval time period, independent of the selection of a particular time window (e.g., 0.5 seconds) before button press. Since the distance error was a continuous variable (as opposed to a categorical 'correct' vs. 'incorrect' distinction), these results suggest that behavioral performance was linearly related to theta bandpower fluctuations.

Additionally, the observation of a strong neural signal relating to successfully recalled information in the short time window preceding recall is in line with other recent studies. Liu et al (2020)³ found that stronger representational similarity analysis occurred in the 1 second time window preceding recall during remembered relative to forgotten trials in an iEEG study. Lifanov et al (2022)⁴ reports peak reactivation at 1.2 s prior to subjective recollection in an EEG-fMRI study. We have added a sentence to the manuscript to highlight these relevant findings (pg. 13 lines 324-325):

"..., and stronger representational similarity analysis of iEEG activity in the 1 s prior to recall during remembered relative to forgotten trials³."

Extended Data Figure 3a-b: Simultaneous impact of multiple variables on theta power. Linear mixed-effects models were calculated to predict each participant's normalized theta bandpower timeseries by a range of predictor variables that were fixed effects with samples blocked according to channel identity. Asterisks denote a significant impact of a variable's beta weight on theta power, * = $p < 0.05$ for $n_{\text{participants}} = 6$. Error bars show the s.e.m. of the beta weights across participants. Models were implemented for (a) the last 0.5 s of the retrieval period (when no halos were visible), and (b) the entire retrieval period (when no halos were visible). Movement speed, angular velocity, proximity to recall (distance to button press), proximity to boundary (distance to nearest wall), and error (distance between the position of recall and target location) were continuous variables. Whether a trial was correct or incorrect (correct/incorrect) was a binary variable and movement direction was a categorical variable comprised of 12 possible binned movement directions (with the mean beta coefficient over all 12 directional bins depicted). Crosses (+) represent the variable impact (beta) colored according to participant.

We have added these results to our manuscript (pgs. 7-8, lines 169 - 184):

"Furthermore, we quantified the impact of movement speed and correct relative to incorrect memory performance on changes in theta bandpower during the last 0.5 s prior to recall on a trial-by-trial basis using a linear mixed-effects model and found that only correct performance but not movement speed significantly predicted increases in theta bandpower during this time window (correct vs. incorrect, $p = 0.028$; movement speed, $p = 0.337$; $n_{\text{participants}} = 6$, Extended Data Fig. 3a).

Next, we examined the simultaneous contribution of multiple behavioral variables (distance to recall, distance to boundary, correct vs. incorrect performance, distance error) and movement-related variables (movement speed, angular velocity, movement direction) on fluctuations in theta bandpower during the entire duration of retrieval trials, until the instant of recall (when no cues were present; Extended Data Fig. 3b). Specifically, proximity to recall (distance to button press or instant of recall) and distance error (distance between button press and target location) were significant predictors of theta bandpower fluctuations (movement speed, $p = 0.37$; angular velocity, $p = 0.998$; proximity to recall, $p = 0.044$; proximity to boundary, $p = 0.741$; correct vs. incorrect, $p = 0.340$; distance error, $p = 0.046$; movement direction = 0.290; $n_{\text{participants}} = 6$, Extended Data Fig. 3b).”

and (pgs. 13-14, lines 335 - 338):

“Moreover, we also found that a continuous metric of memory performance (distance error) was linearly related to changes in theta bandpower over the entire duration of retrieval trials, suggesting that memory retrieval success modulated theta power fluctuations throughout the retrieval period.”

C2: The boundary related theta power changes raise another confound, since boundary condition is associated with theta power changes. There may be more incorrect trials from non—boundary adjacent target cylinders. The authors should show that theta power differences persist for correct/incorrect comparisons for non-boundary events.

Response: The reviewer makes a great suggestion. In order to rule out this possible confound, we compared theta power differences for correct and incorrect retrieval trials and found that our effects persist for non-boundary (inner) environment positions, suggesting that increased theta power for correct versus incorrect trials is not driven by proximity to boundary. We have added these results to our manuscript (pg. 12, lines 294-297):

“Also, recall-related theta increases during correct trials persisted when excluding data from boundary positions, suggesting that theta bandpower differences between correct and incorrect retrieval trials were not driven by boundary-related theta effects (correct vs. incorrect, $p = 0.033$, correct vs. visible, $p = 0.003$, incorrect vs. visible, $p = 0.401$, $n_{\text{channels}} = 19$, Extended Data Fig. 1e).”

Extended Data Figure 1e: Supplementary analyses for successful memory modulation of theta. MTL theta bandpower significantly increased during correct but not incorrect retrieval or visible halo trials for non-boundary positions, corrected using false discovery rate [FDR]. Crosses (+) represent the mean norm'd (normalized) bandpower across all trials for individual channels where separate colors correspond to individual participants. * = $p < 0.05$, ** = $p < 0.01$, *** = $p < 0.001$.

C3: The environment is pretty small, which I understand reflects some practical limits on engineering/space etc. Do the theta power differences persist when individuals are looking at (but physically near) the boundary areas? The confound I am worried about is related to the amount of visual information available at the boundaries, with more pictures/complexity present. I don't know exactly what the best control analysis would be for this but it would help nail down the findings as related to boundaries specifically rather than something related. Similar to above, the boundary cutoff is somewhat arbitrary. Is there a relationship between linear boundary proximity and theta power to support the result using the cutoff value?

Response: We thank the reviewer for raising this point and agree that the more complex visual information available on the walls may represent a possible confound. To address this, we have examined whether the increase in theta bandpower in boundary relative to inner positions persists during instances when the participant is moving towards the nearest wall (when there is increased visual information available) relative to instances when the participant is moving away from the nearest wall (e.g. instances with less visual information available). As shown in Extended Data Fig. 7d, boundary modulation of theta bandpower persists both when participants are walking towards and away from walls, suggesting that the amount of complex visual information available is not driving the boundary modulation of theta bandpower.

Extended Data Figure 7d: Boundary versus inner distance thresholds and control analyses. (d) Mean \pm s.e.m. normalized (norm'd) theta bandpower (4-6 Hz) during arrow trials, after excluding the 0.5 m prior to participant arrival to a visible arrow on each trial for instances when participants are moving towards (left) or away (right) from boundaries. Crosses (+) represent mean power averaged over MTL channels for each participant. *** = $p < 0.001$.

We have added these results to our manuscript (pg. 11-12, lines 284-289):

“To examine whether encoding of visual information on walls was contributing to boundary-modulation of theta power, we examined theta bandpower fluctuations in two separate conditions: when participants were moving towards the (nearest) wall and when participants were moving away from the (nearest) wall. We observed that boundary-modulation of theta bandpower persisted in both conditions (towards: $p < 0.001$, away: $p < 0.001$, $n_{channels} = 19$, Extended Data Fig. 7d).”

and (pg. 14, lines 359-362):

“Importantly, boundary modulation of theta bandpower persisted in conditions both when participants approached and moved away from the wall, suggesting that the visual information available when facing a wall was not driving this spatial representation.”

The reviewer brings up an important question regarding the selection of the boundary threshold. We initially selected a boundary cutoff of 1.2 m based on previous work from our group (Stangl et al, 2021)⁵ using the same room environment that also performed control analysis to support that selection and we also performed two additional analyses to address this. The first was to repeat our analyses for a variety of boundary cutoffs, which showed that the effects are not dependent on our specific boundary cutoff (Extended Data Fig. 7a):

Extended Data Figure 7a: Boundary versus inner distance thresholds and control analyses. (a) Mean \pm s.e.m. normalized (norm'd) theta bandpower (4-6 Hz) during arrow trials, after excluding the 0.5 m prior to participant arrival to a visible arrow on each trial across MTL channels ($n_{\text{channels}} = 19$) for boundary and inner positions using varying threshold definitions for cutoff ranging between 0.8 - 1.8 m. Crosses (+) represent mean power averaged over MTL channels for each participant. *** = $p < 0.001$.

We have added these results to our manuscript (pg. 10, lines 250-251):

"...(although, see Extended Data Fig. 7a for additional cutoffs used)."

and (pg. 12, lines 300-302):

"Lastly, boundary-related increases in theta power were not dependent on the specific boundary vs. inner 1.2 m cutoff (Extended Data Fig. 7a)."

The second approach was to include multiple variables, including distance to boundary, in the linear mixed model (as described above, in response to the reviewer's comment C1) to determine differential contributions to fluctuations in MTL theta bandpower. We found that distance to boundary was a significant predictor of MTL theta bandpower during the arrow search but not memory retrieval condition. Importantly, the distance to boundary was a continuous predictor variable (e.g. bypasses the use of an arbitrary boundary cutoff) suggesting that there is a linear relationship between proximity to boundary and theta bandpower increases, without using a categorical definition of boundary vs. inner positions.

Extended Data Figure 3c: Simultaneous impact of multiple variables on theta power. Linear mixed-effects models were calculated to predict each participant's normalized theta bandpower timeseries by a range of predictor variables that were fixed effects with samples blocked according to channel identity. Asterisks denote a significant impact of a variable's beta weight on theta power, * = $p < 0.05$ for $n_{\text{participants}} = 6$. Error bars show the s.e.m. of the beta weights across participants. Model was implemented for the arrow search period. Movement speed, angular velocity, proximity to boundary (distance to nearest wall), and proximity to arrow (trial-specific position of arrow) were continuous variables. Movement direction was a categorical variable comprised of 12 possible binned

movement directions (with the mean beta coefficient over all 12 directional bins depicted). Crosses (+) represent the variable impact (beta) colored according to participant.

We have added these results to our manuscript (pg. 11, lines 272-278):

“Moreover, while proximity to boundary was not a significant predictor of theta bandpower in the previously described linear mixed-effects model during memory retrieval (Extended Data Fig. 3b), we found that during arrow search periods only distance to (nearest) boundary was a significant predictor of elevated theta bandpower, whereas proximity to the visible arrow cue (distance to arrow) were not (proximity to arrow, $p = 0.267$; proximity to boundary, $p = 0.048$; $n_{\text{participants}} = 6$, Extended Data Fig. 3c), suggesting that there is a linear relationship between boundary proximity and theta bandpower.”

C4: Another concern is related to movement speed. The control analysis shown in extended data includes a ($df=5$) comparison of average speed for the different conditions, which does not reach significance, but there are some differences across individuals and the means. The authors should perform a standard multivariate analysis to show that corr/error differences persist when including movement speed as a factor in the analysis. This same concern applies to whether individuals are stationary or non-stationary at the time of .5 sec window prior to button press.

Response: We thank the reviewer for this great suggestion and now include additional analyses as requested. Specifically, we included movement speed as a predictor variable in our linear mixed-effects model (as described above, C1), in order to examine how movement speed as well as other movement-related and behavioral variables (angular velocity, movement direction, distance to recall, distance to boundary, correct/incorrect trials, accuracy) simultaneously contributed to theta bandpower fluctuations during the memory retrieval period or the arrow search period, respectively.

During memory retrieval, across participants, only memory-related variables (shorter distance to recall and smaller errors) were significant predictors of increased theta bandpower, while movement-related variables (speed, angular velocity, movement direction) were not. Using the same approach, we found that during arrow search periods only distance to boundary was a significant predictor of elevated theta bandpower while movement variables (speed, angular velocity, movement direction) and proximity to the visible arrow cue (distance to arrow) were not. Altogether, these results suggest successful memory retrieval and boundary-related theta increases are not driven by movement speed, or other movement-related variables more generally.

Extended Data Figure 3b-c: Simultaneous impact of multiple variables on theta power. Linear mixed-effects models were calculated to predict each participant’s normalized theta bandpower timeseries by a range of predictor variables that were fixed effects with samples blocked according to channel identity. Asterisks denote a significant impact of a variable’s beta weight on theta power, $* = p < 0.05$ for $n_{\text{participants}} = 6$. Error bars show the s.e.m. of the beta weights across participants. Models were implemented for (b) the entire retrieval period (when no halos were visible) and (c) the arrow search period. Movement speed, angular velocity, proximity to recall (distance to button press), proximity to boundary (distance to nearest wall), proximity to arrow (trial-specific position of arrow), and error (distance between the position of recall and target location) were continuous variables. Whether a trial was correct or incorrect (correct/incorrect) was a binary variable and movement direction was a categorical variable comprised of 12 possible binned movement directions (with the mean beta coefficient over all 12 directional bins depicted). Crosses (+) represent the variable impact (beta) colored according to participant.

In addition, as suggested by the reviewer, we also modified Extended Data Fig. 2 to illustrate the mean speed between conditions on a participant level during the last 0.5 seconds before retrieval (button press).

Extended Data Figure 2: Movement speed during different task conditions. Mean speed (\pm standard error of the mean [s.e.m.]) across participants (crosses (+), $n_{\text{participants}} = 6$) compared between task conditions for (a) correct versus incorrect retrieval trials during the 0.5 s prior to recall, (b) first versus second context, (c) before versus during the 0.5 s prior to recall, (d) retrieval (excluding 0.5 s prior to recall) versus arrow (excluding 0.5 m preceding arrival at arrow) trials, (e) retrieval (excluding 0.5 s prior to recall) versus visible feedback (excluding 0.5 s preceding arrival at visible halo position), and (f) positions in the boundary (< 1.2 m from walls) versus inner area (> 1.2 m from walls) of the room. ns = $p > 0.05$, * = $p < 0.05$.

Finally, the reviewer raises an excellent question about whether participants were stationary or non-stationary during this 0.5 second window. We examined the speed profiles of participants during the 0.5 second window prior to button press and found that almost all of the data during this time period is during instances when the participants are moving. In fact, we find that on average, $\sim 90\%$ of the data in this time window is during movement and for most participants, the percentage of time spent moving in this time window is closer to 95-99%. This leaves $\sim 10\%$ of the data in this 0.5 second time window arising from instances when participants are stationary, and these stationary periods occur, almost exclusively, in two participants. However, to control for the possibility that the memory-related effect observed in this time window is confounded by a difference between movement and stationary periods, we computed the difference between correct and incorrect trials in this time window after excluding instances when the participants were stationary and find that the memory-related increase in theta bandpower persists.

To further control for movement speeds during the 0.5 seconds prior to recall, we constructed a linear mixed effects model to predict theta bandpower specifically in the last 0.5 seconds prior to recall and included movement speed and the correct (relative to incorrect) trial performance as predictor variables. We found that only correct (relative to incorrect) trial performance and not movement speed was a significant predictor of theta bandpower increases in the last 0.5 seconds prior to recall.

Extended Data Figure 3a: Simultaneous impact of multiple variables on theta power. Linear mixed-effects models were calculated to predict each participant’s normalized theta bandpower timeseries by a range of predictor variables that were fixed effects with samples blocked according to channel identity. Asterisks denote a significant impact of a variable’s beta weight on theta power, * = $p < 0.05$ using a one-sided permutation test for $n_{\text{participants}} = 6$. Error bars show the s.e.m. of the beta weights across participants. Models were implemented for (a) the last 0.5 s of the retrieval period (when no halos were visible). Movement speed was a continuous variable. Whether a trial was correct or incorrect (correct/incorrect) was a binary variable. Crosses (+) represent the variable impact (beta) colored according to participant.

As we have previously noted (see C1, C3), we have added these results to our manuscript. For completeness, we are copying the excerpt from the manuscript again (pgs. 7-8, lines 169-184):

“Furthermore, we quantified the impact of movement speed and correct relative to incorrect memory performance on changes in theta bandpower during the last 0.5 s prior to recall on a trial-by-trial basis using a linear mixed-effects model and found that only correct performance but not movement speed significantly predicted increases in theta bandpower during this time window (correct vs. incorrect, $p = 0.028$; movement speed, $p = 0.337$; $n_{\text{participants}} = 6$, Extended Data Fig. 3a).

Next, we examined the simultaneous contribution of multiple behavioral variables (distance to recall, distance to boundary, correct vs. incorrect performance, distance error) and movement-related variables (movement speed, angular velocity, movement direction) on fluctuations in theta bandpower during the entire duration of retrieval trials, until the instant of recall (when no cues were present; Extended Data Fig. 3b). Specifically, proximity to recall (distance to button press or instant of recall) and distance error (distance between button press and target location) were significant predictors of theta bandpower fluctuations (movement speed, $p = 0.37$; angular velocity, $p = 0.998$; proximity to recall, $p = 0.044$; proximity to boundary, $p = 0.741$; correct vs.

incorrect, $p = 0.340$; distance error, $p = 0.046$; movement direction = 0.290; $n_{\text{participants}} = 6$, Extended Data Fig. 3b).”

and (pg. 11, lines 272-277):

“Moreover, while proximity to boundary was not a significant predictor of theta bandpower in the previously described linear mixed-effects model during memory retrieval (Extended Data Fig. 3b), we found that during arrow search periods only distance to (nearest) boundary was a significant predictor of elevated theta bandpower, whereas proximity to the visible arrow cue (distance to arrow) were not (proximity to arrow, $p = 0.267$; proximity to boundary, $p = 0.048$; $n_{\text{participants}} = 6$, Extended Data Fig. 3c), suggesting that there is a linear relationship between boundary proximity and theta bandpower.”

and (pgs. 13-14, lines 335-338):

“Moreover, we also found that a continuous metric of memory performance (distance error) was linearly related to changes in theta bandpower over the entire duration of retrieval trials, suggesting that memory retrieval success modulated theta power fluctuations throughout the retrieval period.”

C5: *Regarding the statistical methods, I believe the main result is based on 1) calculating a power difference across the mean power for the corr/error trials 2) shuffling the corr/error trial labels, recalculating the power difference, and comparing the true values to the distribution of shuffled power differences. But I don't totally understand when the p value is extracted from the true/null distribution comparison (I'm sorry if I missed it). Some additional clarification is required. I don't think the shuffle occurs at the level of electrode, because that would only have 19 values to shuffle. The authors should also report the statistical result (in terms of the test statistic used) for the non-bootstrapped test as well as the shuffle test, which would make it easier to interpret the results with df reported.*

Response: We thank the reviewer for raising this point and have now added clarification regarding our statistical methods (pgs. 38-39, lines 748-777):

“Statistical comparisons between two conditions were performed using a paired-sample permutation test as follows: To compare two paired arrays of values (e.g., each of the recording channels' average bandpower during 'correct' versus 'incorrect' trials, or in the 'boundary' versus the 'inner' room area), the paired-sample permutation test calculates whether the mean difference between paired values is significantly different from zero. It estimates the sampling distribution of the mean difference under the null hypothesis, which assumes that the mean difference between the two conditions (correct vs. incorrect, or boundary vs. inner) is zero, by shuffling the condition assignments and recalculating the mean difference many times ($n_{\text{perm}} = 1000$). The observed mean difference between conditions was then compared to this null distribution as a test of significance. The key steps of this procedure are described in more detail below.

Step 1: The observed difference between conditions is calculated, by first calculating the difference between conditions for each value pair (condition 1 value – condition 2 value), and then calculating the average difference across pairs.

Step 2: Condition labels are randomly shuffled within each value pair and the difference between 'shuffled conditions' is calculated, by first calculating the difference between randomly labeled conditions for each value pair (value randomly labeled with condition 1 – value randomly labeled with condition 2), and then calculating the average difference across pairs. This step is repeated n_{perm} times (in n_{perm} permutations), to generate a distribution of n_{perm} “random differences” between conditions.

Step 3: The observed difference between conditions (calculated in step 1) is compared with the distribution of random differences from shuffling condition labels (calculated in step 2). The p -value is calculated by the number of random differences that are larger than the observed difference, divided by the total number of samples in the distribution. Two-sided permutations tests were used unless otherwise noted.

For multiple comparisons correction (e.g., when performing statistical tests for multiple frequency steps in a bandpower analysis, or across multiple conditions), *p* values were adjusted using the false discovery rate (FDR)^{18,19}. For top-down maps of theta bandpower (e.g., Fig. 3-5), the room was divided into 19 x 19 bins. Mean bandpower over condition was computed for each bin, specifically the bandpower for all samples in which the participant was positioned in a bin was summed, then divided by the number of samples the participant occupied in that bin. A gaussian smoothing kernel of 0.2 standard deviations was applied to this heatmap, normalized to the peak power and finally, interpolated (using MATLAB function 'interp2' with *k* = 7) for visualization."

For the permutation test, there are no test statistics or degrees of freedom to report, only *p* values. Additionally, to further validate the results of our permutation tests, we carried out parametric equivalents to the permutation tests (e.g. t-tests) and found that both approaches (permutation tests versus t-tests) led to virtually identical results. We have included here a comparison of *p*-values for the main effects using a permutation test and using a t-test:

	Permutation	T-test
Correct vs. incorrect	0.003	0.0051
Close vs. far	0.002	0.0148
Boundary vs. inner	< 0.001	< 0.001

C6: Finally, there are only 6 subjects, which I think is ok given the difficulty of obtaining these really interesting data. This precludes a random effects analysis. But, the authors should add some confidence to the findings by showing that the main results related to corr/error and boundary hold when iteratively throwing out one subject at a time. This will not affect the *df* (less than 19 contacts) too much but would help to show that the results are not driven by strong effects in one individual.

Response: As the reviewer suggests, we repeated our main analyses when iteratively excluding one participant at a time, which did not change our overall findings (Extended Data Fig. 1d).

d

Leave-one-out p values	
Excluded Participant	Correct vs. incorrect
1	0.009
2	< 0.001
3	0.004
4	0.002
5	0.002
6	0.003

Extended Data Figure 1d: Supplementary analyses for successful memory modulation of theta. Correct versus incorrect increases in MTL theta bandpower remained significant when using a leave-one-out approach where each of the 6 participants were excluded one at a time.

e

Leave-one-out p values	
Excluded Participant	Boundary vs. Inner
1	< 0.001
2	< 0.001
3	< 0.001
4	< 0.001
5	< 0.001
6	< 0.001

Extended Data Figure 7e: Boundary versus inner distance thresholds and control analyses. Leave-one-out approach with analysis run after excluding one participant each time and associated statistic shown.

Moreover, in response to comments from other reviewers, we have also performed additional analyses to further confirm that our findings are not driven by individual participants. We repeated our analyses for each individual participant separately (Extended Data Fig. 1c; Extended Data Fig. 7b), as well as across participants (rather than channels; averaging across all channels within a participant; Extended Data Fig. 1b; Extended Data Fig. 7c). Altogether, we believe these new results provide confidence in the findings in that they are not driven by single participants, but are instead consistent across individuals (Extended Data Fig. 1, Extended Data Fig. 7):

Extended Data Figure 1b-c: Supplementary analyses for successful memory modulation of theta. (b-c) MTL theta bandpower in the 0.5 s prior to recall (button press) for correct and incorrect trials (b) illustrated across channels in individual participants and (c) when averaging over individual channels per participant ($n_{\text{participants}} = 6$). Crosses (+) represent the mean norm'd (normalized) bandpower across all trials for (a,b,e) individual channels and (c) across channels in a participant where separate colors correspond to individual participants. * = $p < 0.05$.

Extended Data Figure 7b-c: Boundary versus inner distance thresholds and control analyses. Mean \pm s.e.m. normalized (norm'd) theta bandpower (4-6 Hz) during arrow trials, after excluding the 0.5 m prior to participant arrival to a visible arrow on each trial (b) in MTL channels for each participant for boundary and inner positions using a 1.2 m threshold, and (c) when averaging across each participant's individual channels ($n_{\text{participants}} = 6$). Crosses (+) represent mean power averaged over MTL channels for each participant. *** = $p < 0.001$.

We have added these results to the manuscript (pg. 6, lines 144-149):

“However, this finding... was numerically present across participants (Extended Data Fig. 1b), persisted when averaging over channels for each participant ($p < 0.001$, $n_{\text{participants}} = 6$, Extended Data Fig. 1c), and remained after a leave-one-out approach when each participant's data was excluded one at a time (Extended Data Fig. 1d), suggesting that findings were not driven by individual subjects.”

and (pg. 12, lines 300-306):

“Lastly, boundary-related increases in theta power...occurred within individual participants (Extended Data Fig. 7b), when averaged over individual channels of participants ($p < 0.001$, $n_{\text{participants}} = 6$, Extended Data Fig. 7c), and persisted during a leave-one-out approach when each participant’s data was excluded one at a time (Extended Data Fig. 7e), suggesting that findings were consistent across participants and not driven by individual subjects.”

C7: *I would like to reiterate that this is a novel dataset, difficult to acquire, that will be of interest to a wide range of investigators. I think it has great potential for impact.*

Response: We thank the reviewer for their positive feedback and for their time in reviewing the manuscript.

Additionally, we would like to highlight a modification to the manuscript regarding the Morlet wavelet parameter used to compute oscillatory power as we would like to raise this point for full transparency. Please refer to our detailed discussion in response to Reviewer #2, C14.

Reviewer #2 [R2]: *The authors examine intracranial EEG recordings from the medial temporal lobe (MTL) of freely ambulating people with epilepsy completing a spatial memory task in immersive virtual reality. They demonstrate that accurate spatial memory retrieval is associated with increased theta power (in the 6-8Hz band) immediately preceding a button press response; that increased theta power (in the same frequency band) is observed when participants pass close to (<2m) invisible, previously learned non-target goal locations; and that increased theta power (in the 4-6Hz band) is observed when participants are located close to the boundaries of the environment in visually guided navigation (‘arrow’) trials.*

While these data are difficult to collect and rely on a relatively new (and exciting) invasive recording technology, the findings described here are not particularly novel (aside from the increases in theta power associated with proximity to non-target goal locations, which is not extensively investigated) and come from a relatively small sample (of $n=6$ patients). As such, it is hard to see how these results provide any new insight into the role of MTL theta in human memory function. Nonetheless, they add to a growing literature that utilises an ethological approach to study human memory function and demonstrate that some previous findings from immobile, desktop VR paradigms (i.e. increased MTL theta power during accurate memory retrieval, described in references 24-27 and others reviewed in reference 11) can be translated to more realistic contexts, which may be of interest to the community.

Response: We thank the reviewer for taking their time to review the manuscript and for their suggestions. Please find our responses to each additional comment below.

Major Concerns:

C1: *All statistical analyses consider each electrode contact to be an independent sample ($n=19$ channels), despite the fact that many pairs of channels are located millimetres apart in the same brain tissue. This is not an appropriate use of statistics, although I appreciate that (sadly) it is fairly standard in this field of research. Ideally, the authors should repeat their analyses with data averaged across channels in each patient as $n=6$ independent samples. At the very least, the authors should include some indication of whether these effects are present in each patient (and / or hemisphere), and colour code the scatter points on Figures 3H, 4B, and 5B to indicate which data come from which patient.*

Response: We thank the reviewer for this important comment. In response, we have now updated all of the results to include color coded individual data points for each participant in the study. Moreover, in response to this comment as well as comments from reviewer #1 and #3, we have performed several additional analyses to show that our findings are statistically robust and not driven by individual participants: First, we repeated our main analyses when iteratively excluding one participant at a time, which did not change our overall findings (Extended Data Fig. 1d). Second, we show the main effects (i.e., comparisons of theta power for correct vs. incorrect trials, and for boundary vs. inner room areas) for each individual participant separately (Extended Data Fig. 1c, Extended Data Fig. 7b), not only on a group level. And third, we performed these main comparisons across participants (rather than across channels) by averaging across all channels within a participant (Extended Data Fig. 1b, Extended Data Fig. 7c). Together, we believe that these additional analyses and illustrations provide a transparent presentation of data from individual participants, and suggest that our results are robust across participants and not driven by a single individual (Extended Data Figure 1, Extended Data Figure 7):

Extended Data Figure 1b-d: Supplementary analyses for successful memory modulation of theta. (b-c) MTL theta bandpower in the 0.5 s prior to recall (button press) for correct and incorrect trials **(b)** illustrated across channels in individual participants and **(c)** when averaging over individual channels per participant ($n_{\text{participants}} = 6$). **(d)** Correct vs. incorrect increases in MTL theta bandpower remained significant when using a leave-one-out approach where each of the 6 participants were excluded one at a time. Crosses (+) represent the mean norm'd (normalized) bandpower across all trials for **(a,b,e)** individual channels and **(c)** across channels in a participant where separate colors correspond to individual participants. * = $p < 0.05$.

Extended Data Figure 7b,c,e: Boundary versus inner distance thresholds and control analyses. (b-c) Mean \pm s.e.m. normalized (norm'd) theta bandpower (4-6 Hz) during arrow trials, after excluding the 0.5 m prior to participant arrival to a visible arrow on each trial **(b)** in MTL channels for each participant for boundary and inner positions using a 1.2 m threshold, and **(c)** when averaging across each participant's individual channels ($n_{\text{participants}} = 6$). Crosses (+) represent mean power averaged over MTL channels for each participant. *** = $p < 0.001$. **(e)** Leave-one-out approach with analysis run after excluding one participant each time and associated statistic shown.

We have added these results to the manuscript (pg. 6, lines 144-149):

“However, this finding... was numerically present across participants (Extended Data Fig. 1b), persisted when averaging over channels for each participant ($p < 0.001$, $n_{\text{participants}} = 6$, Extended Data Fig. 1c), and remained after a leave-one-out approach when each participant's data was excluded one at a time (Extended Data Fig. 1d), suggesting that findings were not driven by individual subjects.”

and (pg. 12, lines 300-306):

“Lastly, boundary-related increases in theta power were not dependent on the specific boundary vs. inner 1.2 m cutoff (Extended Data Fig. 7a) and occurred within individual participants (Extended Data Fig. 7b), when averaged over individual channels of participants ($p < 0.001$, $n_{\text{participants}} = 6$, Extended Data Fig. 7c), and

persisted during a leave-one-out approach when each participant's data was excluded one at a time (Extended Data Fig. 7e), suggesting that findings were consistent across participants and not driven by individual subjects.”

C2: On a related note, the statistical analyses of power spectra (i.e. as illustrated in Fig 3A-C and 5A) are not described in sufficient detail. As far as I can tell, the authors examined 118 different frequency bands (from 3 to 120Hz in steps of 1Hz) and then used FDR correction to find significant increases or decreases in power between conditions, but it is not clear if they applied any cluster correction, or if the power increases illustrated in Fig 3A and 5A just happen to be restricted to adjacent 3Hz bands? Moreover, it is not clear – in the memory retrieval analysis – how the authors compared the 0.5s period prior to recall with other time windows (or which other time windows were examined); or what is meant by the statement: “memory-retrieval-related increases in MTL theta bandpower were not present over the entire retrieval period”. Please clarify

Response: We have added clarification to our statistical analyses. Specifically, we compared power increases (3-120 Hz) for each frequency in 1 Hz steps (118 total) during correct and incorrect retrieval trials for the entire duration of retrieval trials. Next, we applied an FDR correction across all (118) p-values to account for the number of statistical tests performed. Indeed, as the reviewer has pointed out, we empirically observed that the 6, 7, and 8 Hz frequency steps were significantly elevated in correct trials but not incorrect trials and that the 4, 5, and 6 Hz frequency steps were significantly elevated in boundary but not inner positions, which is why we then used these frequency bands for our subsequent analyses. We did not apply any cluster corrections; rather, we assume that the consecutive nature of the significant 3 Hz bands present in Fig. 3A and 5A reflects the strength of theta bandpower oscillations in these frequency ranges in relation to these memory and spatial processes.

We thank the reviewer for pointing out that our statement regarding this effect over memory retrieval trials is unclear. In response to this comment as well as related comments from other reviewers (e.g., reviewer # 1 C1), we performed additional analyses that shed further light on to the specific time periods during which the memory retrieval effect persists.

Specifically, we investigated successful memory-related increases in MTL theta bandpower over a range of time windows prior to recall (button press) and found that 6-8 Hz theta bandpower is significantly higher in correct relative to incorrect trials in time windows ranging between 0.2-1 s prior to recall, suggesting that this effect does not depend on the selected 0.5 s time window.

Extended Data Figure 1a: Supplementary analyses for successful memory modulation of theta. MTL theta bandpower in the 0.5 s prior to recall (button press) for correct and incorrect trials across various time windows within 1 s (0.25 – 1 s, $n_{\text{channels}} = 19$). Crosses (+) represent the mean norm'd (normalized) bandpower across all trials for individual where separate colors correspond to individual participants. * = $p < 0.05$, ** = $p < 0.01$, *** = $p < 0.001$.

We have added these results to the manuscript (pg. 6, lines 144-145):

“However, this finding was not dependent on the specific temporal window (0.5 s, Extended Data Fig. 1a).…”

Additionally, in response to other reviewer's comments (e.g., reviewer #1 C1, reviewer #3 C7, and reviewer #1 C4), we have completed two additional analyses that provide insight into the time windows relevant to memory-retrieval-related increases in theta bandpower.

First, we examined whether memory performance modulated theta bandpower immediately after cue presentation. As discussed previously (reviewer #1 C1, Extended Data Fig 4a), after cue presentation, theta power initially appears to be similar between correct and incorrect trials, but a significant difference can be detected around 1.5 seconds after cue onset.

Please note, however, that our experimental task was largely self-paced and participants were freely moving, and thus we cannot fully determine at what exact point in time participants retrieved the memory of the actual target location. In fact, participants could freely decide at what point in time after cue onset they would start to move, and some participants may have had multiple movement onset periods (for instance, a participant may have started to walk immediately after cue onset, then stop again, think about the correct target location and perhaps re-consider their initial choice, before walking to the actual recalled position). As a result, we expect the time window immediately after cue onset to contain substantial 'noise' (i.e., variability between participants and trials), as this time window likely contains data from some trials when participants retrieved the correct memory and other trials when they retrieved the correct memory at a later point in time. Thus, we hypothesized that another interesting period most likely captures a time window critical for memory retrieval: the last movement onset before button press (i.e., the time when participants likely made their final decision as to where to go next and where to press the button). Interestingly, another reviewer raised a closely related point, and asked whether differences in theta power for correct vs. incorrect trials are present after cue onset and during immobility (i.e., before movement onset) of participants, and whether these effects might be present in the low-theta frequency range (see Reviewer 3, C7 for more details).

In response, we performed an additional analysis to test whether low-frequency (around 3-6 Hz) and high-frequency (around 6-8 Hz) theta oscillations exhibit differences in power between correct and incorrect trials after cue presentation and before movement onset. Indeed, in line with the reviewer's hypothesis, we found that low-frequency theta power was increased already around 0.5 after cue onset, as well as around movement onset (Extended Data Fig. 4c-d).

Together, these results suggest that, regarding differences in theta power between correct and incorrect trials, the earliest effects directly after cue onset and before movement onset are present in the low-frequency theta band (around 3-6 Hz), while later effects around 1.5 sec after cue onset and before button press appear to be more prevalent in the higher theta range (around 6-8 Hz).

Extended Data Figure 4: Successful memory modulation of theta bandpower after cue and movement onset. Mean (\pm s.e.m.) norm'd (normalized) theta bandpower across MTL channels ($n_{\text{channels}} = 19$) (a,c) after cue presentation and (b,d) around movement onset (last movement onset prior to button press in each retrieval trial) in (a-b) 6-8 Hz and (c-d) 3-6 Hz bandpower. Note, halos were not visible during correct (green) or incorrect (red) retrieval trials. Gray bar indicates timepoints where $p < 0.05$ (one-sided permutation test at each time point, representing 4 ms steps at 250 Hz sampling rate).

We have added a discussion of this to the manuscript (pg. 8, lines 185-199):

“We also explored successful memory-related theta bandpower changes during other time periods (Extended Data Fig. 4) and found that while theta bandpower (6-8 Hz) initially appeared to be similar between correct and incorrect trials after initial cue presentation during retrieval trials, a difference was detected around 1.5 seconds after cue onset (Extended Data Fig. 4a). Previous work has suggested that, within a broader theta frequency range, low frequency theta oscillations (e.g. type II theta) are related to episodic memory and higher frequency theta oscillations (e.g. type I theta) are movement-related^{2,23}. As such, we also investigated differences between correct and incorrect trials in low frequency oscillations (3-6 Hz) after cue presentation, and before the onset of movement. Since participants often had multiple movement onset periods within a single trial, we specifically examined the last movement onset before button press, which we hypothesized would better capture the time window when participants initiated memory retrieval to determine their final recalled position for the indicated target halo on any given trial. Indeed, we found that low-frequency theta power was increased already around 0.5 s after cue onset (Extended Data Fig. 4c) and prior to movement onset during retrieval trials (Extended Data Fig. 4d).”

and (pg. 13, lines 330-335):

“In line with this hypothesis, we found elevated low-frequency theta bandpower (e.g. memory-related type II theta) in two time windows associated with less movement: around (1) 0.5 s after cue presentation and (2) 0.5 s prior to movement onset, while there was elevated higher-frequency theta bandpower (e.g. movement-related type I theta) in two time windows associated with more movement: (1) around 1.5 s after cue presentation and (2) in the 0.5 s prior to recall^{2,23}.”

Second, we used a linear mixed effects model (a type of general linear model) to evaluate the simultaneous impact of movement-related (movement speed, angular velocity, movement direction) and behavioral variables (proximity to recall, proximity to boundary, correct/incorrect performance, distance error) on theta bandpower

over the entire retrieval trial period, while taking into account repeated measures from different recording channels with different amounts of data per participant. In this model, we found that after accounting for all variables, proximity to recall (distance to the point of subjective recall) and distance error (distance between button press and target location) were significant predictors, suggesting that there is a linear relationship between theta bandpower and a continuous metric of memory performance (as opposed to a categorical ‘correct’ vs. ‘incorrect’ distinction).

Extended Data Figure 3b: Simultaneous impact of multiple variables on theta power. Linear mixed-effects models were calculated to predict each participant’s normalized theta bandpower timeseries by a range of predictor variables that were fixed effects with samples blocked according to channel identity. Asterisks denote a significant impact of a variable’s beta weight on theta power, $* = p < 0.05$ for $n_{\text{participants}} = 6$. Error bars show the s.e.m. of the beta weights across participants. A model was implemented for the entire retrieval period (when no halos were visible). Movement speed, angular velocity, proximity to recall (distance to button press), proximity to boundary (distance to nearest wall), and error (distance between the position of recall and target location) were continuous variables. Whether a trial was correct or incorrect (correct/incorrect) was a binary variable and movement direction was a categorical variable comprised of 12 possible binned movement directions (with the mean beta coefficient over all 12 directional bins depicted). Crosses (+) represent the variable impact (beta) colored according to participant.

Altogether, these results suggest that memory-related modulation of theta bandpower is robustly present prior to recall, and furthermore, theta bandpower has a linear relationship with a continuous memory performance (distance error) metric throughout memory retrieval trials. We have added the additional analyses and findings to the revised version of our manuscript (pg. 7-8, lines 175-184), and given these new insights, we have also removed our original statement mentioned by the reviewer, regarding the absence of “memory-retrieval-related increases in MTL theta bandpower ... over the entire retrieval period”, from the manuscript.

“Next, we examined the simultaneous contribution of multiple behavioral variables (distance to recall, distance to boundary, correct vs. incorrect performance, distance error) and movement-related variables (movement speed, angular velocity, movement direction) on fluctuations in theta bandpower during the entire duration of retrieval trials, until the instant of recall (when no cues were present; Extended Data Fig. 3b). Specifically, proximity to recall (distance to button press or instant of recall) and distance error (distance between button press and target location) were significant predictors of theta bandpower fluctuations (movement speed, $p = 0.37$; angular velocity, $p = 0.998$; proximity to recall, $p = 0.044$; proximity to boundary, $p = 0.741$; correct vs. incorrect, $p = 0.340$; distance error, $p = 0.046$; movement direction = 0.290; $n_{\text{participants}} = 6$, Extended Data Fig. 3b).”

and (pg. 13-14, lines 335-338):

“Moreover, we also found that a continuous metric of memory performance (distance error) was linearly related to changes in theta bandpower over the entire duration of retrieval trials, suggesting that memory retrieval success modulated theta power fluctuations throughout the retrieval period.”

C3: Finally, the analyses illustrated in Figure 4A and B would appear to be the most egregious example of double dipping – the authors first look for a radius that dissociates theta power close to and far from the non-target halo positions, and then show that theta power close to and far from those positions (defined using this radius) is significantly different. It is hard to see how this approach can be justified. It would also be useful to know what proportion of the total environment and what proportion of the boundary region falls into each ('close to' or 'far from') condition.

Response: We thank the reviewer for pointing out the issue with Figure 4a-b. Our intention by presenting the data this way was as follows: As a first step (shown in Fig. 4a), we ran an unbiased (and FDR corrected) statistical test across different radius thresholds (in order to empirically identify the significant thresholds). The second step (shown in Fig. 4b) was not intended to confirm/corroborate the result from the first step, but was supposed to merely reflect an illustration of the same underlying data (providing a higher level of detail by showing individual datapoints) for the selected threshold identified in the first step. However, we agree that it can be misleading to present results of a statistical analysis for Fig. 4b, as this can make the impression of a confirmatory or corroborating analysis. Nevertheless, we still believe that an in-detail illustration of the data for the significant thresholds, as shown in Fig. 4b, might be beneficial to readers for clarity and transparency. Consequently, In the revised version of the manuscript, we now have clarified that Figure 4b is shown for illustrative purposes, highlighting that the diamond in panel a is shown in detail in panel b. We have also rephrased the text in the figure caption and in the main text, to clearly point out to readers that Fig. 4b is included for “illustrative purposes”.

Figure 4: MTL theta bandpower increased at non-target halo positions. (a) Mean normalized (norm'd) theta (6-8 Hz) bandpower during retrieval trials (excluding last 0.5 s prior to recall of target halos) in positions close versus far from non-target halo positions shown across varying radius (distance to halo) thresholds used to determine the cutoff between 'close' and 'far' positions. Gray box highlights radius thresholds where theta bandpower significantly differed between 'close' and 'far' positions ($p < 0.05$, corrected using false discovery rate [FDR]^{19,20}). Diamond indicates radius threshold (2 m) visually illustrated in (b). (b) Detailed view of the mean norm'd theta bandpower across individual recording channels using 2 m threshold (diamond in (a)), shown for illustrative purposes. Crosses (+) represent the mean norm'd bandpower for each channel (colors correspond to individual participants) during retrieval trials, excluding the last 0.5 s prior to recall of a target halo. (c-d) Top-down view of theta bandpower in an example channel across room positions, during retrieval trials, when no visible halos were present, and excluding the 0.5 s preceding recall of target halos. Circles represent positions where target halos were recalled (button press), split by stone (c) and wooden (d) contexts and with colors corresponding to the color of the halo that was recalled. ** = $p < 0.01$.

and (pg. 9, lines 230-233):

“The difference in MTL theta bandpower between ‘close’ and ‘far’ positions peaked at a distance threshold of 2 m from non-target halo positions (distance thresholds of 1.25, 1.5, and 2 m: all $p < 0.05$, $n_{channels \times conditions} = 38$, Fig. 4a, FDR corrected; illustration of 2 m threshold: close vs. far, $p = 0.008$, Fig. 4b).”

Moreover, as per the reviewer’s request, we have compiled the proportion of each condition (“close” and “far”) that falls into the boundary and inner regions and what proportion of the total room environment that the conditions (“close” and “far”) occupy. We computed these proportions using three threshold cutoffs for defining

“close” and “far”, since we found significantly higher theta bandpower in close relative to far using these three cutoffs (Fig. 4a).

Importantly, we find that the relative percentage of boundary to inner regions is lower in close compared to far regions using either threshold, suggesting that boundary modulation of theta bandpower does not drive the non-target modulation of theta bandpower; in fact, these effects opposed each other. In other words, if higher theta power in close relative to far positions was driven by boundary modulation of theta, we would expect there to be a higher ratio of boundary relative to inner positions in the close region, compared to the far regions. However, we observe that there is a much lower percentage of boundary relative to inner positions in the close region, compared to the percentage of boundary and inner positions in the far region.

We thank the reviewer for this insightful comment, especially given that the majority of the data for the far region arises from positions when the participant is located in the boundary of the environment. As such, we further controlled for this by examining whether close vs. far proximity to non-target halo modulation of theta bandpower persists independent of boundary proximity. We completed the same analysis only for instances when the participants are located in the boundary of the environment and find that theta bandpower is also elevated in close relative to far positions.

Altogether, our results highlight that theta bandpower is modulated by close relative to far proximity from non-target halos, independent of boundary proximity.

Extended Data Figure 6d-e: Close vs. far control analyses. (d) Mean normalized (norm'd) theta bandpower in positions close to (< 2 m) versus far away (> 2 m) from non-target halo centers was significantly increased when analyzed only in boundary positions. (e) Percentages of “close” and “far” regions that fall into boundary and inner positions for a 1.25 m, 1.5 m, and 2 m threshold. Total percentage of room area that falls into “close” and “far” regions is also listed. Crosses (+) represent the mean norm'd bandpower for each channel during retrieval trials in each context, excluding the last 0.5 s prior to recall of a target halo and are colored according to participant. ** = $p < 0.01$.

We have added these results to the manuscript (pg. 12, lines 298-300):

“Similarly, non-target modulation of theta bandpower further persisted when examined only in the boundary region of the environment (Extended Data Fig. 6d-e), suggesting that this effect was also not driven by boundary modulation of theta bandpower.”

C4: *It is not clear to me why the authors interpret increased theta power in the proximity of learned goal locations as driven by memory representations of those locations, but increased theta power in the proximity of arrow locations as driven by the environmental boundary, rather than the prominent visual arrow cue that just happens to be located near to the boundary. Can they provide any additional analyses to convince us that theta power increases in arrow trials are genuinely driven by the boundary, and not the visual cue? For example, if they compare theta power close to (i.e. within 2m) and further from (i.e. >2m from) the visible arrow, within the 1.2m boundary region, do they see any power differences? Or if they examine theta power within the 1.2m boundary region, excluding locations around the visual arrow cues, does the effect persist?*

Response: We thank the reviewer for the suggestions. Given that we defined the boundary region as 1.2 m from the wall and that participants spent most of the time in the arrow trials walking across the room rather than along the perimeter of the room, we did not have enough data to compare close (< 2 m) versus far (> 2m) positions from the visible arrow within the 1.2 m boundary region, since this would require data where they are walking along the perimeter of the room. We do find, however, that theta increases persist during boundary compared to inner room positions when excluding 0.5 m around the visible arrow trials.

Figure 5a-b: MTL theta bandpower is modulated by position relative to environmental boundaries. (a-b) Analysis performed over arrow trials, excluding the 0.5 m leading up to arrival at arrows. **(a)** Mean (\pm s.e.m.) normalized (norm'd) difference in power across frequencies (3-120 Hz) and MTL channels ($n_{\text{channels}} = 19$) between positions near (< 1.2 m of walls, based on prior work¹⁶) versus away from boundaries. Significant differences in norm'd power in boundary compared to inner positions occurs for theta frequencies (4-6 Hz, horizontal pink bar = $p < 0.05$, corrected using false discovery rate [FDR]). **(b)** Mean \pm s.e.m. norm'd theta bandpower (4-6 Hz) across MTL channels ($n_{\text{channels}} = 19$) for boundary and inner positions. Crosses (+) represent individual channels with colors corresponding to individual participants. *** = $p < 0.001$.

Further, we used a linear mixed effects modeling approach (as previously described) to determine whether proximity to boundaries (perpendicular distance to nearest boundary) or proximity to arrows (distance to visible arrow on current trial) significantly predicted theta bandpower increases during arrow trials. We found that shorter distance to the (nearest) boundary was the only significant predictor of elevations in theta bandpower, and proximity to the visible arrow and other movement variables (speed, angular velocity, and yaw) were not significant predictors of variations in theta bandpower, suggesting that the effect was driven by boundary proximity rather than the visible arrow cue. It appears that, numerically, proximity to the visible arrow is also negatively associated with theta power; however, this relationship does not reach statistical significance.

Extended Data Figure 3c: Simultaneous impact of multiple variables on theta power. Linear mixed-effects models were calculated to predict each participant's normalized theta bandpower timeseries by a range of predictor variables that were fixed effects with samples blocked according to channel identity. Asterisks denote a significant impact of a variable's beta weight on theta power, $* = p < 0.05$ for $n_{\text{participants}} = 6$. Error bars show the s.e.m. of the beta weights across participants. Model was implemented for the arrow search period. Movement speed, angular velocity, proximity to boundary (distance to nearest wall), and proximity to arrow (trial-specific position of arrow) were continuous variables. Movement direction was a categorical variable comprised of 12 possible binned movement directions (with the mean beta coefficient over all 12 directional bins depicted). Crosses (+) represent the variable impact (beta) colored according to participant.

We have added these results to the revised version of the manuscript (pg. 11, lines 272-278):

“Moreover, while proximity to boundary was not a significant predictor of theta bandpower in the previously described linear mixed-effects model during memory retrieval (Extended Data Fig. 3b), we found that during arrow search periods only distance to (nearest) boundary was a significant predictor of elevated theta bandpower, whereas proximity to the visible arrow cue (distance to arrow) were not (proximity to arrow, $p = 0.267$; proximity to boundary, $p = 0.048$; $n_{\text{participants}} = 6$, Extended Data Fig. 3c), suggesting that there is a linear relationship between boundary proximity and theta bandpower.”

Additionally, we examined whether boundary modulation of theta bandpower persisted in two different sub-conditions of arrow trials (with the 0.5 meters prior to arrival at the visible arrow excluded): when participants were (1) moving towards the nearest wall (e.g. in the direction of the arrow) (2) moving away from the nearest wall (e.g. away from the arrow). We found that theta bandpower was significantly elevated in boundary relative to inner positions both when participants were moving towards and away from boundaries, and specifically during arrow trials (excluding the 0.5 m prior to arrival), suggesting that the boundary modulation was not driven by the visual arrow cues.

Extended Data Figure 7d: Boundary versus inner distance thresholds and control analyses. Mean \pm s.e.m. normalized (norm'd) theta bandpower (4-6 Hz) during arrow trials, after excluding the 0.5 m prior to participant arrival to a visible arrow on each trial for instances when participants are moving towards (left) or away (right) from boundaries. Crosses (+) represent mean power averaged over MTL channels for each participant. *** = $p < 0.001$.

We have added these results to the manuscript (pgs. 11-12, lines 284-289):

“To examine whether encoding of visual information on walls was contributing to boundary-modulation of theta power, we examined theta bandpower fluctuations in two separate conditions: when participants were moving towards the (nearest) wall and when participants were moving away from the (nearest) wall. We observed that boundary-modulation of theta bandpower persisted in both conditions (towards: $p < 0.001$, away: $p < 0.001$, $n_{channels} = 19$, Extended Data Fig. 7d).”

and (pg. 14, lines 359-362):

“Importantly, boundary modulation of theta bandpower persisted in conditions both when participants approached and moved away from the wall, suggesting that the visual information available when facing a wall was not driving this spatial representation.”

Altogether, these results suggest that theta fluctuations during arrow trials are driven by proximity to environmental boundaries and not visual arrow cues. However, since we cannot rule-out the potential impact of proximity to visible cues (e.g., arrows) near the boundary, we have also added a discussion point acknowledging the fact that boundary-related effects in this study could be – at least to some extent – also influenced by non-visible (non-target) arrow positions since visible arrow cues were randomly dispersed along the boundaries of the room (pg. 15, lines 365-367):

“However, although proximity to the (nearest) boundary but not proximity to the visual cue (arrow) was linearly related to theta bandpower, we cannot fully rule out the possibility that visible cues (arrows) contributed in some way to boundary-modulation of theta bandpower.”

C5: *Finally, in terms of motivation and interpretation, there are various theoretical models (including some cited here) that describe why we might expect to see increased theta power during memory retrieval, but it is not clear at all why we might expect increased theta power near the boundaries of an environment. What hypothesis were the authors testing with these analyses? What mechanism might account for these findings? What do they tell us about human MTL memory function?*

Response: We thank the reviewer for this comment. Previous work has shown similar effects, with increased theta power near boundaries, both during ambulatory real-world navigation (Stangl et al., 2021) and during view-based navigation in virtual reality (Lee et al., 2018)⁶; however, the underlying neural mechanisms that give rise to this effect are, to the best of our knowledge, largely unknown. Theoretically, on a single-neuron level, several cell types could contribute to the observed findings. Most directly, increased theta power near boundaries could be related to boundary-related firing of border cells (Solstad et al., 2008)⁷, and boundary-vector cells (Lever et al., 2009)⁸. In addition, given that oscillatory brain signals measured in this study likely

reflect the summed activity of populations of neurons, it is possible that other spatially-tuned neural responses contribute to the observed effects, such as activity from object-vector cells (Høydal et al., 2019)⁹ or increased place cell activity near boundaries, goals, or objects (Barry et al., 2006; Bourboulou et al., 2019; Dupret et al., 2010; Hollup et al., 2001)^{10–13}. Importantly, throughout the existing literature, the above single-neuron mechanisms appear to be driven by spatial variables (e.g., proximity to boundaries for border cells and boundary-vector cells), even in the absence of an overt memory task, therefore suggesting that such spatial representations can occur independent of memory processes.

By analyzing changes in theta bandpower associated with proximity to boundaries, we aimed to replicate the earlier findings (in particular, Stangl et al. 2021, and Lee et al. 2018)^{5,6} and demonstrate that such neural representations of spatial information are present – in parallel – with memory-related representations, and that human MTL oscillatory activity can reflect both spatial information and memory processes in a temporally flexible manner, depending on changing behavioral goals.

In order to convey these points more clearly to readers of our manuscript, we have included a more detailed discussion of the rationale for our analyses and the possible neural mechanisms underlying our findings (pgs. 15-16, lines 388-396):

“Our results demonstrated a linear relationship between theta bandpower fluctuations and proximity to the position of recall (button press) during retrieval trials. Recently reported object vector trace cells³⁸ may represent a population of hippocampal neurons that could contribute to this effect by modulating firing rate patterns to create a vector field pointing to a previously encountered object’s position. Our results also highlight that theta power is modulated by non-visible and non-target locations of previously learned halos in a latent manner. Consistent with this finding, object trace cells in the entorhinal cortex selectively fire in the location of previously encountered object positions, even at long time periods after the object has been removed from the environment³⁹. Finally, we found robust boundary representations elicited in our task consistent with the existence of border cells and boundary-vector cells^{40,41}, which increase their firing rate when animals are near borders of an environment.”

Minor Concerns

C6: *Given that the authors repeatedly allude to the distinction between low and high theta oscillations that has previously been made in the literature, it would be useful for them to state explicitly whether the effects they report are truly specific to the high theta band (i.e. 6-8Hz), or whether the 1Hz high-pass hardware filter simply prevents them from examining low theta. Would it be possible to show the frequency response of this filter as a supplementary figure, to get an idea of how much signal attenuation it introduces across low frequencies?*

Response: We thank the reviewer for raising this important point about the hardware filter. To ensure that low frequencies of interest in the theta band (3-12 Hz) were not affected by the hardware filter (1st order Butterworth bandpass filter in the 1-90 Hz range, with 3 dB attenuation at cutoff frequencies), we visualized the filter response (Extended Data Figure 9). As shown, with the current filter setting, the full theta band (red box) is well outside of the low cutoff of the filter (1 Hz, green vertical line) and the attenuation throughout the theta band is close to 0 (blue curve). As such, it is unlikely that the hardware filter is affecting the frequencies of interest in the current study, be it low or high theta oscillations. We have added this figure and related text to the manuscript (pg. 33, lines 629-630).

“For the duration of the experiment, amplifier settings on the RNS System (320 model) were programmed to apply a 1 Hz high pass filter and a 90 Hz low pass filter (see Extended Data Fig. 8 for filter response).”

Extended Data Figure 8: Hardware filter response. Illustration of filter response profile of RNS System hardware filter settings using a 1st order Butterworth bandpass filter in 1-90 Hz range with 3 dB attenuation. Broad theta bandpower range (3-12 Hz, shaded in red) does not undergo amplitude attenuation (blue line shows amplitude attenuation of 0) and is above the low cutoff at 1 Hz (green line).

C7: It would be useful to add to Table 1 some indication of how much data from each patient was excluded due to IEDs

Response: We thank the reviewer for this suggestion and have added details regarding the percent of IEDs removed from each channel for all participants in Extended Data Table 1:

Participant		P1	P2	P3	P4	P5	P6
Age		50	38	43	54	33	42
Sex		female	male	male	female	female	female
Total retrieval blocks (stone + wooden contexts):		6	12	18	17	20	8
# of trials in retrieval block #1 (Stone)		30	15	18	18	15	69
# of trials in retrieval block #1 (Wooden)		21	30	27	18	21	60
Recording duration (minutes):		32	143	114	103	155	135
Electrode 1	Hemisphere	left	left	left	left	left	left
	Contact Spacing (mm)	3.5	10	3.5	3.5	10	10
	Number of MTL channels	2	1	2	2	1	2
	Ch 1 localization	HP/HP	HP/HP	HP/HP	HP/HP	ERC/PRC	HP/HP
	Ch 1 IED (% excluded)	4	2	5	4	3	10
	Ch 2 localization	HP/HP	Extra-MTL	HP/HP	HP/PRC	Extra-MTL	HP/HP
	Ch 2 IED (% excluded)	4	4	5	4	3	6
Electrode 2	Hemisphere	right	right	right	Left	right	right
	Contact Spacing (mm)	10	10	3.5	10	10	3.5
	Number of MTL channels	1	1	2	2	1	2
	Ch 1 localization	HP/PRC	HP/HP	HP/HP	HP/HP	ERC/PRC	ERC/ERC
	Ch 1 IED (% excluded)	3	4	5	2	3	4
	Ch 2 localization	Extra-MTL	Extra-MTL	HP/HP	HP/HP	Extra-MTL	PRC/PRC
	Ch 2 IED (% excluded)	7	2	5	2	3	7

Extended Data Table 1: Participant demographics, experimental task info, and localizations of electrodes. The number of retrieval blocks and trials for the six participants (P1-6) who completed the ambulatory spatial navigation task in the stone and wooden contexts. Localizations of electrode contact pairs for each bipolar recording channel (Ch): hippocampus (HP), perirhinal cortex (PRC), entorhinal cortex (ERC). Extra-MTL indicates contacts that were localized to regions outside of the MTL. Also shown is the percent of data that contained an inter-epileptic discharge (IED) and thus was excluded (% excluded).

C8: *In the introduction, the authors suggest that rodent hippocampal theta oscillations fall in the 6-8Hz band, but it is usually more like 6-12Hz (6-8Hz is just the frequency band used here) – please edit accordingly. On a similar note, the authors suggest that rodent hippocampal theta is implicated in memory retrieval, but a more representative summary of the literature would be that it is implicated in memory encoding (retrieval is just the focus of this study), so perhaps ‘memory function’ would be more accurate?*

Response: We agree with the reviewer and have revised this sentence accordingly (pg. 3, lines 58-61):

“Current evidence from rodent studies suggests that oscillatory activity in the theta frequency band (~6-12 Hz)⁵ in the MTL supports spatial navigation^{6,7} and successful memory function^{2,8} through its ability to temporally organize neural activity locally and across brain regions^{8,9}.”

C9: *Where the authors cite their own study of iEEG recordings during ‘real world’ movement, they might also cite the study from Bohbot et al. published around the same time in Nature Communications*

Response: We thank the reviewer for this suggestion and now include the study by Bohbot et al. (pg. 3, lines 72-74):

“Recent technological advancements in human mobile neuroimaging¹⁴, however, have enabled the discovery of MTL higher frequency (~7 Hz) theta oscillations that are modulated by physical movement (e.g., walking)¹⁵⁻¹⁷ and proximity to environmental boundaries¹⁶.”

C10: *In the first paragraph of the results, where the authors first describe the structure of the task, they refer to ‘arrow trials’, before these have been introduced or described. This is a little confusing for the reader – perhaps they could describe them as ‘visually guided navigation (‘arrow’) trials’ or similar, for the sake of clarity*

Response: We agree that it may be unclear to the reader to refer to it as an arrow trial before explaining the arrow trial structure, and as such we have revised the sentence accordingly (pg. 4, lines 96-98):

“The spatial memory task consisted of learning (encoding) trials, visually guided navigation (‘arrow’) trials, and memory recall (retrieval) trials (Fig. 1c-f).”

C11: *In the results, the authors state that “...these behavioral findings showcase the ability of ambulatory immersive VR combined with motion tracking to be used to precisely assess spatial memory performance in freely moving human participants with simultaneous iEEG recordings” – but this implies that this is the first demonstration of that technology, which is misleading. Perhaps they could add “consistent with previous studies”, or similar, and cite the 2020 Neuron paper, 2021 Nature paper, and / or recent Nature Neuroscience paper from the same lab that describe this technology in more detail*

Response: We apologize for not clarifying this point sufficiently. While there are previous studies that have used iEEG recordings, motion tracking, and VR (Topalovic et al., 2020; Topalovic et al., 2023)^{15,16}, they did not do so to assess spatial memory. Studies investigating spatial memory (Stangl et al., 2021)⁵ have thus far only used motion capture and iEEG in humans during real-world navigation but not during VR navigation. We have further clarified this point in the discussion (pg. 13, lines 313-315):

“While previous studies have used simultaneous ambulatory iEEG recordings and immersive VR^{15,16}, this is the first to collect empirical data to investigate human spatial memory.”

C12: *Similarly, in the discussion, they state that “this study is the first to our knowledge to combine simultaneous ambulatory iEEG recordings and immersive VR”, but that would appear to be patently false, given the list of previous publications above. Please clarify*

Response: We thank the reviewer for raising this point and have clarified this sentence in the discussion as highlighted in the previous comment (C11, pg. 13, lines 313-315):

“While previous studies have used simultaneous ambulatory iEEG recordings and immersive VR^{15,16}, this is the first to collect empirical data to investigate human spatial memory.”

C13: *In the discussion, “have been done in” is a little informal – perhaps “have been carried out in” would be preferable?*

Response: We have updated the aforementioned sentence in the discussion (pg. 16, line 407-408 as recommended):

“Traditional human neuroimaging studies of memory retrieval and spatial navigation have been carried out in stationary participants viewing stimuli on a computer screen.”

C14: *In the methods, I believe ‘sixth order Morlet wavelet’ should be ‘six cycle Morlet wavelet’*

Response: We thank the reviewer for this observation and have updated the corresponding text in the Methods to reflect this terminology (pg. 36, lines 702-703).

“Time frequency analysis was performed by computing the oscillatory power at individual frequency steps (1 Hz) between 3-120 Hz using the BOSC toolbox with a three cycle Morlet wavelet.”

Please note that in reviewing this comment and completing additional data analyses, we noticed that the calculation of oscillatory power for our time-frequency analysis was using a three-cycle Morlet wavelet, while in the original version of the manuscript we had reported using a six-cycle Morlet wavelet. Consequently, we re-ran our analyses with both a three-cycle and a six-cycle wavelet in order to directly compare results, and we found that neither our findings nor the interpretation of results were affected by this parameter (i.e., none of the results in the manuscript text changed from ‘significant’ to ‘non-significant’ or vice-versa, and all patterns of data were virtually identical between the two analysis approaches). We further consulted with UCLA-internal and external signal processing experts, who confirmed that this parameter should not have a substantial impact on our findings, and that a three cycle Morlet wavelet would be an equally legitimate parameter choice. In response, we have thus updated our description in the Methods section accordingly, which now correctly details the use of a three-cycle Morlet wavelet for all our analyses, and we wanted to raise this point here for full transparency. We have included here a summary of the statistics for the main effects using a 3-cycle and 6-cycle Morlet wavelet.

	Morlet	3-cycle	6-cycle
Correct vs. incorrect		0.003	0.003
Close vs. far		0.002	0.013
Boundary vs. inner		< 0.001	0.001

We have added to this note to the manuscript (pg. 36, lines 703-707):

“We also repeated all analyses with a six cycle Morlet wavelet given previous approaches^{15,16} and the results were qualitatively the same (i.e., none of the results in the manuscript text changed from ‘significant’ to ‘non-significant’ or vice-versa, and all patterns of data were virtually identical between the two analysis approaches), suggesting the robustness of the results with regards to this analysis parameter.”

Reviewer #3 [R3]: *In this manuscript, Maoz and colleagues leverage a unique opportunity to record intracranial EEG from the human medial temporal lobe (MTL) as subjects are freely ambulating. This allows the authors to investigate an important outstanding question in the field, which is the role of theta oscillations in spatial navigation and memory. There has been substantial evidence that such oscillations are prominent in rodents as they navigate spatial environments, but the evidence in humans is mixed. This is largely because most intracranial recordings are captured from subjects who are confined to their hospital bed, and navigation in these cases is limited to simple virtual navigation on a laptop. Here the authors use an RNS system for*

capturing data in an ambulatory setting, a method they have previously published on and demonstrated, to ask what these oscillations in the MTL are doing during a spatial navigation and memory task. In brief, they find that theta oscillations become stronger as individuals approach locations of targets that they have retrieved from memory. In addition, they also find evidence of stronger theta oscillations as individuals approach the boundaries of the environment. The main frequency band identified is in the higher theta range, which is consistent with theta observed in freely moving animals and somewhat distinct from theta that has been described in humans in immobile settings. Together, these data provide valuable insights into the potential role of theta oscillations in the human MTL and would be an important addition to the literature. The experiment is well conducted and the analyses are clear and appropriate. Overall, this is a good manuscript, but there are some suggestions that could either strengthen the conclusions of the work or provide additional novel insights.

Response: We thank the reviewer for their time and effort to review our manuscript, and for their positive feedback. We also thank the reviewer for their suggestions, which we have incorporated in the revised manuscript and detailed below.

C1: *The authors report improved learning over blocks, which would suggest that individuals learn the overall task structure. But if the halos are randomized between blocks, then one should also look at learning within a block. Does this also improve over the trials in a block. Presumably, this is the role of the feedback that is provided. In addition, it would be helpful to also know about improvements in accuracy (as determined by whether the button press was correct or incorrect), to complement the changes in mean spatial error.*

Response: We thank the reviewer for this suggestion.

First, we would like to clarify a pertinent aspect of the task design: The halo colors and associated positions were fixed throughout the task. There were two separate contexts which each had three uniquely-colored and -positioned halos located in each room environment and the halo positions were identical across different blocks. Each block (consisting of 15 total trials; 5 trials per halo in an environment) alternated between contexts (and thus the 3 fixed halos in the context). See below the locations of the three halos within the 'Stone' and 'Wooden' context, which were fixed throughout all task blocks (i.e., did not change throughout the whole experiment), overlaid on one example participant's movement pattern (black lines):

We have also further clarified this point in the manuscript (pg. 4, lines 98-99):

“During encoding trials, participants were instructed to navigate to a halo (Fig. 1c, Supplemental video 1) and learn its spatial location, which was fixed over the course of the task.”

and (pg. 5, lines 112-115):

“Participants completed the task with 15 retrieval and arrow trials (constituting one retrieval block) and alternated between two environmental contexts (stone room: Fig. 1g, wooden room: Fig. 1h) each of which contained three halos with unique colors and positions that differed between the two contexts and were fixed over the duration of the task.”

As suggested, we now also include data showing improvements in accuracy between the first and last block of the task, where accuracy was determined based on whether the button press was correct (≤ 0.75 m from the target) or incorrect (> 0.75 m from the target), similar to how distance error was determined. Consistent with

our results of reduced mean spatial error signifying improvement in memory, we find a significant increase in accuracy across participants in the last compared to the first block (Fig. 2b). We cannot look at improvements in accuracy within a block unfortunately, since there is only one trial per halo within each block (6 total, wooden: 3, stone: 3).

We have included this new analysis and result in the revised version of our manuscript, see Results section (pg. 5, lines 121-126):

“Accuracy (calculated as % correct) was also computed during the same retrieval blocks based on a 0.75 meter (m) radial distance threshold (from the center of the halo), which was used to provide visual feedback to the participant (“correct” or “incorrect”). Mean accuracy was 65% (\pm 8.5% s.e.m.) across participants and improved significantly during the last compared to the first retrieval block (see Methods for further details, $p = 0.036$, Fig. 2b-c).”

Fig. 2: Memory performance during the ambulatory spatial navigation task. (a-b) Difference in (a) mean memory performance (error) and (b) accuracy (% correct) between the first (trials before learning criteria was met) and last retrieval block. Lines show data from individual participants. * = $p < 0.05$. (c) Mean error was measured for each of the six participants (P1-6, colored lines, $n_{\text{halos}} = 3$ halos) by calculating the average distance between recalled and correct halo locations across trials. The 1st retrieval block included a variable number of trials for P1-5 (15-30 trials) depending on when a learning criterion (error for 15 consecutive trials < 1.5 m) was reached. P6 did not show learning during retrieval block #1 and thus was manually advanced to subsequent retrieval blocks. Mean performance across participants is also shown (black line, $n_{\text{participants}} = 5$ participants, P1-5; P6 excluded due to inability to meet learning criterion). The total number of retrieval blocks varied across participants (5-10 blocks). (d) Top-down view of an example participant's walking trajectory collapsed over all encoding, retrieval, and arrow trials in the stone (top) and wooden (bottom) contexts. Halo colors and positions are indicated in each of the two environments.

C2: *The analyses are all conducted by averaging the data across 19 electrodes from all participants. Clearly, in every patient there are only one or two contacts, which has motivated this approach. But it would be helpful to know what these effects look like at the subject level. Could it be that only 1 or 2 subjects are driving this overall effect, or is this consistent across subjects. This will likely be similar to the original results that are presented, since there are only a small number of electrodes per subject, but it would be helpful nonetheless to see the analyses averaged across participants rather than electrodes.*

Response: We thank the reviewer for this suggestion and have now included analyses at the subject-level. We find similar results when looking at analyses averaged across participants rather than electrodes in both main effects (memory retrieval and boundary modulation; Extended Data Figure 1 & 7). Additionally, in response to reviewer #1 and #2, we have illustrated the effect in channels in each participant and also used a leave-one-out approach in which channels from each participant are iteratively thrown out on each run. We believe that these control analyses strengthen the main results and highlight that they are not driven by individual participants.

Extended Data Figure 1b-d: Supplementary analyses for successful memory modulation of theta. (b-c) MTL theta bandpower in the 0.5 s prior to recall (button press) for correct and incorrect trials **(b)** illustrated across channels in individual participants and **(c)** when averaging over individual channels per participant ($n_{\text{participants}} = 6$). **(d)** Correct vs. incorrect increases in MTL theta bandpower remained significant when using a leave-one-out approach where each of the 6 participants were excluded one at a time. Crosses (+) represent the mean norm'd (normalized) bandpower across all trials for **(a,b,e)** individual channels and **(c)** across channels in a participant where separate colors correspond to individual participants. * = $p < 0.05$.

We have added these results to the manuscript (pg. 6, lines 144-149):

“However, this finding... was numerically present across participants (Extended Data Fig. 1b), persisted when averaging over channels for each participant ($p < 0.001$, $n_{\text{participants}} = 6$, Extended Data Fig. 1c), and remained after a leave-one-out approach when each participant’s data was excluded one at a time (Extended Data Fig. 1d), suggesting that findings were not driven by individual subjects.”

Extended Data Figure 7b,c,e: Boundary versus inner distance thresholds and control analyses. (b-c) Mean \pm s.e.m. normalized (norm'd) theta bandpower (4-6 Hz) during arrow trials, after excluding the 0.5 m prior to participant arrival to a visible arrow on each trial **(b)** in MTL channels for each participant for boundary and inner positions using a 1.2 m threshold, and **(c)** when averaging across each participant’s individual channels ($n_{\text{participants}} = 6$). Crosses (+) represent mean power averaged over MTL channels for each participant. *** = $p < 0.001$. **(e)** Leave-one-out approach with analysis run after excluding one participant each time and associated statistic shown.

and (pg. 12, lines 300-306):

“Lastly, boundary-related increases in theta power were not dependent on the specific boundary vs. inner 1.2 m cutoff (Extended Data Fig. 7a) and occurred within individual participants (Extended Data Fig. 7b), when averaged over individual channels of participants ($p < 0.001$, $n_{\text{participants}} = 6$, Extended Data Fig. 7c), and persisted during a leave-one-out approach when each participant’s data was excluded one at a time (Extended Data Fig. 7e), suggesting that findings were consistent across participants and not driven by individual subjects.”

C3: The comparison is made between correct trials to the period of visible feedback. But it seems like there is a natural and perfect build in control in the experimental design, which is the arrow trials. How does theta power look when approaching the arrow trials as compared to the approach during correct retrieval trials.

Response: We thank the reviewer for this insightful comment. In response, we have performed the suggested new analysis, and found no significant increase in theta bandpower during arrow trials. We have further included this new analysis in the main text and of the revised manuscript, and updated Figure 3 (see panels d,h) to illustrate this result:

Figure 3: MTL theta bandpower increased during correct compared to incorrect retrieval, visible halo, or arrow trials. (a-d) Mean (\pm standard error of the mean [s.e.m.]) normalized (norm'd) power across MTL channels ($n_{\text{channels}} = 19$) for frequencies 3-120 Hz during the 0.5 s period prior to either the button press during (a) correct or (b) incorrect recall during retrieval trials, or (c) arrival at visible halos during feedback, or (d) arrival at arrows during arrow trials. MTL theta bandpower significantly increased during correct but not incorrect retrieval, visible halo, or arrow trials. Horizontal pink bar indicates significant power increase/decrease ($p < 0.05$, corrected using false discovery rate [FDR]^{19,20}). (e-h) Top-down view of norm'd theta bandpower (6-8 Hz) in an example MTL channel (participant 5) averaged across all samples during retrieval (e) correct, (f) incorrect, (g) visible halo, and (h) arrow trials. Note, halos were not visible during correct or incorrect retrieval nor during arrow trials. (e-f) Colored circles reflect all recalled locations during retrieval for correct and incorrect trials. (g) Colored circles reflect locations of halos during visible feedback. (i) Mean (\pm s.e.m.) norm'd theta bandpower across MTL channels ($n_{\text{channels}} = 19$). Vertical gray dotted line (time = 0) indicates the moment (button press) when participants arrived at the remembered halo position (correct/incorrect) during retrieval trials, visible halo (visible, blue) during feedback, or arrow (orange) during arrow trials. (j) Mean (\pm s.e.m.) norm'd theta bandpower across MTL channels ($n_{\text{channels}} = 19$) during correct/incorrect trials, visible feedback periods, and arrow trials. MTL theta bandpower significantly increased during the 0.5 s prior to recall for correct compared to incorrect, visible (feedback), and arrow trials. Crosses (+) represent the mean norm'd bandpower across all trials for an individual channel with each color corresponding to channels from a single participant. * = $p < 0.05$, ** = $p < 0.01$, *** = $p < 0.0001$, FDR corrected.

We have also summarized these results in the manuscript (pg. 6, lines 136-144):

“During this time period, MTL oscillatory power significantly increased only at theta (6-8 Hz) frequencies (6-8 Hz: all individual frequencies $p < 0.05$, after correcting for multiple comparisons using the false discovery rate [FDR]^{17,18}, $n_{\text{channels}} = 19$, Fig. 3a-d). Specifically, this theta (6-8 Hz) bandpower was significantly elevated during

correct but not incorrect retrieval trials, arrival at visible halos during feedback, or arrival at arrows during arrow trials (Fig. 3i) and this increase occurred during the 0.5 s prior to recall (Fig. 3j; correct vs. incorrect, $p = 0.003$; correct vs. visible halo, $p < 0.001$; correct vs. arrow, $p = 0.047$; incorrect vs. visible halo, $p = 0.190$; arrow vs. incorrect, $p = 0.280$; arrow vs. visible, $p = 0.022$; FDR corrected, $n_{channels} = 19$, Supplemental video 3).”

C4: The analysis of spatial position suggests that there is an increase in MTL theta activity near meaningful positions (e.g., non-targets). But is there also an overall spatial modulation of theta. From the maps provided, it appears that the bouts of theta may be somewhat structured and regular as one simply navigates around the environment. Is this the case? Again, the arrow trials could be an opportunity to examine this.

Response: The reviewer raises an interesting point about how theta is spatially modulated during navigation throughout the environment. To address this question, we performed an additional analysis in order to test whether there are spatial or memory representations tied to the room positions during the non-memory-related (directed navigation) portion of the task (i.e., during arrow trials). We applied the same approach that we used to examine non-target and non-visible halo representations during retrieval trials (as shown in Fig. 4), in order to examine whether theta was modulated by the (non-visible) halo positions in the environment during the arrow trials. Specifically, we compared theta power in instances when participants were close relative to far away (e.g. $<$ vs. $>$ 2m, empirically determined based on a multiple-threshold analysis, Fig. 4a) from (context-specific) non-visible halo positions during arrow trials. Interestingly, we found no significant change in theta power related to the proximity of (non-target) halo locations during arrow trials (Extended Data Fig. 6a). This suggests that during non-memory-related navigation (arrow trials) the predominant spatial representation is proximity to boundaries (Fig. 5) and that the modulation of theta power by halo locations, present during retrieval trials (Fig. 4), is abolished during arrow trials. Since arrow and retrieval trials alternate over the course of the entire task, this suggests that spatial/memory representations are dynamically and rapidly changing based on momentary task goals. We have added these analysis and results to the revised version of our manuscript, together with a brief discussion of these new findings (pgs. 9-10, lines 233-236):

Extended Data Figure 6a: Close vs. far control analyses. Mean normalized (norm'd) theta bandpower in positions close to ($<$ 2 m) versus far away ($>$ 2 m) from non-target halo centers was not significantly different when computed during arrow trials. Crosses (+) represent the mean norm'd bandpower for each channel during retrieval trials in each context, excluding the last 0.5 s prior to recall of a target halo and are colored according to participant.

“Interestingly, we did not observe such a pattern of results during arrow trials (Extended Data Fig. 6a), indicating that proximity to (non-target) halo locations modulated theta power only during memory retrieval but not when participants walked towards a visible cue.”

C5: The spatial map appears to change between contexts. But if this map is determined largely by non-targets, then does the map also change within contexts but between blocks, when the location of the learned halos has changed?

Response: The reviewer raises an interesting question about how theta modulation over spatial positions changed across different conditions. We have investigated whether there is any spatial modulation of theta in a number of conditions. First, we quantified the effect demonstrating that theta bandpower is significantly elevated in close relative to far positions in each context (pg. 10, lines 236-240):

“The spatial distribution of theta (6-8 Hz) bandpower increases was specific to relevant positions within each context separately (stone: $p = 0.012$; wooden: $p = 0.041$, $n_{channels} = 19$, Extended Data Fig. 6b), suggesting that MTL spatial representations can remap based on the perceived environment (see example channel showing theta activity in the stone (Fig. 4c) and wooden (Fig. 4d) context).”

Next, we examined whether this spatial modulation by non-target halos changed across blocks. It is important to clarify (and as noted above) that halo positions and colors are fixed over the entire task with 3 uniquely colored halos present in each context (i.e., the halo positions within a context did not change throughout different task blocks).

As such, we would not expect the spatial map to change within contexts across different blocks, since the locations of the learned halos are fixed. However, we examined how the spatial map (e.g. increased theta bandpower during close relative far regions) changed over time (over blocks, when performance continued to improve). We applied a median split to the total number of blocks completed for each participant and found that - while patterns of results appear (numerically) similar between earlier and later task blocks - the difference in theta bandpower in close relative to far regions was significant in later blocks but not earlier blocks, likely associated with when participants had formed robust spatial maps. As we believe that this might be of interest to readers, we have added these new analyses and results to the revised version of our manuscript (pg. 10, lines 240-242):

“Furthermore, the difference in theta bandpower between “close” and “far” positions was strongest in later blocks, likely after the participants developed robust spatial maps (Extended Data Fig. 6c).”

Extended Data Figure 6b-c: Close vs. far control analyses. (b-c) Mean normalized (norm'd) theta bandpower in positions close to (< 2 m) versus far away (> 2 m) from non-target halo centers **(b)** was significantly increased within each context, and **(c)** significantly increased during later blocks but not in earlier blocks (one-sided permutation test). Crosses (+) represent the mean norm'd bandpower for each channel during retrieval trials in each context, excluding the last 0.5 s prior to recall of a target halo and are colored according to participant. * = $p < 0.05$.

C6: *The boundary analysis is performed during arrow trials and shows elevated theta near boundaries. Clearly, these trials involve moving to the arrows that are located at the boundary. So is this a boundary effect (and similarly a memory effect for inner regions), or is the increase in theta simply related to the goal of the trial. Interestingly, the analysis excludes the .5 s before arriving at the arrow. And this effect is also seen when analyzing the entire task. Could boundary activity be related to the pictures on walls, which could serve as memory cues?*

Response: We thank the reviewer for raising these important points. In order to control for the potential confound of differing amounts of visual information available as participants approach the walls, we have analyzed whether theta bandpower is elevated in boundary relative to inner positions in two sub-conditions of the arrow trials (again excluding the 0.5 m prior to arrival at the arrow, as we have done in the main analysis): when participants are (1) moving towards the nearest boundary, which is associated with the approach to the arrow/trial goal and (2) when participants are moving away from the nearest boundary, which is associated with less visual wall information and trial goal-orientation. We find that, in both of these conditions, boundary positions are associated with increased theta bandpower relative to inner positions, suggesting that this effect is not driven by the pictures on the wall and/or trial target goals. However, we still cannot completely rule out the possibility that visual wall information, previously experienced arrow cues, and the specific trial arrow goal may contribute to this effect.

Extended Data Figure 7d: Boundary versus inner distance thresholds and control analyses. Mean \pm s.e.m. normalized (norm'd) theta bandpower (4-6 Hz) during arrow trials, after excluding the 0.5 m prior to participant arrival to a visible arrow on each trial for instances when participants are moving towards (left) or away (right) from boundaries. Crosses (+) represent mean power averaged over MTL channels for each participant. *** = $p < 0.001$.

As such we have added this as a discussion point (pgs. 11-12, lines 284-289):

“To examine whether encoding of visual information on walls was contributing to boundary-modulation of theta power, we examined theta bandpower fluctuations in two separate conditions: when participants were moving towards the (nearest) wall and when participants were moving away from the (nearest) wall. We observed that boundary-modulation of theta bandpower persisted in both conditions (towards: $p < 0.001$, away: $p < 0.001$, $n_{channels} = 19$, Extended Data Fig. 7d).”

and (pg. 14, lines 359-362):

“Importantly, boundary modulation of theta bandpower persisted in conditions both when participants approached and moved away from the wall, suggesting that the visual information available when facing a wall was not driving this spatial representation.”

C7: *In the discussion, they state that MTL theta activity is only elevated right before retrieval during successful recall, but this is also true for incorrect trials (compared to feedback at least). It would also be interesting to see what happens when the subjects are first given the target cue and still have not moved yet. Are there also increases in theta during this immobile period before they make a movement and are these increases now in low theta. This might suggest that the bands can be dissociated depending on whether movement is involved.*

Response: We agree with the reviewer in that theta power appears to be (numerically) elevated relative to the visible feedback condition also for incorrect trials. However, please note that after multiple comparisons correction (now also accounting for the analyses of arrow trials added during the revision process) this effect is not statistically significant ($p = 0.176$, FDR corrected).

As the reviewer suggests, we have included analyses after cue onset, prior to movement to the target halo location. During this immobile period (all time points leading up to the first onset of movement) we did not see a difference between correct and incorrect trials in either 6-8 Hz theta nor in a lower frequency band (3-6 Hz). Specifically, we examined theta bandpower difference between correct and incorrect trials during all timepoints after cue presentation until the first onset of movement on each trial. Crosses (+) represent mean power averaged over MTL channels for each participant.

However, as we have highlighted in response to reviewer # 1 (C1), it is important to note that our experimental task was largely self-paced and participants were freely moving, and thus we cannot fully determine at what exact point in time participants retrieved the memory of the actual target location. In fact, participants could freely decide at what point in time after cue onset they would start to move, and some participants may have had multiple movement onset periods (for instance, a participant may have started to walk immediately after cue onset, then stop again, think about the correct target location and perhaps re-consider their initial choice, before walking to the actual recalled position). As a result, we expected that quantifying theta bandpower over all timepoints during immobility after cue onset may not necessarily capture the instant when participants initiated memory retrieval (consider, for example, that those participants who began most trials with immediate movement initiated retrieval while moving).

To provide better temporal resolution and also in response to reviewer # 1 (C1), we thus examined theta bandpower fluctuations across timepoints following cue onset. As shown in Extended Data Fig. 4a, after cue presentation, theta power initially appears to be similar between correct and incorrect trials, but a significant difference can be detected around 1.5 seconds after cue onset. However, for the same reasons detailed, we still expect the time window immediately after cue onset to contain substantial 'noise' (i.e., variability between participants and trials), as this time window likely contains data from some trials when participants retrieved the correct memory immediately and other trials when they retrieved the correct memory at a later point in time during a given trial. Thus, we hypothesized that another interesting period most likely captures a time window critical for memory retrieval: the last movement onset before button press (i.e., the time when participants likely made their final decision as to where to go and press the button).

We thus performed an additional analysis to test whether low-frequency (around 3-6 Hz) and high-frequency (around 6-8 Hz) theta oscillations exhibit differences in power between correct and incorrect trials after cue presentation and before movement onset. Indeed, in line with the reviewer's hypothesis, we found that low-frequency theta power was increased already around 0.5 after cue onset, as well as around movement onset Extended Data Fig. 4c-d.

Together, these results suggest that, regarding differences in theta power between correct and incorrect trials, the earliest effects directly after cue onset and before movement onset are present in the low-frequency theta band (around 3-6 Hz), while later effects around 1.5 sec after cue onset and before button press appear to be more prevalent in the higher theta range (around 6-8 Hz).

Extended Data Figure 4: Successful memory modulation of theta bandpower after cue and movement onset. Mean (\pm s.e.m.) norm'd (normalized) theta bandpower across MTL channels ($n_{\text{channels}} = 19$) (a,c) after cue presentation and (b,d) around movement onset (last movement onset prior to button press in each retrieval trial) in (a-b) 6-8 Hz and (c-d) 3-6 Hz bandpower. Note, halos were not visible during correct (green) or incorrect (red) retrieval trials. Gray bar indicates timepoints where $p < 0.05$ (one-sided permutation test at each time point, representing 4 ms steps at 250 Hz sampling rate).

We have added a discussion of these new findings to the manuscript (pg. 8, lines 185-199):

“We also explored successful memory-related theta bandpower changes during other time periods (Extended Data Fig. 4) and found that while theta bandpower (6-8 Hz) initially appeared to be similar between correct and incorrect trials after initial cue presentation during retrieval trials, a difference was detected around 1.5 seconds after cue onset (Extended Data Fig. 4a). Previous work has suggested that, within a broader theta frequency range, low frequency theta oscillations (e.g. type II theta) are related to episodic memory and higher frequency theta oscillations (e.g. type I theta) are movement-related^{1,2}. As such, we also investigated differences between correct and incorrect trials in low frequency oscillations (3-6 Hz) after cue presentation, and before the onset of movement. Since participants often had multiple movement onset periods within a single trial, we specifically examined the last movement onset before button press, which we hypothesized would better capture the time window when participants initiated memory retrieval to determine their final recalled position for the indicated target halo on any given trial. Indeed, we found that low-frequency theta power was increased already around 0.5 s after cue onset (Extended Data Fig. 4c) and prior to movement onset during retrieval trials (Extended Data Fig. 4d).”

and (pg. 13, lines 330-335):

“In line with this hypothesis, we found elevated low-frequency theta bandpower (e.g. memory-related type II theta) in two time windows associated with less movement: around (1) 0.5 s after cue presentation and (2) 0.5 s prior to movement onset, while there was elevated higher-frequency theta bandpower (e.g. movement-related type I theta) in two time windows associated with more movement: (1) around 1.5 s after cue presentation and (2) in the 0.5 s prior to recall^{1,2}.”

C8: *The results suggest that there are no changes in theta prevalence. In addition, there is higher prevalence for other higher frequencies. What does the analysis of theta prevalence reveal during only the .5 s retrieval time period?*

Response: The reviewer is correct in that we found no changes in theta prevalence across conditions (correct vs. incorrect trials). As suggested by the reviewer, we have also performed this analysis focusing only on the 0.5 s time window prior to recall, and also found no differences in theta prevalence between correct versus incorrect trials. We have now added these results to the supplementary materials and results section (Extended Data Fig. 5; pgs. 8-9, lines 204-212).

“We found that MTL theta bandpower increases did occur in transient bouts and occurred at similar rates compared to previous studies^{15,16}, however, the prevalence of these bouts did not significantly differ between task conditions either during the entire retrieval trial period (retrieval vs. arrow vs. visible halo trials, $p > 0.05$; correct vs. incorrect, $p > 0.05$; across all individual frequencies between 3-25 Hz, $n_{channels} = 19$, Extended Data Fig. 5a-c) or the last 0.5 s prior to recall (retrieval vs. arrow vs. visible halo trials, $p < 0.05$; correct vs. incorrect, $p > 0.05$; across all individual frequencies between 3-25 Hz, $n_{channels} = 19$, Extended Data Fig. 5e-f), suggesting successful memory retrieval results in increased MTL theta bandpower in the absence of changes in its prevalence.”

Extended Data Figure 5: Oscillatory prevalence across task conditions. Mean (\pm standard error of the mean [s.e.m.]) oscillatory prevalence across frequencies (3 – 25 Hz) and MTL channels ($n_{channels} = 19$) during **(a)** the entire task, **(b)** retrieval, arrow, and visible halo trials, **(c)** correct and incorrect retrieval trials, **(d)** boundary (< 1.2 m from room walls) and inner (> 1.2 m from room walls) room positions, **(e)** retrieval, arrow, and visible halo trials (500 ms before recall), and **(f)** correct and incorrect retrieval trials (0.5 s before recall). No significant differences were found in oscillatory prevalence across frequencies (3 – 25 Hz) for any conditions shown **(b-f)**. $p > 0.05$ for all frequencies, FDR corrected.

C9: *It is not clear how much time elapses between encoding the halos and the retrieval period. Is there a distraction period or a distraction task?*

Response: We thank the reviewer for asking this question, which gives us the opportunity to further clarify details of the task that might be of interest to readers. In the first task block, participants were presented with a halo appearing in its location in the environment and were instructed to navigate to the visible halo and learn its position (encoding trial). Next, participants completed an arrow trial in which an arrow appeared in a random position along the perimeter of the environment and participants were instructed to navigate towards it, followed by the next encoding trial (encoding of another halo location). After encoding each halo in the environment one time and alternating with arrow trials (thus completing the encoding phase), the participants began the retrieval period in the same environmental context. While there was no distractor task between encoding and retrieval trials, there was an arrow trial interleaved between each of the encoding trials and also between each of the retrieval trials (thus, technically, arrow trials could be interpreted as a kind of ‘distractor’ task interleaved between trials). For each arrow trial, an arrow appeared in an unpredictable and random

location along the perimeter of the room environment. Thus, each arrow trial began with the participant visually searching for the newly appeared arrow, then navigating to it. Since the location of the arrow on each arrow trial was randomized, arrow trials lacked any memory component and further ensured that the participant began each retrieval trial from a new position in the environment that was necessarily displaced from where they ended their previous retrieval trial. Furthermore, given the spatial nature of this memory task and structured and interleaving retrieval/arrow trial format, participants could only retrieve one halo at a time before advancing to an arrow trial.

As our experimental task was largely self-paced, the time between encoding a particular halo and retrieving the same halo's position varied between participants. The delay between the encoding and first retrieval trial for a particular halo was ~2 minutes, depending on the amount of time needed for a participant to navigate to the two other halos in the environment (encoding trials for other halos) each interleaved with an arrow trial.

--

Additionally, we would like to highlight a modification to the manuscript regarding the Morlet wavelet parameter used to compute oscillatory power as we would like to raise this point for full transparency. Please refer to our detailed discussion in response to Reviewer #2, C14.

References

1. Jacobs J. Hippocampal theta oscillations are slower in humans than in rodents: implications for models of spatial navigation and memory. *Philos Trans R Soc B Biol Sci.* 2014;369(1635). doi:10.1098/RSTB.2013.0304
2. Stangl M, Maoz SL, Suthana N, Sciences B. *Invasive Electrophysiological Recordings from Humans during Navigation.* Second Edi. Elsevier; 2024. doi:10.1016/B978-0-12-820480-1.00017-6
3. Liu J, Zhang H, Yu T, et al. Transformative neural representations support long-term episodic memory. *Sci Adv.* 2021;7(41). doi:10.1126/SCIADV.ABG9715/SUPPL_FILE/SCIADV.ABG9715_SM.PDF
4. Lifanov J, Griffiths BJ, Linde-Domingo J, et al. Reconstructing spatio-temporal trajectories of visual object memories in the human brain. *bioRxiv.* 2022;2.
5. Stangl M, Topalovic U, Inman CS, et al. Boundary-anchored neural mechanisms of location-encoding for self and others. *Nature.* 2021;589(7842):420-425. doi:10.1038/s41586-020-03073-y
6. Lee SA, Miller JF, Watrous AJ, et al. Electrophysiological Signatures of Spatial Boundaries in the Human Subiculum. *J Neurosci.* 2018;38(13):3265-3272. doi:10.1523/JNEUROSCI.3216-17.2018
7. Solstad T, Boccara CN, Kropff E, Moser MB, Moser EI. Representation of geometric borders in the entorhinal cortex. *Science (80-).* 2008;322(5909):1865-1868. doi:10.1126/SCIENCE.1166466/SUPPL_FILE/SOLSTAD.SOM.PDF
8. Lever C, Burton S, Jeewajee A, O'Keefe J, Burgess N. Boundary Vector Cells in the Subiculum of the Hippocampal Formation. *J Neurosci.* 2009;29(31):9771-9777. doi:10.1523/JNEUROSCI.1319-09.2009
9. Høydal ØA, Skytøen ER, Andersson SO, Moser M-B, Moser EI. Object-vector coding in the medial entorhinal cortex. *Nature.* 2019;568(7752):400-404. doi:10.1038/s41586-019-1077-7
10. Barry C, Lever C, Hayman R, et al. The boundary vector cell model of place cell firing and spatial memory. *Rev Neurosci.* 2006;17(1-2):71-97. doi:10.1515/REVNEURO.2006.17.1-2.71
11. Dupret D, O'Neill J, Pleydell-Bouverie B, Csicsvari J. The reorganization and reactivation of hippocampal maps predict spatial memory performance. *Nat Neurosci.* 2010;13(8):995-1002.

doi:10.1038/NN.2599

12. Hollup SA, Molden S, Donnett JG, Moser MB, Moser EI. Accumulation of hippocampal place fields at the goal location in an annular watermaze task. *J Neurosci*. 2001;21(5):1635-1644. doi:10.1523/JNEUROSCI.21-05-01635.2001
13. Bourboulou R, Marti G, Michon FX, et al. Dynamic control of hippocampal spatial coding resolution by local visual cues. *Elife*. 2019;8. doi:10.7554/ELIFE.44487
14. Poulter S, Lee SA, Dachtler J, Wills TJ, Lever C. Vector Trace cells in the Subiculum of the Hippocampal formation. *Nat Neurosci*. 2021;24(2):266. doi:10.1038/S41593-020-00761-W
15. Topalovic U, Aghajan Z, Villaroman D, et al. Wireless Programmable Recording and Stimulation of Deep Brain Activity in Freely Moving Humans. 2020. doi:10.1101/2020.02.12.946434
16. Topalovic U, Barclay S, Ling C, et al. A wearable platform for closed-loop stimulation and recording of single-neuron and local field potential activity in freely moving humans. *Nat Neurosci* 2023. 2023;8:1-11. doi:10.1038/s41593-023-01260-4
17. Benjamini Y, Yekutieli D. The control of the false discovery rate in multiple testing under dependency. <https://doi.org/10.1214/aos/1013699998>. 2001;29(4):1165-1188. doi:10.1214/AOS/1013699998
18. Benjamini Y, Hochberg Y. Controlling the False Discovery Rate: A Practical and Powerful Approach to Multiple Testing. *J R Stat Soc Ser B*. 1995;57(1):289-300. doi:10.1111/J.2517-6161.1995.TB02031.X

Reviewers' Comments:

Reviewer #1:

Remarks to the Author:

The authors have done a thorough job in considering and formulating a response to the previous round of criticisms. The subject level data especially is a nice addition to give confidence to the results given the very small number of subjects. They have clearly engaged seriously with the reviewer comments overall. I think the manuscript has been substantially improved.

I think the lower frequency theta result at time of cue/onset of first movement is more convincing, mainly because you know exactly what the subject is doing when cued. I think this should be emphasized more in the analysis, perhaps even presented as the primary result. At least, it should be more fully investigated including the linear models testing the other variables, etc. This frequency range is also closer to the boundary theta, which is quite interesting.

I understand that the free ranging paradigm makes the analysis difficult. In future, adding more trials for each location in order to fit a learning model to the behavior would probably help to figure out appropriate time windows for memory retrieval. Doesn't the type 1 vs type 2 theta model suggest that the higher frequency theta activity at time of button press is not necessarily related to retrieval? The interpretation was a little confusing. How does the boundary theta, which is at 4–6 Hz, match up with this finding? I think the boundary controls are pretty good, with the distance variable and the linear model. I think the interpretation would benefit from some explanation about why this boundary theta occurs during arrow but not memory behavior trials though.

I agree a GLM is a nice way to analyze the navigation activity following cue. In extended data 3, I'm a little confused about the coefficients though. Why does "distance error" and "correct/incorrect" not have the opposite sign on the coefficients? Isn't a large distance error associated with incorrect trials which would have a lower theta power value? I think I just don't understand how the variables were input in the GLM.

Is this frequency band significant in the full spectral analysis (FDR corrected across all frequencies) that was originally used to filter the frequency range that was then used for the subsequent analyses? Adding this same type of figure for cue would add confidence to this result, which to me is more convincing than the .5 sec prior to button press.

Does the comparison between correct and arrow trials survive FDR correction for this time period (figure 3). The x axis label of "time relative to halo recall" is a little misleading. In general, I think calling the button press the "instance of recall" is confusing, given the variability in time after cue. The Liu and Lifanov papers are not really about spatial navigation. The concern about the time window before retrieval is more about the behavior than about whether theta power differences may occur over short periods of time. Perhaps the authors could cite other spatial navigation experiments.

Can the authors describe the normalization step in a little more detail? Was this carried out across all time samples in a long time series?

I am overall impressed with how much work the authors have done. These data are complex to analyze. I think the addition of the lower frequency theta result when behavior is controlled is especially interesting.

Reviewer #2:

Remarks to the Author:

I thank the authors for their detailed and considered response to my previous comments. I believe the

manuscript is much improved. I have only one final comment regarding the statistical analyses, and two minor comments about presentation and terminology. I am happy to leave these changes to the authors discretion - I do not need to see the manuscript again.

Comment 1: The authors still present their statistical analyses across $n=19$ channels first, before going on (in some cases) to show that these findings are also significant when analysed correctly (i.e. either using data averaged across channels in each patient, to give $n=6$; or using a linear mixed effects model with patient ID as a random effect). I do not understand this approach, and I think it makes the paper appear statistically naive. All reviewers shared this concern in some form, as far as I can tell. So given that it doesn't seem to change the findings, why not just present the correct statistical analyses and remove all analyses across $n=19$ channels from the manuscript completely? At the very least, the authors should highlight if any of their findings are not observed across $n=6$ patients (i.e. for close v far in Fig 4a; across environments in Extended Figure 6b; boundary v inner in Fig 5b; towards v away from the boundary in Extended Figure 7d; and correct v incorrect excluding boundary positions in Extended Figure 1e)

Comment 2: I think 'bandpower' should be 'band power' throughout

Comment 3: On page 7, when describing the linear mixed effects model, it would be useful to clarify whether 'distance to recall' and 'distance to button press' are truly referring to spatial variables (i.e. in m) or whether 'duration' would be more accurate (i.e. in s)

Reviewer #3:

Remarks to the Author:

The authors have presented a comprehensive reply to the original reviews of their manuscript. They have sufficiently addressed the concerns and presented several new and control analyses that I feel have strengthened the overall conclusions. This is a good study and will be valuable for the field.

Dear Reviewers,

We would like to thank you for your generous and constructive feedback on our revised manuscript, “Dynamic neural representations of memory and space in freely moving humans”. Each reviewer made suggestions that would greatly strengthen the manuscript. We have now incorporated each of these suggestions into a revised manuscript and detail our changes below (in blue text) made in response to each point raised by the reviewers.

Reviewer Comments:

Reviewer #1 [R1] (Remarks to Author): *The authors have done a thorough job in considering and formulating a response to the previous round of criticisms. The subject level data especially is a nice addition to give confidence to the results given the very small number of subjects. They have clearly engaged seriously with the reviewer comments overall. I think the manuscript has been substantially improved.*

Response: We thank the reviewer for their time and effort to review our manuscript, and for their positive feedback. Please find our responses to each additional comment below.

Comment 1 [C1]: *I think the lower frequency theta result at time of cue/onset of first movement is more convincing, mainly because you know exactly what the subject is doing when cued. I think this should be emphasized more in the analysis, perhaps even presented as the primary result. At least, it should be more fully investigated including the linear models testing the other variables, etc. This frequency range is also closer to the boundary theta, which is quite interesting.*

Response: We thank the reviewer for this comment and agree that the lower frequency theta band power effects after cue onset are interesting and important. Based on the reviewer’s suggestion, we have added “distance from cue” as a variable in a linear mixed model in which it comes out as the only significant predictor of low frequency theta band power during the 0.5 s after cue (Supplementary Fig. 3c). We now include these results in the manuscript:

Supplementary Fig. 3: Simultaneous impact of multiple variables on theta power. Linear mixed-effects models were calculated to predict each participant’s normalized theta band power timeseries by a range of predictor variables that were fixed effects with samples blocked according to channel identity. Asterisks denote a significant impact of a variable’s beta weight on theta power, * = $p < 0.05$ for $n_{\text{participants}} = 6$. Error bars show the s.e.m. of the beta weights across participants. Models were implemented for (a) the last 0.5 s of retrieval (when no halos were visible), (b) the entire retrieval period (when no halos were visible), (c) the first 0.5 s of retrieval (after cue onset), and (d) the arrow search period. Movement speed, angular velocity, distance to recalled position (distance in meters to the button press location), distance from cue (distance in meters from the retrieval cue), distance to boundary (distance in meters to the nearest wall), distance to arrow (distance in meters to the target arrow), and distance error (distance in meters between the recalled position and target location) were included as continuous variables. A significant negative impact of a distance variable (i.e., distance to recalled position, distance from cue, distance error, distance to arrow, and distance to boundary) reflects increased theta band power for shorter distances. Whether a trial was correct or incorrect (correct/incorrect) was a binary variable and movement direction was a categorical variable comprised of 12 possible binned movement directions (with the mean beta coefficient over all 12 directional bins depicted). A significant positive impact of the correct/incorrect variable signifies higher theta band power values observed for correct compared to incorrect trials. Crosses (+) represent the variable impact (beta) colored according to participant. Source data are provided as a Source Data file.

Additionally, in line with the reviewer’s suggestion, results of a detailed analysis of low frequency theta band power (3-6 Hz) after cue and movement onset are also shown in Supplementary Fig. 4c-f:

Supplementary Fig. 4: Successful memory modulation of theta band power after cue and movement onset. Mean (\pm s.e.m.) norm'd (normalized) theta band power across MTL channels ($n_{\text{channels}} = 19$) (a,c) after cue presentation and (b,d) around movement onset (last movement onset prior to button press in each retrieval trial) in (a-b) 6-8 Hz and (c-d) 3-6 Hz band power. Note, halos were not visible during correct (green) or incorrect (red) retrieval trials. Gray bar indicates timepoints where $p < 0.05$ (one-sided permutation test at each time point, representing 4 ms steps at 250 Hz sampling rate). (e) Mean (\pm s.e.m.) norm'd theta band power across MTL channels ($n_{\text{channels}} = 19$). Vertical gray dotted line (time = 0) indicates trial onset when the instructional cue was presented to navigate to the target halo during retrieval trials, visible halo (visible, blue) during feedback, or arrow (orange) during arrow trials. (f) Mean (\pm s.e.m.) norm'd theta band power across MTL channels ($n_{\text{channels}} = 19$) during 0.25-0.5 s after cue for correct/incorrect trials, visible feedback periods, and arrow trials. MTL theta band power significantly increased during this period after cue for correct compared to incorrect and visible (feedback) trials. Crosses (+) represent the mean norm'd band power across all trials for an individual channel with each color corresponding to channels from a single participant. * = $p < 0.05$, ** = $p < 0.01$, *** = $p < 0.0001$, FDR corrected. Source data are provided as a Source Data file.

C2: I understand that the free ranging paradigm makes the analysis difficult. In future, adding more trials for each location in order to fit a learning model to the behavior would probably help to figure out appropriate time windows for memory retrieval. Doesn't the type 1 vs type 2 theta model suggest that the higher frequency theta activity at time of button press is not necessarily related to retrieval? The interpretation was a little confusing. How does the boundary theta, which is at 4–6 Hz, match up with this finding? I think the boundary controls are pretty good, with the distance variable and the linear model. I think the interpretation would benefit from some explanation about why this boundary theta occurs during arrow but not memory behavior trials though.

Response: We thank the reviewer for raising this important point. As the reviewer suggests, we have included a plausible explanation for the presence of boundary-related theta during arrow trials but not during memory-related behavior trials (Discussion, pg. 15):

“This spatial remapping of oscillatory activity based on the behavioral goal (memory recall versus cue-driven navigation) suggests that MTL theta band power can dynamically reflect multiple spatial and mnemonic variables in an “on/off” and flexible manner, where the presence of specific neural representations depends on the immediate cognitive requirements of the task at hand. For instance, boundary-related increases in theta power could be present exclusively during periods when the proximity to the spatial boundary is relevant (as seen in arrow trials, where arrows are positioned at room boundaries). On the other hand, during memory retrieval phases when spatial boundaries are momentarily less relevant, these boundary-related representations might be notably absent... Future studies will be needed to better understand the exact role of low- (type II) versus high- (type I) frequency theta oscillations in this context.”

C3: I agree a GLM is a nice way to analyze the navigation activity following cue. In extended data 3, I'm a little confused about the coefficients though. Why does “distance error” and “correct/incorrect” not have the opposite sign on the coefficients? Isn't a large distance error associated with incorrect trials which would have a lower theta power value? I think I just don't understand how the variables were input in the GLM.

Response: The reviewer's assessment is accurate in recognizing the relationship between "distance error" and "correct/incorrect" variables, where they are expected to exhibit opposing coefficients. We indeed observe this expected pattern when considering only the models where these variables emerge as significant predictors. For instance, in the model that focuses on the time period just before arrival at the recalled location (Supplementary Fig. 3a), the "correct/incorrect" variable demonstrates a significant positive coefficient. In the comprehensive retrieval model (Supplementary Fig. 3b), the coefficient for the "correct/incorrect" variable displays a negative mean value, but it lacks statistical significance. This is likely due to the heightened variability introduced by this variable's impact across the entirety of the retrieval trials, as the effects of correct/incorrect primarily influence key periods during retrieval (e.g., 0.5 s prior to arrival at the recalled location). To provide further clarity, we have added explanations on these variables (Methods, "Linear mixed effect model analysis" section):

"Models were constructed to describe theta band power fluctuations across four specific time periods during the task: (1) the final 0.5 s of retrieval trials prior to the button press, (2) the initial 0.5 s of retrieval trials following cue onset, (3) the complete duration of retrieval trials, and (4) arrow trials. In addition to movement-related variables, these models incorporated several task-related behavioral variables as follows: For model (1) and (3), distance to the recalled position when a button press was made ("distance to recalled position", measured in meters), distance to the closest boundary ("distance to boundary", measured in meters), distance between the button press location and halo position ("distance error", measured in meters) all as continuous variables, and correct versus incorrect trial performance ("correct/incorrect", treated as a categorical variable) were included. For model (2) all the variables present in (1) and (3) were included, while also introducing an additional variable: the distance from the cue onset position ("distance from cue", measured in meters). For model (4) the added behavioral variables were: "distance to boundary" and distance to the trial-specific arrow ("distance to error", measured in meters), both as continuous variables."

We also discuss the direction of the variable coefficients in the linear mixed model effects and consequent interpretation in the caption of Supplementary Fig. 3:

"Movement speed, angular velocity, distance to recalled position (distance in meters from the button press location), distance from cue (distance in meters to the retrieval cue), distance to boundary (distance in meters to the nearest wall), distance to arrow (distance in meters to the target arrow), and distance error (distance in meters between the recalled position and target location) were included as continuous variables. A significant negative impact of a distance variable (i.e., distance to recalled position, distance from cue, distance error, distance to arrow, and distance to boundary) reflects increased theta band power for shorter distances. Whether a trial was correct or incorrect (correct/incorrect) was a binary variable and movement direction was a categorical variable comprised of 12 possible binned movement directions (with the mean beta coefficient over all 12 directional bins depicted). A significant positive impact of the correct/incorrect variable signifies higher theta band power values observed for correct compared to incorrect trials."

C4: *Is this frequency band significant in the full spectral analysis (FDR corrected across all frequencies) that was originally used to filter the frequency range that was then used for the subsequent analyses? Adding this same type of figure for cue would add confidence to this result, which to me is more convincing than the .5 sec prior to button press.*

Response: We appreciate the reviewer for raising this point. In the full spectral analysis, significant effects of theta band power emerged exclusively within the high frequency theta range (6-8 Hz), concentrated within the 0.5-second interval preceding the button press. No such effects were observed during the same time interval following the cue onset after FDR correction. In light of these findings, we have included the detailed analyses of low frequency theta band power (3-6 Hz) in the Supplementary figures (Supplementary Fig. 3c, Supplementary Fig. 4c-f), while the main figures emphasize the results pertaining to the 6-8 Hz theta band power effects (see also C1, R1 above).

C5: *Does the comparison between correct and arrow trials survive FDR correction for this time period (figure 3). The x axis label of "time relative to halo recall" is a little misleading. In general, I think calling the button press the "instance of recall" is confusing, given the variability in time after cue. The Liu and Lifanov papers are not really about spatial navigation. The concern about the time window before retrieval is more about the behavior than about whether theta power differences may occur over short periods of time. Perhaps the authors could cite other spatial navigation experiments.*

Response: The comparison between correct and incorrect trials but not arrow trials survives FDR correction for the 0.25-0.5 second period after cue onset (Supplementary Fig. 4e-f):

Supplementary Fig. 4: Successful memory modulation of theta band power after cue and movement onset. Mean (\pm s.e.m.) norm'd (normalized) theta band power across MTL channels ($n_{\text{channels}} = 19$) (a,c) after cue presentation and (b,d) around movement onset (last movement onset prior to button press in each retrieval trial) in (a-b) 6-8 Hz and (c-d) 3-6 Hz band power. Note, halos were not visible during correct (green) or incorrect (red) retrieval trials. Gray bar indicates timepoints where $p < 0.05$ (one-sided permutation test at each time point, representing 4 ms steps at 250 Hz sampling rate). (e) Mean (\pm s.e.m.) norm'd theta band power across MTL channels ($n_{\text{channels}} = 19$). Vertical gray dotted line (time = 0) indicates trial onset when the instructional cue was presented to navigate to the target halo during retrieval trials, visible halo (visible, blue) during feedback, or arrow (orange) during arrow trials. (f) Mean (\pm s.e.m.) norm'd theta band power across MTL channels ($n_{\text{channels}} = 19$) during 0.25-0.5 s after cue for correct/incorrect trials, visible feedback periods, and arrow trials. MTL theta band power significantly increased during this period after cue for correct compared to incorrect and visible (feedback) trials. Crosses (+) represent the mean norm'd band power across all trials for an individual channel with each color corresponding to channels from a single participant. * = $p < 0.05$, ** = $p < 0.01$, *** = $p < 0.0001$, FDR corrected. Source data are provided as a Source Data file.

We have also revised the x-axis of Fig. 3i to state “Time relative to arrival at target (s)” based on the reviewer’s suggestion:

Fig. 3: MTL theta band power increased during correct compared to incorrect retrieval, visible halo, or arrow trials. (a-d) Mean (\pm s.e.m.) normalized (norm'd) power across MTL channels ($n_{\text{channels}} = 19$) for frequencies 3-120 Hz during the 0.5 s period prior to either the button press during (a) correct or (b) incorrect recall during retrieval trials, or (c) arrival at visible halos during feedback, or (d) arrival at arrows during arrow trials. MTL theta band power significantly increased during correct but not incorrect retrieval, visible halo, or arrow trials. Horizontal pink bar indicates significant power increase/decrease ($p < 0.05$, corrected using false discovery rate [FDR]^{19,20}). (e-h) Top-down view of norm'd theta band power (6-8 Hz) in an example MTL channel averaged across all samples during retrieval (e) correct, (f) incorrect, (g) visible halo, and (h) arrow trials. Note, halos were not visible during correct or incorrect retrieval nor during arrow trials. (e-f) Colored circles reflect all recalled locations during retrieval for correct and incorrect trials. (g) Colored circles reflect locations of halos during visible feedback. (i) Mean (\pm s.e.m.) norm'd theta band power across MTL channels ($n_{\text{channels}} = 19$). Vertical gray dotted line (time = 0) indicates the moment (button press) when participants arrived at the remembered halo position (correct/incorrect) during retrieval trials, visible halo (visible, blue) during feedback, or arrow (orange) during arrow trials. (j) Mean (\pm s.e.m.) norm'd theta band power across MTL channels ($n_{\text{channels}} = 19$) during correct/incorrect trials, visible feedback periods, and arrow trials. MTL theta band power significantly increased during the 0.5 s prior to arrival at the target position for correct compared to incorrect, visible (feedback), and arrow trials (correct vs. incorrect $p = 0.003$, correct vs. visible $p = 0.0004$, incorrect vs. visible $p = 0.19$, correct vs. arrow $p = 0.047$, arrow vs. incorrect $p = 0.28$, arrow vs. visible $p = 0.022$). Crosses (+) represent the mean norm'd band power across all trials for an individual channel with each color corresponding to channels from a single participant. * = $p < 0.05$, ** = $p < 0.01$, *** = $p < 0.0001$, 2-sided pairwise permutation test, FDR corrected. Source data are provided as a Source Data file.

We have incorporated the reviewer’s feedback by including references to spatial navigation tasks alongside non-spatial memory tasks that focus on memory retrieval (pg. 13):

“This pattern of results echoes previous findings in stationary humans, showing hippocampal reinstatement of low frequency theta oscillations during early retrieval time windows (specifically within the first 0.5 s after a retrieval cue was presented)²⁶, stronger representational similarity of iEEG activity in the 1 s prior to recall during remembered relative to forgotten trials²⁷, and increased theta power during spatial memory retrieval in view-based navigation tasks²⁸⁻³¹.”

C6: *Can the authors describe the normalization step in a little more detail? Was this carried out across all time samples in a long time series?*

Response: Yes, the normalization step was carried out across all time samples (after excluding samples with interictal epileptiform discharges (IED)) in the long time series. We have added clarification on this normalization step in our methods (Methods, “iEEG data analysis” section):

“This normalization procedure was performed for each recording channel separately, and involved initially computing the mean and standard deviation across the complete timeseries for each recording session. Subsequently, each individual data sample within the entirety of the recording session duration (with the exception of IED samples) was subjected to z-score transformation using the computed mean and standard deviation values. Recorded timeseries that were separated by longer breaks (more than ~40 minutes; e.g. before/after a participant's lunch break) were treated as independent recording sessions and normalized separately.”

C7: *I am overall impressed with how much work the authors have done. These data are complex to analyze. I think the addition of the lower frequency theta result when behavior is controlled is especially interesting.*

Response: We thank the reviewer for their positive comments and for their time to review the manuscript.

Reviewer #2 [R2] (Remarks to the Author): *I thank the authors for their detailed and considered response to my previous comments. I believe the manuscript is much improved. I have only one final comment regarding the statistical analyses, and two minor comments about presentation and terminology. I am happy to leave these changes to the authors discretion - I do not need to see the manuscript again.*

Response: We thank the reviewer for their positive comments.

C1: *The authors still present their statistical analyses across n=19 channels first, before going on (in some cases) to show that these findings are also significant when analysed correctly (i.e. either using data averaged across channels in each patient, to give n=6; or using a linear mixed effects model with patient ID as a random effect). I do not understand this approach, and I think it makes the paper appear statistically naive. All reviewers shared this concern in some form, as far as I can tell. So given that it doesn't seem to change the findings, why not just present the correct statistical analyses and remove all analyses across n=19 channels from the manuscript completely? At the very least, the authors should highlight if any of their findings are not observed across n=6 patients (i.e. for close v far in Fig 4a; across environments in Extended Figure 6b; boundary v inner in Fig 5b; towards v away from the boundary in Extended Figure 7d; and correct v incorrect excluding boundary positions in Extended Figure 1e).*

Response: While the results are significant when analyzing across participants (n=6) for most analyses (Fig. 3, 5, Supplementary Fig. 1c, Supplementary Fig. 2, Supplementary Fig. 3a, Supplementary Fig. 7d), it is not the case for all. We thus followed the reviewer's suggestion and now highlight any findings that were not observed across participants in the Methods (“Statistical comparisons” section):

“Statistical tests for main effects were conducted separately across channels and across participants (averaged the results across each participant's individual channels). Significant effects were observed in both analyses, except for the following instances where significance was only observed across channels but not across participants: Fig. 4a, Supplementary Fig. 6b, Supplementary Fig. 1e.”

C2: *I think 'bandpower' should be 'band power' throughout.*

Response: We have made this change throughout the manuscript.

C3: On page 7, when describing the linear mixed effects model, it would be useful to clarify whether 'distance to recall' and 'distance to button press' are truly referring to spatial variables (i.e. in m) or whether 'duration' would be more accurate (i.e. in s).

Response: To clarify, 'distance to recall' and 'distance to button press' were truly spatial variables (measured in meters). Based on R1, C3 we have also renamed this variable to "distance to recalled position" and added this clarification to the caption of Supplementary Fig. 3:

"Movement speed, angular velocity, distance to recalled position (distance in meters from the button press location), distance from cue (distance in meters from the retrieval cue), distance to boundary (distance in meters to the nearest wall), distance to arrow (distance in meters to the target arrow), and distance error (distance in meters between the recalled position and target location) were included as continuous variables."

Reviewer #3 [R3] (Remarks to the Author): *The authors have presented a comprehensive reply to the original reviews of their manuscript. They have sufficiently addressed the concerns and presented several new and control analyses that I feel have strengthened the overall conclusions. This is a good study and will be valuable for the field.*

Response: We thank the reviewer for their time and effort to review our manuscript, and for their positive feedback.